# Sharp Generalization for Nonparametric Regression by Over-Parameterized Neural Networks: A Distribution-Free Analysis in Spherical Covariate

## Abstract

Sharp generalization bound for neural networks trained by gradient descent (GD) is of central interest in statistical learning theory and deep learning. In this paper, we consider nonparametric regression by an over-parameterized two-layer NN trained by GD. We show that, if the neural network is trained by GD with early stopping, then the trained network renders a sharp rate of the nonparametric regression risk of $\mathcal{O}(\varepsilon_n^2)$, which is the same rate as that for the classical kernel regression trained by GD with early stopping, where $\varepsilon_n$ is the critical population rate of the Neural Tangent Kernel (NTK) associated with the network and $n$ is the size of the training data. It is remarked that our result does not require distributional assumptions on the covariate as long as the covariate lies on the unit sphere, in a strong contrast with many existing results which rely on specific distributions such as the spherical uniform data distribution or distributions satisfying certain restrictive conditions. As a special case of our general result, when the eigenvalues of the associated NTK decay at a rate of $\lambda_j \asymp j^{-\frac{d}{d-1}}$ for $j \geq 1$ which happens under certain distributional assumption such as the training features follow the spherical uniform distribution, we immediately obtain the minimax optimal rate of $\mathcal{O}(n^{-\frac{d}{2d-1}})$, which is the major results of several existing works in this direction. The neural network width in our general result is lower bounded by a function of only $d$ and $\varepsilon_n$, and such width does not depend on the minimum eigenvalue of the empirical NTK matrix whose lower bound usually requires additional assumptions on the training data. Our results are built upon two significant technical results which are of independent interest. First, uniform convergence to the NTK is established during the training process by GD, so that we can have a nice decomposition of the neural network function at any step of the GD into a function in the Reproducing Kernel Hilbert Space associated with the NTK and an error function with a small $L^\infty$-norm. Second, local Rademacher complexity is employed to tightly bound the Rademacher complexity of the function class comprising all the possible neural network functions obtained by GD. Our result formally fills the gap between training a classical kernel regression model and training an over-parameterized but finite-width neural network by GD for nonparametric regression without distributional assumptions about the spherical covariate.

## 1 Introduction

With the stunning success of deep learning in various areas of machine learning (LeCun et al., 2015), generalization analysis for neural networks is of central interest for statistical learning learning and deep learning. Considerable efforts have been made to analyze the optimization of deep neural networks showing that gradient descent (GD) and stochastic gradient descent (SGD) provably achieve vanishing training loss (Du et al., 2019b; Allen-Zhu et al., 2019b; Du et al., 2019a; Arora et al., 2019; Zou & Gu, 2019; Su & Yang, 2019). There are also extensive efforts devoted to generalization analysis of deep neural networks (DNNs) with algorithmic guarantees, that is, the generalization bounds

for neural networks trained by gradient descent or its variants. It has been shown that with sufficient over-parameterization, that is, with enough number of neurons in hidden layers, the training dynamics of deep neural networks (DNNs) can be approximated by that of a kernel method with the kernel induced by the neural network architecture, termed the Neural Tangent Kernel (NTK), while other studies such as (Yang & Hu, 2021) show that infinite-width neural networks can still learn features. The key idea of NTK based generalization analysis is that, for highly over-parameterized networks, the network weights almost remain around their random initialization. As a result, one can use the first-order Taylor expansion around initialization to approximate the neural network functions and analyze their generalization capability (Cao & Gu, 2019; Arora et al., 2019; Ghorbani et al., 2021).

Many existing works in generalization analysis of neural networks focus on clean data, but it is a central problem in statistical learning that how neural networks can obtain sharp convergence rates for the risk of nonparametric regression where the observed data are corrupted by noise. Considerable research has been conducted in this direction which shows that various types of DNNs achieve optimal convergence rates for smooth (Yarotsky, 2017; Bauer & Kohler, 2019; Schmidt-Hieber, 2020; Jiao et al., 2023; Zhang & Wang, 2023) or non-smooth (Imaizumi & Fukumizu, 2019) target functions for nonparametric regression. However, most of these works do not have algorithmic guarantees, that is, the DNNs in these works are constructed specially to achieve optimal rates with no guarantees that an optimization algorithm, such as GD or its variants, can obtain such constructed DNNs. To this end, efforts have been made in the literature to study the minimax optimal risk rates for nonparametric regression with over-parameterized neural networks trained by GD with either early stopping (Li et al., 2024) or $\ell^2$-regularization (Hu et al., 2021; Suh et al., 2022). However, most existing works either require spherical uniform data distribution on the unit sphere (Hu et al., 2021; Suh et al., 2022) or certain restrictive conditions on the data distribution.

It remains an interesting and important question for the statistical learning and theoretical deep learning literature that if an over-parameterized neural network trained by GD can achieve sharp risk rates for nonparametric regression with milder assumptions or restrictions on the distribution of the covariate, so that theoretical guarantees can be obtained for data in more practical scenarios. In this paper, we give a confirmative answer to this question. We present sharp risk rate for nonparametric regression with an over-parameterized two-layer NN trained by GD with early stopping, which is distribution-free in spherical covariate. Throughout this paper, distribution-free in spherical covariate means that there are no distributional assumptions about the covariate as long as the covariate lies on the unit sphere. Furthermore, our results give confirmative answers to certain open questions or address particular concerns in the literature of training over-parameterized neural networks by GD with early stopping for nonparametric regression with minimax optimal rates, such as the characterization of the stopping time in the early-stopping mechanism, the lower bound for the network width, and the constant learning rate used in GD. Benefiting from our analysis which is distribution-free in spherical covariate, our answers to these open questions or concerns do not require distributional assumptions about spherical covariate. Section 3 summarizes our main results with their significance and comparison to existing works.

We organize this paper as follows. We first introduce the necessary notations in the remainder of this section. We then introduce in Section 2 the problem setup for nonparametric regression. Our main results are summarized in Section 3 and detailed in Section 5. The training algorithm for the over-parameterized two-layer neural network is introduced in Section 4. The roadmap of proofs is presented in Section 6.

**Notations.** We use bold letters for matrices and vectors, and regular lower letter for scalars throughout this paper. The bold letter with a single superscript indicates the corresponding column of a matrix, e.g., $\mathbf{A}_i$ is the $i$-th column of matrix $\mathbf{A}$, and the bold letter with subscripts indicates the corresponding element of a matrix or vector. We put an arrow on top of a letter with subscript if it denotes a vector, e.g., $\vec{\mathbf{x}}_i$ denotes the $i$-th training feature. $\|\cdot\|_F$ and $\|\cdot\|_p$ denote the Frobenius norm and the vector $\ell^p$-norm or the matrix $p$-norm. $[m\colon n]$ denotes all the natural numbers between $m$ and $n$ inclusively, and $[1\colon n]$ is also written as $[n]$. $\mathrm{Var}\,[\cdot]$ denotes the variance of a random variable. $\mathbf{I}_n$ is a $n \times n$ identity matrix. $\mathbb{1}_{\{E\}}$ is an indicator function which takes the value of 1 if event $E$ happens, or 0 otherwise. The complement of a set $A$ is denoted by $A^c$, and $|A|$ is the cardinality of the set $A$. $\mathrm{vec}\,(\cdot)$ denotes the vectorization of a matrix or a set of vectors, and $\mathrm{tr}\,(\cdot)$ is the trace of a matrix. We denote the unit sphere in $d$-dimensional Euclidean space by $\mathbb{S}^{d-1} \coloneqq \{\mathbf{x}\colon \mathbf{x} \in \mathbb{R}^d, \|\mathbf{x}\|_2 = 1\}$. Let $L^2(\mathbb{S}^{d-1}, \mu)$ denote the space of square-integrable functions on $\mathbb{S}^{d-1}$ with probability mea-

sure $\mu$, and the inner product $\langle \cdot, \cdot \rangle_\mu$ and $\|\cdot\|_\mu^2$ are defined as $\langle f, g \rangle_{L^2} := \int_{\mathbb{S}^{d-1}} f(x)g(x)\mathrm{d}\mu(x)$ and $\|f\|_{L^2}^2 := \int_{\mathbb{S}^{d-1}} f^2(x)\mathrm{d}\mu(x) < \infty$. $\mathbf{B}(\mathbf{x}; r)$ is the Euclidean closed ball centered at $\mathbf{x}$ with radius $r$. Given a function $g: \mathbb{S}^{d-1} \to \mathbb{R}$, its $L^\infty$-norm is denoted by $\|g\|_\infty := \sup_{\mathbf{x} \in \mathbb{S}^{d-1}} |g(\mathbf{x})|$. $L^\infty$ is the function class whose elements almost surely have bounded $L^\infty$-norm. $\langle \cdot, \cdot \rangle_{\mathcal{H}}$ and $\|\cdot\|_{\mathcal{H}}$ denote the inner product and the norm in the Hilbert space $\mathcal{H}$. $a = \mathcal{O}(b)$ or $a \lesssim b$ indicates that there exists a constant $c > 0$ such that $a \leq cb$. $\tilde{\mathcal{O}}$ indicates there are specific requirements in the constants of the $\mathcal{O}$ notation. $a = o(b)$ and $a = w(b)$ indicate that $\lim |a/b| = 0$ and $\lim |a/b| = \infty$, respectively. $a \asymp b$ or $a = \Theta(b)$ denotes that there exists constants $c_1, c_2 > 0$ such that $c_1 b \leq a \leq c_2 b$. Throughout this paper we let the input space $\mathcal{X} = \mathbb{S}^{d-1}$, and $\mathrm{Unif}(\mathcal{X})$ denotes the uniform distribution on $\mathcal{X}$. The constants defined throughout this paper may change from line to line. For a Reproducing Kernel Hilbert Space $\mathcal{H}$, $\mathcal{H}(\mu_0)$ denotes the ball centered at the origin with radius $\mu_0$ in $\mathcal{H}$. We use $\mathbb{E}_P[\cdot]$ to denote the expectation with respect to the distribution $P$.

## 2 PROBLEM SETUP

We introduce the problem setups for nonparametric regression in this section.

### 2.1 TWO-LAYER NEURAL NETWORK

We are given the training data $\left\{ (\vec{\mathbf{x}}_i, y_i) \right\}_{i=1}^n$ where each data point is a tuple of feature vector $\vec{\mathbf{x}}_i \in \mathcal{X}$ and its response $y_i \in \mathbb{R}$. Throughout this paper we assume that no two training features coincide, that is, $\vec{\mathbf{x}}_i \neq \vec{\mathbf{x}}_j$ for all $i, j \in [n]$ and $i \neq j$. We denote the training feature vectors by $\mathbf{S} = \left\{ \vec{\mathbf{x}}_i \right\}_{i=1}^n$, and denote by $P_n$ the empirical distribution over $\mathbf{S}$. All the responses are stacked as a vector $\mathbf{y} = [y_1, \ldots, y_n]^\top \in \mathbb{R}^n$. The response $y_i$ is given by $y_i = f^*(\vec{\mathbf{x}}_i) + w_i$ for $i \in [n]$, where $\{w_i\}_{i=1}^n$ are i.i.d. sub-Gaussian random noise with mean 0 and variance proxy $\sigma_0^2$, that is, $\mathbb{E}[\exp(\lambda w_i)] \leq \exp(\lambda^2 \sigma_0^2 / 2)$ for any $\lambda \in \mathbb{R}$. $f^*$ is the target function to be detailed later. We define $\mathbf{y} := [y_1, \ldots, y_n]$, $\mathbf{w} := [w_1, \ldots, w_n]^\top$, and use $f^*(\mathbf{S}) := \left[ f^*(\vec{\mathbf{x}}_1), \ldots, f^*(\vec{\mathbf{x}}_n) \right]^\top$ to denote the clean target labels. The feature vectors in $\mathbf{S}$ are drawn i.i.d. according to an underlying unknown continuous data distribution $P$ with $\mu$ being the probability measure for $P$.

We consider a two-layer NN (NN) in this paper whose mapping function is

$$f(\mathbf{W}, \mathbf{x}) = \frac{1}{\sqrt{m}} \sum_{r=1}^m a_r \sigma\left( \vec{\mathbf{w}}_r^\top \mathbf{x} \right), \tag{1}$$

where $\mathbf{x} \in \mathcal{X}$ is the input, $\sigma(\cdot) = \max\{\cdot, 0\}$ is the ReLU activation function, $\mathbf{W} = \left\{ \vec{\mathbf{w}}_r \right\}_{r=1}^m$ with $\vec{\mathbf{w}}_r \in \mathbb{R}^d$ for $r \in [m]$ denotes the weighting vectors in the first layer and $m$ is the number of neurons. $\boldsymbol{a} = [a_1, \ldots, a_m] \in \mathbb{R}^m$ denotes the weights of the second layer. Throughout this paper we also write $\mathbf{W}$ as $\mathbf{W}_{\mathbf{S}}$ so as to indicate that the weighting vectors in $\mathbf{W}$ are trained on the training features $\mathbf{S}$.

### 2.2 KERNEL AND KERNEL REGRESSION FOR NONPARAMETRIC REGRESSION

We define the kernel function

$$K(\mathbf{u}, \mathbf{v}) := \frac{\langle \mathbf{u}, \mathbf{v} \rangle}{2\pi} (\pi - \arccos \langle \mathbf{u}, \mathbf{v} \rangle), \quad \forall \, \mathbf{u}, \mathbf{v} \in \mathcal{X}, \tag{2}$$

which is in fact the NTK associated with the two-layer NN (1), and $K$ is a positive semi-definite (PSD) kernel. Let the gram matrix of $K$ over the training data $\mathbf{S}$ be $\mathbf{K} \in \mathbb{R}^{n \times n}$, $\mathbf{K}_{ij} = K(\vec{\mathbf{x}}_i, \vec{\mathbf{x}}_j)$ for $i, j \in [n]$, and $\mathbf{K}_n := \mathbf{K}/n$ is the empirical NTK matrix. Let the eigendecomposition of $\mathbf{K}_n$ be $\mathbf{K}_n = \mathbf{U}\boldsymbol{\Sigma}\mathbf{U}^\top$ where $\mathbf{U}$ is a $n \times n$ orthogonal matrix, and $\boldsymbol{\Sigma}$ is a diagonal matrix with its diagonal elements $\left\{ \widehat{\lambda}_i \right\}_{i=1}^n$ being eigenvalues of $\mathbf{K}_n$ and sorted in a non-increasing order. It is proved in existing works, such as (Du et al., 2019b), that $\mathbf{K}_n$ is non-singular, and it can be verified

that $\widehat{\lambda}_1 \in (0, 1/2)$. Let $\mathcal{H}_K$ be the Reproducing Kernel Hilbert Space (RKHS) associated with $K$. Because $K$ is continuous on the compact set $\mathcal{X} \times \mathcal{X}$, the integral operator $T_K \colon L^2(\mathcal{X}, \mu) \to L^2(\mathcal{X}, \mu), (T_K f)(\mathbf{x}) \coloneqq \int_{\mathcal{X}} K(\mathbf{x}, \mathbf{x}') f(\mathbf{x}') \mathrm{d}\mu(\mathbf{x}')$ is a positive, self-adjoint, and compact operator on $L^2(\mathcal{X}, \mu)$. By the spectral theorem, there is a countable orthonormal basis $\{e_j\}_{j \geq 1} \subseteq L^2(\mathcal{X}, \mu)$ and $\{\lambda_j\}_{j \geq 1}$ with $\frac{1}{2} \geq \lambda_1 \geq \lambda_2 \geq \ldots > 0$ such that $e_j$ is the eigenfunction of $T_K$ with $\lambda_j$ being the corresponding eigenvalue. That is, $T_K e_j = \lambda_j e_j, j \geq 1$. Let $\{\mu_\ell\}_{\ell \geq 1}$ be the distinct eigenvalues associated with $T_K$, and let $m_\ell$ be the be the sum of multiplicity of the eigenvalue $\{\mu_{\ell'}\}_{\ell'=1}^{\ell}$. That is, $m_{\ell'} - m_{\ell'-1}$ is the multiplicity of $\mu_{\ell'}$. It is well known that $\left\{v_j = \sqrt{\lambda_j} e_j\right\}_{j \geq 1}$ is an orthonormal basis of $\mathcal{H}_K$. For a positive constant $\mu_0$, we define $\mathcal{H}_K(\mu_0) \coloneqq \{f \in \mathcal{H}_K \colon \|f\|_{\mathcal{H}} \leq \mu_0\}$ as the closed ball in $\mathcal{H}_K$ centered at 0 with radius $\mu_0$. We note that $\mathcal{H}_K(\mu_0)$ is also specified by $\mathcal{H}_K(\mu_0) = \left\{f \in L^2(\mathcal{X}, \mu) \colon f = \sum_{j=1}^{\infty} \beta_j e_j, \sum_{j=1}^{\infty} \beta_j^2 / \lambda_j \leq \mu_0^2\right\}$.

**The Task of Nonparametric Regression.** With $f^* \in \mathcal{H}_K(\mu_0)$, the task of the analysis for non-parametric regression is to find an estimator $\widehat{f}$ from the training data $\left\{(\vec{\mathbf{x}}_i, y_i)\right\}_{i=1}^{n}$ so that the risk $\mathbb{E}_P\left[\left(\widehat{f} - f^*\right)^2\right]$ can converge to 0 with a fast rate. In this work, we aim to establish a sharp rate of the risk where the over-parameterized neural network (1) trained by GD with early stopping serves as the estimator $\widehat{f}$.

**Sharp rate of the risk of nonparametric regression using classical kernel regression.** The statistical learning literature has established rich results in the sharp convergence rates for the risk of nonparametric kernel regression (Stone, 1985; Yang & Barron, 1999; Raskutti et al., 2014; Yuan & Zhou, 2016), with one representative result in (Raskutti et al., 2014) about kernel regression trained by GD with early stopping. Let $\varepsilon_n$ be the critical population rate of the PSD kernel $K$, which is also referred to as the critical radius (Wainwright, 2019) of $K$. (Raskutti et al., 2014, Theorem 2) shows the following sharp bound for the nonparametric regression risk of a kernel regression model trained by GD with early stopping when $f^* \in \mathcal{H}_K(\mu_0)$. That is, with probability at least $1 - \Theta\left(\exp(-\Theta(n\varepsilon_n^2))\right)$,

$$\mathbb{E}_P\left[\left(f_{\widehat{T}} - f^*\right)^2\right] \lesssim \varepsilon_n^2, \tag{3}$$

where $\widehat{T}$ is the stopping time whose formal definition is deferred to Section 5.1, and $f_{\widehat{T}}$ is the kernel regressor at the $\widehat{T}$-th step of GD for the optimization problem of kernel regression. The risk bound (3) is rather sharp, since it is minimax optimal in several popular learning setups, such as the setup where the eigenvalues $\{\lambda_i\}_{i \geq 1}$ exhibit a certain polynomial decay. Such risk bound (3) also holds for a general PSD kernel rather than the NTK (2), and the risk bound (3) is also minimax optimal when the PSD kernel is low rank. It is also remarked that the risk bound (3) is distribution-free in the bounded covariate, that is, there are no distributional assumptions about the covariate when it is in a bounded input space. Interested readers are referred to (Raskutti et al., 2014) for more details.

The main result of this paper is that the over-parameterized two-layer NN (1) trained by GD with early stopping achieves the same order of risk rate as that in (3) with arbitrary continuous distribution of the spherical covariate, which are summarized in the next section.

## 3 SUMMARY OF MAIN RESULTS.

Our main results are summarized in this section.

First, Theorem 5.1 in Section 5.2 shows that the neural network (1) trained by GD with early stopping using Algorithm 1 enjoys a sharp rate of the nonparametric regression risk, $\mathcal{O}\left(\varepsilon_n^2\right)$, which is the same as that for the classical kernel regression in (3). Such rate of nonparametric regression risk in Theorem 5.1 is distribution-free in spherical covariate, and it immediately leads to minimax optimal rates for certain special cases. For example, when the eigenvalues of the integral operator associated with $K$ has a particular polynomial eigenvalue decay rate (EDR), that is, $\lambda_j \asymp j^{-\frac{d}{d-1}}$ for $j \geq 1$, then in this case $\varepsilon_n^2 \asymp n^{-\frac{d}{2d-1}}$ according to (Raskutti et al., 2014, Corollary 3), and Theorem 5.1 renders the rate of the nonparametric regression risk of $\mathcal{O}(n^{-\frac{d}{2d-1}})$ which is minimax

Table 1: Comparison between our result and the existing works on the risk rates and assumptions for nonparametric regression by training over-parameterized neural networks with algorithmic guarantees, and the listed results here are under a common and popular setup that $f^* \in \mathcal{H}_{\tilde{K}}$ and the responses $\{y_i\}_{i=1}^n$ are corrupted by i.i.d. Gaussian noise with zero mean and variance $\sigma^2$.

| Existing Works and Our Result | Distributional Assumptions | Eigenvalue Decay Rate (EDR) | Rate of Nonparametric Regression Risk |
|---|---|---|---|
| (Kuzborskij & Szepesvári, 2021, Theorem 2) | No | – | Not minimax optimal, $\sigma^2 + \mathcal{O}(n^{\frac{-2}{2+d}})$ |
| (Hu et al., 2021, Theorem 5.2), (Suh et al., 2022, Theorem 3.11) | $P$ is Unif $(\mathcal{X})$ | $\lambda_j \asymp j^{-\frac{d}{d-1}}$ | minimax optimal, $\mathcal{O}(n^{\frac{-d}{2d-1}})$ |
| (Li et al., 2024, Proposition 13) | $P$ satisfies a restrictive condition: the density $p(\mathbf{x})$ for $\mathbf{x} \in \mathbb{R}^d$ satisfies $p(x) \lesssim (1 + \|\mathbf{x}\|_2^2)^{-(d+2)/2}$. | $\lambda_j \asymp j^{-\frac{d}{d-1}}$ | minimax optimal, $\mathcal{O}(n^{\frac{-d}{2d-1}})$ |
| Our Result (Theorem 5.1) | No distributional assumption about $P$ as long as $\mathcal{X} = \mathbb{S}^{d-1}$ | No requirement for EDR | $\mathcal{O}\left(\varepsilon_n^2\right)$, which leads to the minimax optimal rate $\mathcal{O}(n^{\frac{-d}{2d-1}})$ claimed in (Hu et al., 2021; Suh et al., 2022) and (Li et al., 2024) as special cases. |

optimal for this special case (Stone, 1985; Yang & Barron, 1999; Yuan & Zhou, 2016). We refer to such EDR the polynomial EDR in the sequel. It is shown in (Bietti & Mairal, 2019; Bietti & Bach, 2021; Li et al., 2024) that the polynomial EDR holds for our NTK in (2) if $P = \text{Unif}(\mathcal{X})$, or $P$ satisfies the distributional assumption for (Li et al., 2024, Proposition 13) in Table 1.

We remark that such a minimax optimal rate $\mathcal{O}(n^{-\frac{d}{2d-1}})$ is derived from Theorem 5.1 under the special case of polynomial EDR, and this minimax optimal rate is also the major result of a series of existing works in nonparametric regression by training over-parameterized neural networks (Hu et al., 2021; Suh et al., 2022; Li et al., 2024) when the target function $f^*$ belongs to $\mathcal{H}_{\tilde{K}}$, the RKHS associated with the NTK $\tilde{K}$ of the network in each particular existing work. We note that $\tilde{K}$ is the NTK of the network considered in a particular existing work which may not be the same as our NTK in (2). We also note that one needs to set $s = 1$ in (Li et al., 2024, Proposition 13) so that $f^* \in \mathcal{H}_{\tilde{K}}$, and in this case the risk rate for nonparametric regression in (Li et al., 2024, Proposition 13) is $\mathcal{O}(n^{-\frac{d}{2d-1}})$. To the best of our knowledge, Theorem 5.1 presents the first sharp risk rate for nonparametric regression which is distribution-free in spherical covariate, which is closer to practical scenarios. In contrast, the minimax rates in (Hu et al., 2021; Suh et al., 2022) require spherical uniform data distribution on $\mathcal{X}$. The recent work (Ko & Huo, 2024) also requires certain distributional assumptions for the results about regression convergence rates which does not have algorithmic guarantees. Although the minimax rate in another recent work (Li et al., 2024) does not need the spherical uniform distribution, it still requires a restrictive condition on the data distributions detailed in Table 1, and such condition is met by sub-Gaussian distributions. It is under this condition that (Li et al., 2024) derives the polynomial EDR. Table 1 compares our work to existing works for nonparametric regression with a common setup, that is, $f^* \in \mathcal{H}_{\tilde{K}}$ and the responses $\{y_i\}_{i=1}^n$ are corrupted by i.i.d. Gaussian noise. We further note that although the result in (Kuzborskij & Szepesvári, 2021, Theorem 2) does not require distributional assumptions about the covariate, its risk rate under this common setup is not minimax optimal due to the term $\sigma^2$ in the risk bound. Furthermore, the other term $\mathcal{O}(n^{\frac{-2}{2+d}})$ in its risk bound suffers from the curse of dimension with a slow rate to 0 for high-dimensional data. We also note that (Kuzborskij & Szepesvári, 2021, Theorem 1) shows the minimax optimal rate of $\mathcal{O}(n^{-\frac{2}{2+d}})$, however, this rate is derived for the noiseless case where the responses are not corrupted by noise.

Second, our results provide confirmative answers to several outstanding open questions or address particular concerns in the existing literature about training over-parameterized neural networks for nonparametric regression by GD with early stopping and sharp risk rates, which are detailed below.

**Stopping time in the early-stopping mechanism.** An open question raised in (Kuzborskij & Szepesvári, 2021; Hu et al., 2021) is how to characterize the stopping time in the early-stopping mechanism when training the over-parameterized network by GD. Let $\widehat{T}$ be the stopping time, (Li et al., 2024, Proposition 13) shows that the stopping time should satisfy $\widehat{T} \asymp n^{\frac{d}{2d-1}}$ under the distributional assumption in Table 1. In contrast, Theorem 5.1 provides a characterization of $\widehat{T}$ showing that $\widehat{T} \asymp \varepsilon_n^{-2}$, which is distribution-free in spherical covariate. Theorem 5.1 further suggests that for each neural network function $f_t$ obtained at the $t$-th step of GD with $t \asymp \varepsilon_n^{-2}$, the sharp risk rate of $\mathcal{O}\left(\varepsilon_n^2\right)$ is obtained.

**Lower bound for the network width** $m$**.** Our main result, Theorem 5.1, requires that the network width $m$, which is the number of neurons in the first layer of the network, satisfies $m \gtrsim d^2/(\varepsilon_n^{16})$. Such lower bound for $m$ solely depends on $d$ and $\varepsilon_n$. Under the polynomial EDR, Corollary 5.2, which is a direct consequence of Theorem 5.1, shows that $m$ should satisfy $m \gtrsim n^{\frac{16\alpha}{2\alpha+1}} d^2$ with $\alpha = d/(2(d-1))$ (see (12)) so that GD with early stopping leads to the minimax rate of $\mathcal{O}(n^{-\frac{d}{2d-1}})$. We remark that this is the first time that the lower bound for the network width $m$ is specified only in terms of $n$ and $d$ under the polynomial EDR with a minimax optimal risk rate for nonparametric regression, which can be easily estimated from the training data. In contrast, under the same polynomial EDR, all the existing works (Hu et al., 2021; Suh et al., 2022; Li et al., 2024) require $m \gtrsim \text{poly}(n, 1/\widehat{\lambda}_n)$. The problem here is that one needs additional assumptions on the training data (Bartlett et al., 2021; Nguyen et al., 2021) to find the lower bound for $\widehat{\lambda}_n$, which is the minimal eigenvalue of the empirical NTK matrix $\mathbf{K}_n$, to further estimate the lower bound for $m$ using the training data.

Corollary 5.2 also gives a competitive and smaller lower bound for the network width $m$ than some existing works which give explicit orders of the lower bound for $m$. For example, under the assumption of uniform spherical distribution, (Suh et al., 2022, Theorem 3.11) requires that $m/\log m \gtrsim L^{20}n^{24}$ where $L$ is the number of layers of the DNN used in that work, and $m/\log m \gtrsim 2^{20}n^{24}$ even with $L = 2$ for the two-layer network (1) used in our work. Furthermore, the proof of (Li et al., 2024, Proposition 13) suggests that $m \gtrsim n^{24}(\log m)^{12}$. Both lower bounds for $m$ in (Suh et al., 2022, Theorem 3.11) and (Li et al., 2024, Proposition 13) are much larger than our lower bound for $m$, $n^{\frac{16\alpha}{2\alpha+1}} d^2$, when $n \to \infty$ and $d$ is fixed, which is the setup considered in (Li et al., 2024). It is worthwhile to mention that (Suh et al., 2022; Li et al., 2024) use DNNs with multiple layers for nonparametric regression. As shown in Table 1, through our careful analysis, a shallow two-layer NN (1) exhibits the same minimax risk rate as its deeper counterpart under the same assumptions with much smaller network width. This observation further support the claim in (Bietti & Bach, 2021) that a shallow over-parameterized neural networks with ReLU activations exhibit the same approximation properties as its deeper counterpart, in our nonparametric regression setup.

**Training the network with learning rate** $\eta = \Theta(1)$**.** It is also worthwhile to mention that our main result, Theorem 5.1, suggests that a constant learning rate $\eta = \Theta(1)$ can be used for GD when training the two-layer NN (1), which could lead to better empirical optimization performance in practice. Some existing works in fact require an infinitesimal $\eta$. For example, (Li et al., 2024, Proposition 13) is obtained by gradient flow where $\eta \to 0$ instead of the practical GD. Furthermore, (Hu et al., 2021, Theorem 5.2) requires the learning rates for both the squared loss and the $\ell^2$-regularization term to have the order of $o(n^{-\frac{3d-1}{2d-1}}) \to 0$ as $n \to \infty$. We note that (Nitanda & Suzuki, 2021) also employs constant learning rate in SGD to train neural networks.

**More discussion about the literature.** We herein provide more discussion about the results of this work and comparison to the existing relevant works with sharp rates for nonparametric regression. While this paper establishes sharp rate which is distribution-free in spherical covariate, such rate still depends on bounded input space ($\mathcal{X} = \mathbb{S}^{d-1}$) and the condition that the target function $f^* \in \mathcal{H}_K(\mu_0)$. Some other existing works consider target function $f^*$ not belonging to the RKHS ball centered at the origin with constant or low radius, such as (Haas et al., 2023; Bordelon et al., 2024). A more detailed discussion is deferred to Section B of the appendix.

## 4   TRAINING BY GRADIENT DESCENT AND PRECONDITIONED GRADIENT DESCENT

In the training process of our network (1), only $\mathbf{W}$ is optimized with $\boldsymbol{a}$ randomly initialized to $\pm 1$ and then fixed. The following quadratic loss function is minimized during the training process:

$$L(\mathbf{W}) := \frac{1}{2n} \sum_{i=1}^{n} \left( f(\mathbf{W}, \vec{\mathbf{x}}_i) - y_i \right)^2. \qquad (4)$$

---

**Algorithm 1** Training the Two-Layer NN by GD

1: $\mathbf{W}(T) \leftarrow$ Training-by-GD$(T, \mathbf{W}(0))$
2: **input:** $T, \mathbf{W}(0)$
3: **for** $t = 1, \ldots, T$ **do**
4:      Perform the $t$-th step of GD by (5)
5: **end for**
6: **return** $\mathbf{W}(T)$

---

In the $(t + 1)$-th step of GD with $t \geq 0$, the weights of the neural network, $\mathbf{W_S}$, are updated by one-step of GD through

$$\operatorname{vec}\left(\mathbf{W_S}(t+1)\right) - \operatorname{vec}\left(\mathbf{W_S}(t)\right) = -\frac{\eta}{n}\mathbf{Z_S}(t)(\widehat{\mathbf{y}}(t) - \mathbf{y}), \tag{5}$$

where $\mathbf{y}_i = y_i, \widehat{\mathbf{y}}(t) \in \mathbb{R}^n$ with $[\widehat{\mathbf{y}}(t)]_i = f(\mathbf{W}(t), \overrightarrow{\mathbf{x}}_i)$. The notations with the subscripts $\mathbf{S}$ indicate the dependence on the training features $\mathbf{S}$. We also denote $f(\mathbf{W}(t), \cdot)$ as $f_t(\cdot)$ as the neural network function with weighting vectors $\mathbf{W}(t)$ obtained after the $t$-th step of GD. We define $\mathbf{Z_S}(t) \in \mathbb{R}^{md \times n}$ which is computed by

$$(\mathbf{Z_S}(t))_{[(r-1)d+1:rd]i} = \frac{1}{\sqrt{m}}\mathbb{1}_{\left\{\overrightarrow{\mathbf{w}}_r(t)^\top \overrightarrow{\mathbf{x}}_i \geq 0\right\}}\overrightarrow{\mathbf{x}}_i a_r, \ i \in [n], r \in [m], \tag{6}$$

where $(\mathbf{Z_S}(t))_{[(r-1)d+1:rd]i} \in \mathbb{R}^d$ is a vector with elements in the $i$-th column of $\mathbf{Z_S}(t)$ with indices in $[(r-1)d+1 : rd]$. We employ the following particular symmetric random initialization so that $\widehat{\mathbf{y}}(0) = \mathbf{0}$, which has been used in existing works such as (Chizat et al., 2019; Zhang et al., 2020). In our two-layer NN, $m$ is even, $\left\{\overrightarrow{\mathbf{w}}_{2r'}(0)\right\}_{r'=1}^{m/2}$ and $\{a_{2r'}\}_{r'=1}^{m/2}$ are initialized randomly and independently according to

$$\overrightarrow{\mathbf{w}}_{2r'}(0) \sim \mathcal{N}(\mathbf{0}, \kappa^2\mathbf{I}_d), a_{2r'} \sim \operatorname{unif}\left(\{-1, 1\}\right), \quad \forall r' \in [m/2], \tag{7}$$

where $\mathcal{N}(\boldsymbol{\mu}, \boldsymbol{\Sigma})$ denotes a Gaussian distribution with mean $\boldsymbol{\mu}$ and covariance $\boldsymbol{\Sigma}$, $\operatorname{unif}\left(\{-1, 1\}\right)$ denotes a uniform distribution over $\{1, -1\}$, $0 < \kappa \leq 1$ controls the magnitude of initialization. We set $\overrightarrow{\mathbf{w}}_{2r'-1}(0) = \overrightarrow{\mathbf{w}}_{2r'}(0)$ and $a_{2r'-1} = -a_{2r}$ for all $r' \in [m/2]$. It then can be verified that $\widehat{\mathbf{y}}(0) = \mathbf{0}$, that is, the initial output of the two-layer network (1) is zero. Once randomly initialized, $\boldsymbol{a}$ is fixed during the training. We use $\mathbf{W}(0)$ to denote the set of all the random weighting vectors at initialization, that is, $\mathbf{W}(0) = \left\{\overrightarrow{\mathbf{w}}_r(0)\right\}_{r=1}^{m}$. We run Algorithm 1 to train the two-layer NN by GD, where $T$ is the total number of steps for GD. Early stopping is enforced in Algorithm 1 through a bounded $T$ via $T \leq \widehat{T}$.

## 5 MAIN RESULTS

We present the definition of kernel complexity in this section, and then introduce the main results for nonparametric regression of this paper.

### 5.1 KERNEL COMPLEXITY

The local kernel complexity has been studied by (Bartlett et al., 2005; Koltchinskii, 2006; Mendelson, 2002). For the PSD kernel $K$, we define the empirical kernel complexity $\widehat{R}_K$ and the population kernel complexity $R_K$ as

$$\widehat{R}_K(\varepsilon) := \sqrt{\frac{1}{n}\sum_{i=1}^{n}\min\left\{\widehat{\lambda}_i, \varepsilon^2\right\}}, \quad R_K(\varepsilon) := \sqrt{\frac{1}{n}\sum_{i=1}^{\infty}\min\left\{\lambda_i, \varepsilon^2\right\}}. \tag{8}$$

It can be verified that both $\sigma R_K(\varepsilon)$ and $\sigma\widehat{R}_K(\varepsilon)$ are sub-root functions (Bartlett et al., 2005) in terms of $\varepsilon^2$. The formal definition of sub-root functions is deferred to Definition A.2 in the appendix. For a given noise ratio $\sigma$, the critical empirical radius $\widehat{\varepsilon}_n > 0$ is the smallest positive solution to the inequality $\widehat{R}_K(\varepsilon) \leq \varepsilon^2/\sigma$, where $\widehat{\varepsilon}_n^2$ is the also the fixed point of $\sigma\widehat{R}_K(\varepsilon)$ as a function of $\varepsilon^2$: $\sigma\widehat{R}_K(\widehat{\varepsilon}_n) = \widehat{\varepsilon}_n^2$. Similarly, the critical population rate $\varepsilon_n$ is defined to be the smallest positive solution to the inequality $R_K(\varepsilon) \leq \varepsilon^2/\sigma$, where $\varepsilon_n^2$ is the fixed point of $\sigma\widehat{R}_K(\varepsilon)$ as a function of $\varepsilon^2$: $\sigma R_K(\varepsilon_n) = \varepsilon_n^2$. In this paper we consider the case that $n\varepsilon_n^2 \to \infty$ as $n \to \infty$, which is also used in standard analysis of nonparametric regression with minimax rates by kernel regression (Raskutti et al., 2014).

Let $\eta_t := \eta t$ for all $t \geq 0$, we then define the stopping time $\widehat{T}$ as

$$\widehat{T} := \min \left\{ t \colon \widehat{R}_K(\sqrt{1/\eta_t}) > (\sigma\eta_t)^{-1} \right\} - 1. \tag{9}$$

The stopping time in fact limit the number of steps $T$ in for Algorithm 1 as to be shown in Section 5.2, which in turn enforces the early stopping mechanism.

## 5.2 RESULTS

**Theorem 5.1.** Let $c_T, c_t \in (0, 1]$ be arbitrary positive constants, and $c_T\widehat{T} \leq T \leq \widehat{T}$. Suppose $f^* \in \mathcal{H}_K(\mu_0)$, and $m$ satisfies

$$m \gtrsim \frac{d^2}{\varepsilon_n^{16}}, \tag{10}$$

and the neural network $f(\mathbf{W}(t), \cdot)$ is trained by GD using Algorithm 1 with the learning rate $\eta \in [1, 2)$ and $T \leq \widehat{T}$. Then for every $t \in [c_tT \colon T]$, with probability at least $1 - \exp(-\Theta(n)) - 7\exp(-\Theta(n\varepsilon_n^2)) - 2/n$ over the random noise $\mathbf{w}$, the random training features $\mathbf{S}$ and the random initialization $\mathbf{W}(0)$, the stopping time satisfies $\widehat{T} \asymp \varepsilon_n^{-2}$, and $f(\mathbf{W}(t), \cdot) = f_t$ satisfies

$$\mathbb{E}_P \left[ (f_t - f^*)^2 \right] \lesssim \varepsilon_n^2. \tag{11}$$

**Significance of Theorem 5.1 and comparison to existing works.** To the best of our knowledge, Theorem 5.1 is the first theoretical result which proves that over-parameterized neural network trained by gradient descent with early stopping achieves sharp rate of $\mathcal{O}(\varepsilon_n^2)$, *without distributional assumption on the covariate* as long as the input space $\mathcal{X}$ is $\mathbb{S}^{d-1}$. To understand the sharpness of the bound for the risk in (11), Corollary 5.2 shows that when the polynomial EDR holds, that is, $\lambda_j \asymp j^{-\frac{d}{d-1}}$, then $\varepsilon_n^2 \asymp n^{-\frac{d}{2d-1}}$, and the rate of the risk is $\mathcal{O}(n^{-\frac{d}{2d-1}})$ which is minimax optimal under the polynomial EDR for $f^* \in \mathcal{H}_K(\mu_0)$ (Stone, 1985; Yang & Barron, 1999; Yuan & Zhou, 2016). (Bietti & Mairal, 2019; Bietti & Bach, 2021; Li et al., 2024) show that such polynomial EDR holds for our NTK (2) if $P$ is $\mathrm{Unif}(\mathcal{X})$, or $P$ satisfies the distributional assumption for (Li et al., 2024, Proposition 13) in Table 1. The existing works (Hu et al., 2021; Suh et al., 2022; Li et al., 2024) prove the same minimax optimal rate for an over-parameterized neural network trained by GD with either regularization or early stopping. However, it is remarked that such minimax optimal rates in these works are proved either for spherical uniform distribution on $\mathcal{X}$ (Hu et al., 2021; Suh et al., 2022), or for distributions satisfying certain restrictive condition (Li et al., 2024). Table 1 compares our result to existing works from the perspective of risk rates for nonparametric regression, required distributional assumptions on the covariate, and the associated EDR.

We also emphasize that Theorem 5.1, for the first time, shows that the network width $m$ required to achieve the minimax rate can be quantized in terms of a well known quantity about the kernel $K$, the critical population rate $\varepsilon_n$, and $d$ in the manner of distribution-free in spherical covariate. More discussions are referred to "**Significance of Corollary 5.2**".

Furthermore, Theorem 5.1, for the first time, gives an explicit characterization of the stopping time $\widehat{T}$ for training an over-parameterized neural network by GD with early stopping which is of the order $\widehat{T} \asymp \varepsilon_n^{-2}$ and distribution-free in spherical covariate. This result suggests that $t$ should be of the order $\Theta(\varepsilon_n^{-2})$ to ensures the sharp rate (11). Such result gives an order of the number of steps for GD when training the over-parameterized NN (1) so as to achieve the sharp risk bound $\mathcal{O}(\varepsilon_n^2)$. Under the polynomial EDR, the stopping time $\widehat{T}$ satisfies $\widehat{T} \asymp n^{\frac{d}{2d-1}}$, which recovers the same result about the stopping time in (Li et al., 2024, Proposition 13).

When the polynomial EDR holds, we can apply Theorem 5.1 to obtain the following corollary.

**Corollary 5.2** (Applying Theorem 5.1 to the special case of polynomial EDR)**.** Suppose $\lambda_j \asymp j^{-2\alpha}$ for $j \geq 1$ and $\alpha > 1/2$. Let $c_T, c_t \in (0, 1]$ be positive constants, and $c_T\widehat{T} \leq T \leq \widehat{T}$. Suppose $m$ satisfies

$$m \gtrsim n^{\frac{16\alpha}{2\alpha+1}} d^2, \tag{12}$$

and the neural network $f(\mathbf{W}(t), \cdot)$ is trained by GD using Algorithm 1 with the learning rate $\eta \in [1, 2)$ and $T \leq \widehat{T}$. Then for every $t \in [c_tT \colon T]$, with probability at least $1 - \exp(-\Theta(n)) -$

$7 \exp\left(-\Theta(n\varepsilon_n^2)\right) - 2/n$ over the random noise $\mathbf{w}$, the random training features $\mathbf{S}$ and the random initialization $\mathbf{W}(0)$, the stopping time satisfies $\widehat{T} \asymp n^{\frac{d}{2d-1}}$,

$$\mathbb{E}_P\left[(f_t - f^*)^2\right] \lesssim \left(\frac{1}{n}\right)^{\frac{2\alpha}{2\alpha+1}}. \tag{13}$$

**Significance of Corollary 5.2.** Corollary 5.2 shows that under the polynomial EDR, GD finds an over-parameterized neural network with minimax optimal rate of $\mathcal{O}(n^{-\frac{2\alpha}{2\alpha+1}}) = \mathcal{O}(n^{-\frac{d}{2d-1}})$, where $\alpha = d/(2(d-1))$, with a specific quantization of $m$ in terms of only $n$ and $d$ in (12). In contrast, all the existing works (Hu et al., 2021; Suh et al., 2022; Li et al., 2024) require $m \gtrsim \text{poly}(n, 1/\widehat{\lambda}_n)$, and additional assumptions on the training data (Bartlett et al., 2021; Nguyen et al., 2021) are required to bound $\widehat{\lambda}_n$ from below so as to estimate the lower bound for $m$ from the training data.

# 6 ROADMAP OF PROOFS

We present the roadmap of our theoretical results which lead to the main result, Theorem 5.1 in Section 5. We first present in the next subsection our results about the uniform convergence to the NTK (2) and more, which are crucial in the analysis of training dynamics by GD.

## 6.1 UNIFORM CONVERGENCE TO THE NTK AND MORE

We define functions

$$h(\mathbf{w}, \mathbf{x}, \mathbf{y}) := \mathbf{x}^\top \mathbf{y} \mathbb{1}_{\{\mathbf{w}^\top \mathbf{x} \geq 0\}} \mathbb{1}_{\{\mathbf{w}^\top \mathbf{y} \geq 0\}}, \qquad \widehat{h}(\mathbf{W}, \mathbf{x}, \mathbf{y}) := \frac{1}{m}\sum_{r=1}^m h(\vec{\mathbf{w}}_r, \mathbf{x}, \mathbf{y}), \tag{14}$$

$$v_R(\mathbf{w}, \mathbf{x}) := \mathbb{1}_{\{|\mathbf{w}^\top \mathbf{x}| \leq R\}}, \qquad \widehat{v}_R(\mathbf{W}, \mathbf{x}) := \frac{1}{m}\sum_{r=1}^m v_R(\vec{\mathbf{w}}_r, \mathbf{x}). \tag{15}$$

Then we have the following theorem stating the uniform convergence of $\widehat{h}(\mathbf{W}(0), \cdot, \cdot)$ to $K(\cdot, \cdot)$ and uniform convergence of $\widehat{v}_R(\mathbf{W}(0), \mathbf{x})$ to $\frac{2R}{\sqrt{2\pi}\kappa}$ for a positive number $R \lesssim \eta T/\sqrt{m}$. While existing works such as (Li et al., 2024) also has uniform convergence results for over-parameterized neural network, our result does not depend on the Hölder continuity of the NTK.

**Theorem 6.1.** The following results hold with $m \geq \max\{d, n, 4\}$ and $m/\log m \geq d$.

(1) With probability at least $1 - 1/n$ over the random initialization $\mathbf{W}(0) = \left\{\vec{\mathbf{w}}_r(0)\right\}_{r=1}^m$,

$$\sup_{\mathbf{x}\in\mathcal{X}, \mathbf{y}\in\mathcal{X}} \left|K(\mathbf{x}, \mathbf{y}) - \widehat{h}(\mathbf{W}(0), \mathbf{x}, \mathbf{y})\right| \leq C_1(m, d, 1/n) \lesssim \sqrt{\frac{d\log m}{m}}. \tag{16}$$

(2) Suppose $m \geq (c_\mathbf{u}\eta T/R_0)^2$ for an arbitrary absolute positive constant $R_0 < \kappa$. Then with probability at least $1 - 1/n$ over the random initialization $\mathbf{W}(0) = \left\{\vec{\mathbf{w}}_r(0)\right\}_{r=1}^m$,

$$\sup_{\mathbf{x}\in\mathcal{X}} |\widehat{v}_R(\mathbf{W}(0), \mathbf{x})| \leq \frac{2R}{\sqrt{2\pi}\kappa} + C_2(m, R_0, 1/n) \lesssim \frac{\sqrt{d}}{m^{1/4}} + \frac{\eta T}{\sqrt{m}}, \tag{17}$$

where $C_1(m, d, 1/n), C_2(m, R_0, 1/n)$ are two positive numbers depending on $(m, d, n)$ and $(m, R_0, n)$, respectively, with their formal definitions deferred to (39) and (41) in Section C.2 of the appendix.

*Proof.* This theorem follows from Theorem C.1 and Theorem C.2 in Section C.2 of the appendix. Note that $\widehat{h}(\mathbf{W}, \mathbf{x}, \mathbf{y}) = \frac{1}{m}\sum_{r=1}^m h(\vec{\mathbf{w}}_r, \mathbf{x}, \mathbf{y}) = \frac{1}{m/2}\sum_{r'=1}^{m/2} h(\vec{\mathbf{w}}_{2r}(0), \mathbf{x}, \mathbf{y})$, then part (1) directly follows from Theorem C.1. Similarly, part (2) directly follows from Theorem C.2, and noting that $m \geq (c_\mathbf{u}\eta T/R_0)^2$ indicates $R \leq R_0$. $\qquad\square$

Define

$$\mathcal{W}_0 \coloneqq \{\mathbf{W}(0) \colon (16), (17) \text{ hold}\} \tag{18}$$

be the set of all the good random initializations which satisfy (16) and (17) in Theorem 6.1. Theorem 6.1 shows that we have good random initialization with high probability, that is, $\Pr[\mathbf{W}(0) \in \mathcal{W}_0] \geq 1 - 2/n$. When $\mathbf{W}(0) \in \mathcal{W}_0$, the uniform convergence results, (16) and (17), hold with high probability, which is crucial for our main result in Theorem 5.1.

## 6.2 ROADMAP OF PROOFS

Because our main result, Theorem 5.1, is proved by Theorem C.10 and Theorem C.11 deferred to Section C.2, we illustrate in Figure 1, deferred to the appendix, the roadmap containing the intermediate theoretical results which lead to our main result, Theorem 5.1.

**Summary of the technical approaches and novel results in the proofs.** Theorem C.8 is the first novel result in this work, showing that with high probability, the neural network function $f(\mathbf{W}(t), \cdot)$ at step $t$ of GD can be decomposed into two functions by $f(\mathbf{W}(t), \cdot) = f_t = h + e$, where $h \in \mathcal{H}_K$ is a function in the RKHS associated with $K$ with bounded $\mathcal{H}_K$-norm. The error function $e$ has a small $L^\infty$-norm, that is, $\|e\|_\infty \leq w$ with $w$ being a small number controlled by the network width $m$, that is, larger $m$ leads to smaller $w$. Theorem C.10 is the second novel result, where we derive sharp and novel bound for the nonparametric regression risk of the neural network function $f(\mathbf{W}(t), \cdot)$ in Theorem C.10, that is, $\mathbb{E}_P\left[(f_t - f^*)^2\right] - 2\mathbb{E}_{P_n}\left[(f_t - f^*)^2\right] \lesssim \varepsilon_n^2 + w$. To the best of our knowledge, Theorem C.10 is among the first in the literature to employ local Rademacher complexity so as to obtain sharp rate for the risk of nonparametric regression which is distribution-free in spherical covariate, and local Rademacher complexity is employed to tightly bound the Rademacher complexity of the function class comprising all the possible neural network functions obtained by GD.

**Novel proof strategy of this work.** We remark that the proof strategy of our main result, Theorem 5.1, is significantly novel and different from the existing works in training over-parameterized neural networks for nonparametric regression with minimax rates (Hu et al., 2021; Suh et al., 2022; Li et al., 2024). In particular, the common proof strategy in these works uses the decomposition $f_t - f^* = (f_t - \widehat{f}_t^{(\text{NTK})}) + (\widehat{f}_t^{(\text{NTK})} - f^*)$ and then show that both $\left\|f_t - \widehat{f}_t^{(\text{NTK})}\right\|_{L^2}$ and $\left\|\widehat{f}_t^{(\text{NTK})} - f^*\right\|_{L^2}$ are bounded by certain minimax optimal rate, where $\widehat{f}_t^{(\text{NTK})}$ is the kernel regressor obtained by either kernel ridge regression (Hu et al., 2021; Suh et al., 2022) or GD with early stopping (Li et al., 2024). The remark after Theorem C.8 details a formulation of $\widehat{f}_t^{(\text{NTK})}$. $\left\|\widehat{f}_t^{(\text{NTK})} - f^*\right\|_{L^2}$ is bounded by the minimax optimal rate under certain distributional assumptions in the covariate, and this is one reason for the distributional assumptions about the covariate in existing works such as (Hu et al., 2021; Suh et al., 2022; Li et al., 2024). In a strong contrast, our analysis does not rely on such decomposition of $f_t - f^*$. Instead of approximating $f_t$ by $\widehat{f}_t^{(\text{NTK})}$, we have a new decomposition of $f_t$ by $f_t = h_t + e_t$ where $f_t$ is approximated by $h_t$ with $e_t$ being the approximation error. As suggested by the remark after Theorem C.8, we have $h_t = \widehat{f}_t^{(\text{NTK})} + \widehat{e}_2(\cdot, t)$ so that $f_t = \widehat{f}_t^{(\text{NTK})} + \widehat{e}_2(\cdot, t) + e_t$. Our analysis only requires the network width $m$ to be suitably large so that the $\mathcal{H}_K$-norm of $\widehat{e}_2(\cdot, t)$ is bounded by a positive constant and $\|e_t\|_\infty \leq w$, while the common proof strategy in(Hu et al., 2021; Suh et al., 2022; Li et al., 2024) needs $m$ to be sufficiently large so that both $\|\widehat{e}_2(\cdot, t)\|_\infty$ and $\|e_t\|_\infty$ are bounded by an infinitesimal number (a minimax optimal rate such as $\mathcal{O}(n^{-\frac{d}{2d-1}})$) and then $\left\|f_t - \widehat{f}_t^{(\text{NTK})}\right\|_{L^2}$ is bounded by such minimax optimal rate. Detailed in Section 3, such novel proof strategy leads to our sharp analysis, rendering a smaller lower bound for $m$ in our main result compared to some existing works.

## 7 CONCLUSION

In this paper, we show that an over-parameterized two-layer neural network trained by gradient descent (GD) with early stopping renders a sharp rate of the nonparametric regression risk with the order of $\Theta(\varepsilon_n^2)$ with $\varepsilon_n$ being the critical population rate or the critical radius of the NTK, which is distribution-free in spherical covariate. We compare our results to the current state-of-the-art with a detailed roadmap of our technical approaches and results in our proofs.

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

We present the basic mathematical results required in our proofs in Section A, then present proofs in the subsequent sections.

## A  MATHEMATICAL TOOLS

We introduce the basic definitions and mathematical results as the basic tools for the subsequent results in the next sections of this appendix.

*Definition* A.1. Let $\{\sigma_i\}_{i=1}^n$ be $n$ i.i.d. random variables such that $\Pr[\sigma_i = 1] = \Pr[\sigma_i = -1] = \frac{1}{2}$. The Rademacher complexity of a function class $\mathcal{F}$ is defined as

$$\mathfrak{R}(\mathcal{F}) = \mathbb{E}_{\left\{\vec{\mathbf{x}}_i\right\}_{i=1}^n, \{\sigma_i\}_{i=1}^n} \left[\sup_{f \in \mathcal{F}} \frac{1}{n} \sum_{i=1}^n \sigma_i f(\vec{\mathbf{x}}_i)\right]. \tag{19}$$

The empirical Rademacher complexity is defined as

$$\widehat{\mathfrak{R}}(\mathcal{F}) = \mathbb{E}_{\{\sigma_i\}_{i=1}^n} \left[ \sup_{f \in \mathcal{F}} \frac{1}{n} \sum_{i=1}^n \sigma_i f(\vec{\mathbf{x}}_i) \right], \tag{20}$$

For simplicity of notations, Rademacher complexity and empirical Rademacher complexity are also denoted by $\mathbb{E}\left[\sup_{f \in \mathcal{F}} \frac{1}{n} \sum_{i=1}^n \sigma_i f(\vec{\mathbf{x}}_i)\right]$ and $\mathbb{E}_\sigma \left[\sup_{f \in \mathcal{F}} \frac{1}{n} \sum_{i=1}^n \sigma_i f(\vec{\mathbf{x}}_i)\right]$ respectively.

For data $\left\{\vec{\mathbf{x}}\right\}_{i=1}^n$ and a function class $\mathcal{F}$, we define the notation $R_n \mathcal{F}$ by $R_n \mathcal{F} := \sup_{f \in \mathcal{F}} \frac{1}{n} \sum_{i=1}^n \sigma_i f(\vec{\mathbf{x}}_i)$.

**Theorem A.1** ((Bartlett et al., 2005, Theorem 2.1)). Let $\mathcal{X}, P$ be a probability space, $\left\{\vec{\mathbf{x}}_i\right\}_{i=1}^n$ be independent random variables distributed according to $P$. Let $\mathcal{F}$ be a class of functions that map $\mathcal{X}$ into $[a, b]$. Assume that there is some $r > 0$ such that for every $f \in \mathcal{F}, \mathrm{Var}\left[f(\vec{\mathbf{x}}_i)\right] \leq r$. Then, for every $x > 0$, with probability at least $1 - e^{-x}$,

$$\sup_{f \in \mathcal{F}} \left(\mathbb{E}_P[f(\mathbf{x})] - \mathbb{E}_{\mathbf{x} \sim P_n}[f(\mathbf{x})]\right) \leq \inf_{\alpha > 0} \left(2(1 + \alpha)\mathbb{E}_{\{\vec{\mathbf{x}}_i\}_{i=1}^n, \{\sigma_i\}_{i=1}^n}[R_n \mathcal{F}] + \sqrt{\frac{2rx}{n}}\right.$$

$$\left. + (b - a)\left(\frac{1}{3} + \frac{1}{\alpha}\right)\frac{x}{n}\right), \tag{21}$$

and with probability at least $1 - 2e^{-x}$,

$$\sup_{f \in \mathcal{F}} \left(\mathbb{E}_P[f(\mathbf{x})] - \mathbb{E}_{\mathbf{x} \sim P_n}[f(\mathbf{x})]\right) \leq \inf_{\alpha \in (0,1)} \left(\frac{2(1 + \alpha)}{1 - \alpha}\mathbb{E}_{\{\sigma_i\}_{i=1}^n}[R_n \mathcal{F}] + \sqrt{\frac{2rx}{n}}\right.$$

$$\left. + (b - a)\left(\frac{1}{3} + \frac{1}{\alpha} + \frac{1 + \alpha}{2\alpha(1 - \alpha)}\right)\frac{x}{n}\right). \tag{22}$$

$P_n$ is the empirical distribution over $\left\{\vec{\mathbf{x}}_i\right\}_{i=1}^n$ with $\mathbb{E}_{\mathbf{x} \sim P_n}[f(\mathbf{x})] = \frac{1}{n} \sum_{i=1}^n f(\vec{\mathbf{x}}_i)$. Moreover, the same results hold for $\sup_{f \in \mathcal{F}} \left(\mathbb{E}_{\mathbf{x} \sim P_n}[f(\mathbf{x})] - \mathbb{E}_P[f(\mathbf{x})]\right)$.

In addition, we have the contraction property for Rademacher complexity, which is due to Ledoux and Talagrand (Ledoux, 1991).

**Theorem A.2.** Let $\phi$ be a contraction, that is, $|\phi(x) - \phi(y)| \leq \mu |x - y|$ for $\mu > 0$. Then, for every function class $\mathcal{F}$,

$$\mathbb{E}_{\{\sigma_i\}_{i=1}^n}[R_n \phi \circ \mathcal{F}] \leq \mu \mathbb{E}_{\{\sigma_i\}_{i=1}^n}[R_n \mathcal{F}], \tag{23}$$

where $\phi \circ \mathcal{F}$ is the function class defined by $\phi \circ \mathcal{F} = \{\phi \circ f : f \in \mathcal{F}\}$.

*Definition* A.2 (Sub-root function,(Bartlett et al., 2005, Definition 3.1)). A function $\psi \colon [0, \infty) \to [0, \infty)$ is sub-root if it is nonnegative, nondecreasing and if $\frac{\psi(r)}{\sqrt{r}}$ is nonincreasing for $r > 0$.

**Theorem A.3** ((Bartlett et al., 2005, Theorem 3.3)). Let $\mathcal{F}$ be a class of functions with ranges in $[a, b]$ and assume that there are some functional $T \colon \mathcal{F} \to \mathbb{R}+$ and some constant $\bar{B}$ such that for every $f \in \mathcal{F}$, $\mathrm{Var}[f] \leq T(f) \leq \bar{B}P(f)$. Let $\psi$ be a sub-root function and let $r^*$ be the fixed point of $\psi$. Assume that $\psi$ satisfies, for any $r \geq r^*$, $\psi(r) \geq \bar{B}\mathfrak{R}(\{f \in \mathcal{F} : T(f) \leq r\})$. Fix $x > 0$, then for any $K_0 > 1$, with probability at least $1 - e^{-x}$,

$$\forall f \in \mathcal{F}, \quad \mathbb{E}_P[f] \leq \frac{K_0}{K_0 - 1}\mathbb{E}_{P_n}[f] + \frac{704K_0}{\bar{B}}r^* + \frac{x\left(11(b - a) + 26\bar{B}K_0\right)}{n}.$$

Also, with probability at least $1 - e^{-x}$,

$$\forall f \in \mathcal{F}, \quad \mathbb{E}_{P_n}[f] \leq \frac{K_0 + 1}{K_0}\mathbb{E}_P[f] + \frac{704K_0}{\bar{B}}r^* + \frac{x\left(11(b - a) + 26\bar{B}K_0\right)}{n}.$$

**Proposition A.4.** Let $\mathcal{F}$ be a class of functions with ranges in $[0, b]$ for some positive constant $b$. Let $\psi$ be a sub-root function such that for all $r \geq 0$, $\Re(\{f \in \mathcal{F} \colon \mathbb{E}_P[f(\mathbf{x})] \leq r\}) \leq \psi(r)$, and let $r^*$ be the fixed point of $\psi$. Then for any $K_0 > 1$, with probability $1 - \exp(-x)$, every $f \in \mathcal{F}$ satisfies

$$\mathbb{E}_P[f] \leq \frac{K_0}{K_0 - 1}\mathbb{E}_{P_n}[f] + \frac{704 K_0}{b}r^* + \frac{x\,(11(b-a) + 26bK_0)}{n}. \tag{24}$$

## B  PROOFS FOR THEOREM 5.1 AND COROLLARY 5.2

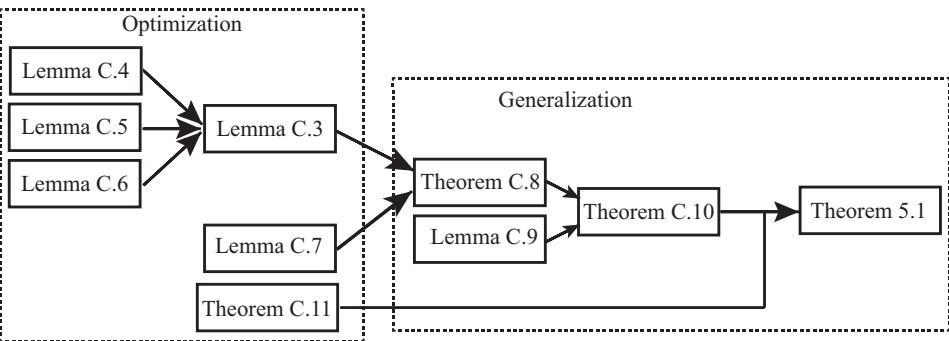

Figure 1: Roadmap of major results leading to the main result, Theorem 5.1. The uniform convergence results in Theorem 6.1 are used in all the optimization results and Theorem C.8.

**More discussion about the literature.** We herein provide more discussion about the results of this work and comparison to the existing relevant works with sharp rates for nonparametric regression. While this paper establishes sharp rate which is distribution-free in spherical covariate, such rate still depends on bounded input space ($\mathcal{X} = \mathbb{S}^{d-1}$) and the condition that the target function $f^* \in \mathcal{H}_K(\mu_0)$. Some other existing works consider target function $f^*$ not belonging to the RKHS ball centered at the origin with constant or low radius, such as (Haas et al., 2023; Bordelon et al., 2024). We also note that in this work, only the first layer of an over-parameterized two-layer neural network is trained, while the weights of the second layer are randomly initialized and then fixed in the training process. In existing works such as (Hu et al., 2021; Suh et al., 2022; Allen-Zhu et al., 2019a), all the layers of a deep neural networks with more than two-layers are trained by GD or its variants. However, this work shows that only training the first layer still leads to sharp rate for nonparametric regression, which supports the claim in (Bietti & Bach, 2021) that a shallow over-parameterized neural networks with ReLU activations exhibit the same approximation properties as its deeper counterpart.

**Proof of Theorem 5.1.** We use Theorem C.10 and Theorem C.11 to prove this theorem.

First of all, it follows by Theorem C.11 that with probability at least $1 - \exp\left(-\Theta(n\widehat{\varepsilon}_n^2)\right)$,

$$\mathbb{E}_{P_n}\left[(f_t - f^*)^2\right] \leq \frac{3}{\eta t}\left(\frac{\mu_0^2}{2e} + 6\right).$$

Plugging such bound for $\mathbb{E}_{P_n}\left[(f_t - f^*)^2\right]$ in (117) of Theorem C.10 leads to

$$\mathbb{E}_P\left[(f_t - f^*)^2\right] - \frac{6}{\eta t}\left(\frac{\mu_0^2}{2e} + 6\right) \leq c_0'(\varepsilon_n^2 + w). \tag{25}$$

Due to the definition of $\widehat{T}$ and $\widehat{\varepsilon}_n^2$, we have

$$\widehat{\varepsilon}_n^2 \leq \frac{1}{\eta\widehat{T}} \leq \frac{2}{\eta(\widehat{T} + 1)} \leq 2\widehat{\varepsilon}_n^2. \tag{26}$$

Lemma C.15 suggests that with probability at least $1 - 4\exp(-\Theta(n\varepsilon_n^2))$ over $\mathbf{S}$, $\widehat{\varepsilon}_n^2 \asymp \varepsilon_n^2$. Since $T \asymp \widehat{T}$, for any $t \in [c_t T, T]$, we have

$$\frac{1}{\eta t} \asymp \frac{1}{\eta T} \asymp \frac{1}{\eta \widehat{T}} \asymp \widehat{\varepsilon}_n^2 \asymp \varepsilon_n^2. \tag{27}$$

We have $\Pr[\mathcal{W}_0] \geq 1 - 2/n$. Let $w = \varepsilon_n^2$, we now verify that $w \in (0,1)$. Due to the definition of the fixed point, $w > 0$. Since $\sum_{i \geq 1} \lambda_i = \int_{\mathcal{X}} K(\mathbf{x}, \mathbf{x}) \mathrm{d}\mu(\mathbf{x}) = 1/2$, we have

$$0 < w = \frac{1}{n} \sum_{i \geq 1} \min\left\{\lambda_i, \varepsilon_n^2\right\} \leq \frac{1}{n} \sum_{i \geq 1} \lambda_i \leq \frac{1}{2n} < 1.$$

(11) then follows from (25) with $w = \varepsilon_n^2$, (27) and the union bound. The condition on $m$ in (85) in Theorem C.10, together with $w = \varepsilon_n^2$ and (27) leads to the condition on $m$ in (10). Furthermore, $\widehat{T} \asymp \varepsilon_n^{-2}$ follows from (27) and $\eta = \Theta(1)$.

$\square$

**Proof of Corollary 5.2.** We apply Theorem 5.1 to prove this corollary.

It is well known, such as (Raskutti et al., 2014, Corollary 3), that $\varepsilon_n^2 \asymp n^{-\frac{2\alpha}{2\alpha+1}}$. It then can be verified by direct calculations that the condition on $m$, (10) in Theorem 5.1, is satisfied with the given condition (12). It then follows from (11) in Theorem 5.1 that $\mathbb{E}_P\left[(f_{\widehat{T}} - f^*)^2\right] \lesssim n^{-\frac{2\alpha}{2\alpha+1}}$.

$\square$

## C   DETAILED PROOFS

Because Theorem 5.1 is proved by Theorem C.10 and Theorem C.11, in this section, we establish and prove all the theoretical results which lead to Theorem C.10 and Theorem C.11, along with the proof of Theorem C.10 and Theorem C.11.

### C.1   BASIC DEFINITIONS

We introduce the following definitions for the proof of Theorem 5.2. We define

$$\mathbf{u}(t) \coloneqq \widehat{\mathbf{y}}(t) - \mathbf{y}. \tag{28}$$

Let $\tau \leq 1$ be a positive number, and $\varepsilon_0 \in (0,1)$ is an arbitrary positive constant. For $t \geq 0$ and $T \geq 1$ we define the following quantities (or recall their definitions if defined before),

$$c_{\mathbf{u}} = \mu_0 / \min\left\{2, \sqrt{2e\eta}\right\} + \sigma + \tau + 1,$$

$$R = \frac{\eta c_{\mathbf{u}} T}{\sqrt{m}}, \tag{29}$$

$$\mathcal{V}_t \coloneqq \left\{\mathbf{v} \in \mathbb{R}^n \colon \mathbf{v} = -\left(\mathbf{I}_n - \eta \mathbf{K}_n\right)^t f^*(\mathbf{S})\right\}, \tag{30}$$

$$\mathcal{E}_{t,\tau} \coloneqq \left\{\mathbf{e} \colon \mathbf{e} = \overrightarrow{\mathbf{e}}_1 + \overrightarrow{\mathbf{e}}_2 \in \mathbb{R}^n, \overrightarrow{\mathbf{e}}_1 = -\left(\mathbf{I}_n - \eta \mathbf{K}_n\right)^t \mathbf{w}, \left\|\overrightarrow{\mathbf{e}}_2\right\|_2 \leq \sqrt{n}\tau\right\}. \tag{31}$$

We define the set of neural network weights and the set of functions represented by the neural network during training as follows.

$$\mathcal{W}(\mathbf{S}, \mathbf{W}(0), T) \coloneqq \left\{\mathbf{W} \colon \exists t \in [T] \text{ s.t. } \mathrm{vec}\left(\mathbf{W}\right) = \mathrm{vec}\left(\mathbf{W}(0)\right) - \sum_{t'=0}^{t-1} \frac{\eta}{n} \mathbf{Z}_{\mathbf{S}}(t') \mathbf{u}(t'),$$

$$\mathbf{u}(t') \in \mathbb{R}^n, \mathbf{u}(t') = \mathbf{v}(t') + \mathbf{e}(t'), \mathbf{v}(t') \in \mathcal{V}_{t'}, \mathbf{e}(t') \in \mathcal{E}_{t',\tau}, \text{ for all } t' \in [0, t-1] \Big\}. \qquad (32)$$

$\mathcal{W}(\mathbf{S}, \mathbf{W}(0), T)$ is the set of weights of neural networks trained by GD on the training data $\mathbf{S}$ and random initialization $\mathbf{W}(0)$ with the preconditioner $\mathbf{M}$ generated by $\mathbf{Q}$ and the steps of GD no greater than $T$. The set of functions represented by the two-layer NN with weights in $\mathcal{W}(\mathbf{S}, \mathbf{W}(0), T)$ is then defined as

$$\mathcal{F}_{\mathrm{NN}}(\mathbf{S}, \mathbf{W}(0), T) \coloneqq \{ f_t = f(\mathbf{W}(t), \cdot) : \exists\, t \in [T], \mathbf{W}(t) \in \mathcal{W}(\mathbf{S}, \mathbf{W}(0), T) \}. \qquad (33)$$

We define the function class $\mathcal{F}_{\mathrm{ext}}(w, T)$ for any $w > 0$ as

$$\mathcal{F}_{\mathrm{ext}}(w, T) \coloneqq \{ f \colon f = h + e, h \in \mathcal{H}_K(B_h), \|e\|_\infty \le w \}, \qquad (34)$$

where

$$B_h \coloneqq \mu_0 + 1 + \sqrt{2}. \qquad (35)$$

### C.2 Theorem C.10, Theorem C.11, and their proofs with related theoretical results

**Theorem C.10 (repeat).** Suppose $w \in (0, 1)$ and $m$ satisfy

$$m \gtrsim \max\left\{ \frac{(\eta T)^4 \left(\sqrt{d}+1\right)^4}{w^4}, (\eta T)^8 d^2 \right\},$$

and the neural network $f(\mathbf{W}(t), \cdot)$ is trained by GD in Algorithm 1 with the learning rate $\eta \in (0, 1/\widehat{\lambda}_1)$ on random initialization $\mathbf{W}(0)$, and $T \le \widehat{T}$. Then for every $t \in [T]$, with probability at least $1 - \exp\left(-\Theta(n)\right) - \exp\left(-\Theta(n\widehat{\varepsilon}_n^2)\right) - \exp\left(-n\varepsilon_n^2\right) - 2/n$ over the random noise $\mathbf{w}$, the random training features $\mathbf{S}$ and the random initialization $\mathbf{W}(0)$,

$$\mathbb{E}_P\left[(f_t - f^*)^2\right] - 2\mathbb{E}_{P_n}\left[(f_t - f^*)^2\right]$$

$$\le c_0 \min_{0 \le Q \le n} \left( \frac{B_0 Q}{n} + w\left(\sqrt{\frac{Q}{n}} + 1\right) + B_h \left( \frac{\sum\limits_{q=Q+1}^{\infty} \lambda_q}{n} \right)^{1/2} \right)^2,$$

Furthermore, with probability at least $1 - \exp\left(-\Theta(n)\right) - \exp\left(-\Theta(n\widehat{\varepsilon}_n^2)\right) - \exp\left(-n\varepsilon_n^2\right) - 2/n$ over the random noise $\mathbf{w}$, the random training features $\mathbf{S}$ and the random initialization $\mathbf{W}(0)$,

$$\mathbb{E}_P\left[(f_t - f^*)^2\right] - 2\mathbb{E}_{P_n}\left[(f_t - f^*)^2\right] \le c_0'(\varepsilon_n^2 + w). \qquad (36)$$

Here $B_0, c_0, c_0'$ are absolute positive constants depending on $\mu_0$, and $c_0'$ also depends on $\sigma$.

**Theorem C.11 (repeat).** Suppose the neural network trained after the $t$-th step of gradient descent, $f_t = f(\mathbf{W}(t), \cdot)$, satisfies $\mathbf{u}(t) = f_t(\mathbf{S}) - \mathbf{y} = \mathbf{v}(t) + \mathbf{e}(t)$ with $\mathbf{v}(t) \in \mathcal{V}_t$ and $\mathbf{e}(t) \in \mathcal{E}_{t,\tau}$ and $T \le \widehat{T}$. If

$$\eta \in [1, 2), \quad \tau \le \frac{1}{\eta T},$$

then for every $t \in [T]$, with probability at least $1 - \exp\left(-\Theta(n\widehat{\varepsilon}_n^2)\right)$ over the random noise $\mathbf{w}$, we have

$$\mathbb{E}_{P_n}\left[(f_t - f^*)^2\right] \le \frac{3}{\eta t}\left(\frac{\mu_0^2}{2e} + 3\right).$$

We have the following two theorems regarding the uniform convergence of $\widehat{h}(\mathbf{W}(0), \cdot, \cdot)$ to $K(\cdot, \cdot)$ and the uniform convergence of $\widehat{v}_R(\mathbf{W}(0), \cdot)$ to $\frac{2R}{\sqrt{2\pi}\kappa}$, which lay the foundation of the main results of this paper. The proofs are deferred to Section C.4.

**Theorem C.1.** Let $\mathbf{W}(0) = \left\{ \vec{\mathbf{w}}_r(0) \right\}_{r=1}^m$, where each $\vec{\mathbf{w}}_r(0) \sim \mathcal{N}(\mathbf{0}, \kappa^2 \mathbf{I}_d)$ for $r \in [m]$. Then for any $\delta \in (0, 1)$, with probability at least $1 - \delta$ over $\mathbf{W}(0)$,

$$\sup_{\mathbf{x} \in \mathcal{X}, \mathbf{y} \in \mathcal{X}} \left| K(\mathbf{x}, \mathbf{y}) - \widehat{h}(\mathbf{W}(0), \mathbf{x}, \mathbf{y}) \right| \leq C_1(m, d, \delta), \tag{37}$$

where

$$C_1(m, d, \delta) := \frac{1}{\sqrt{m}} \left( 6(1 + B\sqrt{d}) + \sqrt{2 \log \frac{2(1 + 2m)^d}{\delta}} \right) + \frac{1}{m} \left( 3 + \frac{7 \log \frac{2(1+2m)^d}{\delta}}{3} \right), \tag{38}$$

and $B$ is an absolute positive constant in Lemma C.19. In addition, when $m \geq \max\{d, n, 4\}$, $m/\log m \geq d$, and $\delta \asymp 1/n$,

$$C_1(m, d, \delta) \lesssim \sqrt{\frac{d \log m}{m}} + \frac{d \log m}{m} \lesssim \sqrt{\frac{d \log m}{m}}. \tag{39}$$

**Theorem C.2.** Let $\mathbf{W}(0) = \left\{ \vec{\mathbf{w}}_r(0) \right\}_{r=1}^m$, where each $\vec{\mathbf{w}}_r(0) \sim \mathcal{N}(\mathbf{0}, \kappa^2 \mathbf{I}_d)$ for $r \in [m]$. Suppose $R \leq R_0$ for an arbitrary absolute positive constant $R_0 < \kappa$. $B$ is an absolute positive constant in Lemma C.19. Then for any $\delta \in (0, 1)$, with probability at least $1 - \delta$ over $\mathbf{W}(0)$,

$$\sup_{\mathbf{x} \in \mathcal{X}} \left| \widehat{v}_R(\mathbf{W}(0), \mathbf{x}) - \frac{2R}{\sqrt{2\pi}\kappa} \right| \leq C_2(m, R_0, \delta), \tag{40}$$

where

$$C_2(m, R_0, \delta) := 3 \left( (B\sqrt{d} + 1) \sqrt{m^{-\frac{1}{2}} + \frac{1}{m}} + \frac{\exp\left( -\frac{(\kappa^2 - R_0^2)^2}{4\kappa^4} m \right)}{\sqrt{m^{-\frac{1}{2}} + \frac{1}{m}}} \right)$$
$$+ \sqrt{\frac{2 \log \frac{2(1+2m)^d}{\delta}}{m}} + \frac{7 \log \frac{2(1+2m)^d}{\delta}}{3m}. \tag{41}$$

In addition, when $m \geq \max\{d, n, 4\}$, $m/\log m \geq d$, and $\delta \asymp 1/n$,

$$C_2(m, R_0, \delta) \lesssim \frac{\sqrt{d}}{m^{1/4}} + \sqrt{\frac{d \log m}{m}} + \frac{d \log m}{m} \lesssim \frac{\sqrt{d}}{m^{1/4}}, $$

**Lemma C.3.** Suppose

$$m \gtrsim (\eta T)^4 (\sqrt{d} + \sqrt{\tau})^4 / \tau^4. \tag{42}$$

and the neural network $f(\mathbf{W}(t), \cdot)$ trained by gradient decent with the learning rate $\eta \in (0, 1/\widehat{\lambda}_1)$ on random initialization $\mathbf{W}(0) \in \mathcal{W}_0$. Then with probability at least $1 - \exp(-\Theta(n))$ over the random noise $\mathbf{w}$, $\mathbf{W}(t) \in \mathcal{W}(\mathbf{S}, \mathbf{W}(0), T)$. Moreover, for all $t \in [0, T]$, $\mathbf{u}(t) = \mathbf{v}(t) + \mathbf{e}(t)$ where $\mathbf{u}(t) = \widehat{\mathbf{y}}(t) - \mathbf{y}$, $\mathbf{v}(t) \in \mathcal{V}_{K,t}$, $\mathbf{e}(t) \in \mathcal{E}_{K,t,\tau}$, and $\|\mathbf{u}(t)\|_2 \leq c_{K,\mathbf{u}} \sqrt{n}$.

*Proof.* First, when $m \gtrsim (\eta T)^4 (\sqrt{d} + \sqrt{\tau})^4 / \tau^4$ with a proper constant, it can be verified that $\mathbf{E}_{m,\eta,\tau} \leq \tau \sqrt{n}/T$ where $\mathbf{E}_{m,\eta,\tau}$ is defined by (52) of Lemma C.5. Also, Theorem C.1 and Theorem C.2 hold when (42) holds. We then use mathematical induction to prove the lemma. We will first prove that $\mathbf{u}(t) = \mathbf{v}(t) + \mathbf{e}(t)$ where $\mathbf{v}(t) \in \mathcal{V}_t$, $\mathbf{e}(t) \in \mathcal{E}_{t,\tau}$, and $\|\mathbf{u}(t)\|_2 \leq c_{\mathbf{u}} \sqrt{n}$ for for all $t \in [0, T]$.

for all $t \in [0, T]$, where $\sum_{t'=1}^t \cdot = 0$ for $t < 1$, and

When $t = 0$, we have

$$\mathbf{u}(0) = -\mathbf{y} = \mathbf{v}(0) + \mathbf{e}(0), \tag{43}$$

where $\mathbf{v}(0) := -f^*(\mathbf{S}) = -(\mathbf{I} - \eta\mathbf{K}_n)^0 f^*(\mathbf{S})$, $\mathbf{e}(0) = -\mathbf{w} = \vec{\mathbf{e}}_1(0) + \vec{\mathbf{e}}_2(0)$ with $\vec{\mathbf{e}}_1(0) = -(\mathbf{I} - \eta\mathbf{K}_n)^0 \mathbf{w}$ and $\vec{\mathbf{e}}_2(0) = \mathbf{0}$. Therefore, $\mathbf{v}(0) \in \mathcal{V}_0$ and $\mathbf{e}(0) \in \mathcal{E}_{0,\tau}$. Also, it follows from the

proof of Lemma C.4 that $\|\mathbf{u}(0)\|_2 \leq c_{\mathbf{u}}$ with probability at least $1 - \exp(-\Theta(n))$ over the random noise $\mathbf{w}$.

Suppose that for all $t_1 \in [0, t]$ with $t \in [0, T-1]$, $\mathbf{u}(t_1) = \mathbf{v}(t_1) + \mathbf{e}(t_1)$ where $\mathbf{v}(t_1) \in \mathcal{V}_{t_1}$, and $\mathbf{e}(t_1) = \vec{\mathbf{e}}_1(t_1) + \vec{\mathbf{e}}_2(t_1)$ with $\mathbf{v}(t_1) \in \mathcal{V}_{t_1}$ and $\mathbf{e}(t_1) \in \mathcal{E}_{t_1, \tau}$, and $\|\mathbf{u}(t_1)\|_2 \leq c_{\mathbf{u}}\sqrt{n}$ for all $t_1 \in [0, t]$

Then it follows from Lemma C.5 that the recursion $\mathbf{u}(t'+1) = (\mathbf{I} - \eta\mathbf{K}_n)\mathbf{u}(t') + \mathbf{E}(t'+1)$ holds for all $t' \in [0, t]$. As a result, we have

$$
\begin{aligned}
\mathbf{u}(t+1) &= (\mathbf{I} - \eta\mathbf{K}_n)\mathbf{u}(t) + \mathbf{E}(t+1) \\
&= -(\mathbf{I} - \eta\mathbf{K}_n)^{t+1} f^*(\mathbf{S}) - (\mathbf{I} - \eta\mathbf{K}_n)^t \mathbf{w} \\
&\quad + \sum_{t'=1}^{t} (\mathbf{I} - \eta\mathbf{K}_n)^{t-t'} \mathbf{E}(t') \\
&= \mathbf{v}(t+1) + \mathbf{e}(t+1),
\end{aligned}
\tag{44}
$$

where $\mathbf{v}(t+1)$ and $\mathbf{e}(t+1)$ are defined as

$$
\mathbf{v}(t+1) := -(\mathbf{I} - \eta\mathbf{K}_n)^{t+1} f^*(\mathbf{S}) \in \mathcal{V}_{t+1},
\tag{45}
$$

$$
\mathbf{e}(t+1) := \underbrace{-(\mathbf{I} - \eta\mathbf{K}_n)^{t+1} \mathbf{w}}_{\vec{\mathbf{e}}_1(t+1)} + \underbrace{\sum_{t'=1}^{t+1} (\mathbf{I} - \eta\mathbf{K}_n)^{t+1-t'} \mathbf{E}(t')}_{\vec{\mathbf{e}}_2(t+1)}.
\tag{46}
$$

We now prove the upper bound for $\vec{\mathbf{e}}_2(t+1)$. With $\eta \in (0, 1/\widehat{\lambda}_1)$, we have $\|\mathbf{I} - \eta\mathbf{K}_n\|_2 \in (0, 1)$. It follows that

$$
\begin{aligned}
&\left\|\vec{\mathbf{e}}_2(t+1)\right\|_2 \\
&\leq \sum_{t'=1}^{t+1} \|\mathbf{I} - \eta\mathbf{K}_n\|_2^{t+1-t'} \|\mathbf{E}(t')\|_2 \\
&\leq \tau\sqrt{n},
\end{aligned}
\tag{47}
$$

where the last inequality follows from the fact that $\|\mathbf{E}(t)\|_2 \leq \mathbf{E}_{m,\eta,\tau} \leq \tau\sqrt{n}/T$ for all $t \in [T]$ and the induction hypothesis. It follows that $\mathbf{e}(t+1) \in \mathcal{E}_{t+1,\tau}$. Also, it follows from Lemma C.4 that

$$
\begin{aligned}
\|\mathbf{u}(t+1)\|_2 &\leq \|\mathbf{v}(t+1)\|_2 + \left\|\vec{\mathbf{e}}_1(t+1)\right\|_2 + \left\|\vec{\mathbf{e}}_2(t+1)\right\|_2 \\
&\leq \left(\frac{\mu_0}{\sqrt{2e\eta}} + \sigma + \tau + 1\right)\sqrt{n} = c_{\mathbf{u}}\sqrt{n},
\end{aligned}
$$

This fact completes the induction step, which also completes the proof. $\qquad\square$

**Lemma C.4.** Let $t \in [T]$, $\mathbf{v} = -(\mathbf{I} - \eta\mathbf{K}_n)^t f^*(\mathbf{S})$, $\mathbf{e} = -(\mathbf{I} - \eta\mathbf{K}_n)^{t+1}\mathbf{w}$, and $\eta \in (0, 1/\widehat{\lambda}_1)$. Then with probability at least $1 - \exp(-\Theta(n))$ over the random noise $\mathbf{w}$,

$$
\|\mathbf{v}\|_2 + \|\mathbf{e}\|_2 \leq \left(\frac{\mu_0}{\sqrt{2e\eta}} + \sigma + 1\right)\sqrt{n}
\tag{48}
$$

*Proof.* When $\mathbf{v} \in \mathcal{V}_t$ for $t \geq 1$, we have $\mathbf{v} = -(\mathbf{I} - \eta\mathbf{K}_n)^t f^*(\mathbf{S})$, and

$$
\begin{aligned}
\|\mathbf{v}(t)\|_2^2 &= \sum_{i=1}^{n} \left(1 - \eta\widehat{\lambda}_i\right)^{2t} \left[\mathbf{U}^\top f^*(\mathbf{S})\right]_i^2 \\
&\overset{\textcircled{1}}{\leq} \sum_{i=1}^{n} \frac{1}{2e\eta\widehat{\lambda}_i t} \left[\mathbf{U}^\top f^*(\mathbf{S})\right]_i^2
\end{aligned}
$$

$$\overset{②}{\leq} \frac{n\mu_0^2}{2e\eta t}. \tag{49}$$

Here ① follows Lemma C.13, ② follows by Lemma C.12.

Moreover, it follows from the concentration inequality about quadratic forms of sub-Gaussian random variables in (Wright, 1973) that

$$\Pr\left[\|\mathbf{w}\|_2^2 - \mathbb{E}\left[\|\mathbf{w}\|_2^2\right] > n\right] \leq \exp\left(-\Theta(n)\right), \tag{50}$$

and $\mathbb{E}\left[\|\mathbf{w}\|_2\right] \leq \sqrt{\mathbb{E}\left[\|\mathbf{w}\|_2^2\right]} = \sqrt{n}\sigma$. Therefore, $\Pr\left[\|\mathbf{w}\|_2 - \sqrt{n}\sigma > \sqrt{n}\right] \leq \exp\left(-\Theta(n)\right)$.

As a result, we have

$$\|\mathbf{v}\|_2 + \|\mathbf{e}\|_2 \leq \sqrt{\frac{n\mu_0^2}{2e\eta}} + \|\mathbf{w}\|_2 \leq \left(\frac{\mu_0}{\sqrt{2e\eta}} + \sigma + 1\right)\sqrt{n}.$$

$\square$

**Lemma C.5.** Let $0 < \eta < 1$, $0 \leq t \leq T - 1$ for $T \geq 1$, and suppose that $\|\widehat{\mathbf{y}}(t') - \mathbf{y}\|_2 \leq c_{\mathbf{u}}\sqrt{n}$ holds for all $0 \leq t' \leq t$. Then

$$\widehat{\mathbf{y}}(t+1) - \mathbf{y} = (\mathbf{I} - \eta\mathbf{K}_n)(\widehat{\mathbf{y}}(t) - \mathbf{y}) + \mathbf{E}(t+1), \tag{51}$$

where $\|\mathbf{E}(t+1)\|_2 \leq \mathbf{E}_{m,\eta,\tau}$, and $\mathbf{E}_{m,\eta,\tau}$ is defined by

$$\mathbf{E}_{m,\eta,\tau} := \eta c_{\mathbf{u}}\sqrt{n}\left(3\left(\frac{2R}{\sqrt{2\pi\kappa}} + C_2(m/2, R_0, \delta)\right) + C_1(m/2, d, \delta)\right)$$

$$\lesssim \eta\sqrt{n}\left(\frac{\sqrt{d}}{m^{1/4}} + \frac{\eta T}{\sqrt{m}}\right). \tag{52}$$

*Proof.* Because $\|\widehat{\mathbf{y}}(t') - \mathbf{y}\|_2 \leq \sqrt{n}c_{\mathbf{u}}$ holds for all $t' \in [0, t]$, by Lemma C.6, we have

$$\left\|\vec{\mathbf{w}}_r(t') - \vec{\mathbf{w}}_r(0)\right\|_2 \leq R, \quad \forall 0 \leq t' \leq t + 1. \tag{53}$$

Define two sets of indices

$$E_{i,R} := \left\{r \in [m]: \left|\mathbf{w}_r(0)^\top \vec{\mathbf{x}}_i\right| > R\right\}, \quad \bar{E}_{i,R} := [m] \setminus E_{i,R}.$$

We have

$$\widehat{\mathbf{y}}_i(t+1) - \widehat{\mathbf{y}}_i(t) = \frac{1}{\sqrt{m}}\sum_{r=1}^m a_r\left(\sigma\left(\vec{\mathbf{w}}_{\mathbf{S},r}^\top(t+1)\vec{\mathbf{x}}_i\right) - \sigma\left(\vec{\mathbf{w}}_{\mathbf{S},r}^\top(t)\vec{\mathbf{x}}_i\right)\right)$$

$$= \underbrace{\frac{1}{\sqrt{m}}\sum_{r \in E_{i,R}} a_r\left(\sigma\left(\vec{\mathbf{w}}_{\mathbf{S},r}^\top(t+1)\vec{\mathbf{x}}_i\right) - \sigma\left(\vec{\mathbf{w}}_{\mathbf{S},r}^\top(t)\vec{\mathbf{x}}_i\right)\right)}_{:=\mathbf{D}_i^{(1)}}$$

$$+ \underbrace{\frac{1}{\sqrt{m}}\sum_{r \in \bar{E}_{i,R}} a_r\left(\sigma\left(\vec{\mathbf{w}}_{\mathbf{S},r}^\top(t+1)\vec{\mathbf{x}}_i\right) - \sigma\left(\vec{\mathbf{w}}_{\mathbf{S},r}^\top(t)\vec{\mathbf{x}}_i\right)\right)}_{:=\mathbf{E}_i^{(1)}}$$

$$= \mathbf{D}_i^{(1)} + \mathbf{E}_i^{(1)}, \tag{54}$$

and $\mathbf{D}^{(1)}, \mathbf{E}^{(1)} \in \mathbb{R}^n$ is a vector with their $i$-th element being $\mathbf{D}_i^{(1)}$ and $\mathbf{E}_i^{(1)}$ defined on the RHS of (54). Now we derive the upper bound for $\mathbf{E}_i^{(1)}$. For all $i \in [n]$ we have

$$\left|\mathbf{E}_i^{(1)}\right| = \left|\frac{1}{\sqrt{m}}\sum_{r \in \bar{E}_{i,R}} a_r\left(\sigma\left(\vec{\mathbf{w}}_{\mathbf{S},r}(t+1)^\top\vec{\mathbf{x}}_i\right) - \sigma\left(\vec{\mathbf{w}}_{\mathbf{S},r}(t)^\top\vec{\mathbf{x}}_i\right)\right)\right|$$

$$\leq \frac{1}{\sqrt{m}} \sum_{r \in \bar{E}_{i,R}} \left| \vec{\mathbf{w}}_{\mathbf{S},r}(t+1)^\top \vec{\mathbf{x}}_i - \vec{\mathbf{w}}_{\mathbf{S},r}(t)^\top \vec{\mathbf{x}}_i \right|$$

$$\leq \frac{1}{\sqrt{m}} \sum_{r \in \bar{E}_{i,R}} \left\| \vec{\mathbf{w}}_{\mathbf{S},r}(t+1) - \vec{\mathbf{w}}_{\mathbf{S},r}(t) \right\|_2$$

$$\overset{\textcircled{1}}{=} \frac{1}{\sqrt{m}} \sum_{r \in \bar{E}_{i,R}} \left\| \frac{\eta}{n} [\mathbf{Z}_{\mathbf{S}}(t)]_{[(r-1)d+1:rd]} (\widehat{\mathbf{y}}(t) - \mathbf{y}) \right\|_2$$

$$\overset{\textcircled{2}}{\leq} \frac{c_{\mathbf{u}}}{\sqrt{m}} \sum_{r \in \bar{E}_{i,R}} \frac{\eta}{\sqrt{m}}$$

$$\leq \eta c_{\mathbf{u}} \cdot \frac{\left| \bar{E}_{i,R} \right|}{m}. \tag{55}$$

Here ①, ② follow from (72) and (73) in the proof of Lemma C.6.

Let $m$ be sufficiently large such that $R \leq R_0$ for the absolute positive constant $R_0 < \kappa$ specified in Theorem 6.1. Then it follows from Theorem C.2 that for any $\delta \in (0,1)$, with probability at least $1 - \delta$ over $\mathbf{W}(0)$,

$$\sup_{\mathbf{x} \in \mathcal{X}} \left| \widehat{v}_R(\mathbf{W}(0), \mathbf{x}) - \frac{2R}{\sqrt{2\pi\kappa}} \right| \leq C_2(m/2, R_0, \delta), \tag{56}$$

where $\widehat{v}_R(\mathbf{W}(0), \mathbf{x}) = \frac{1}{m} \sum_{r=1}^{m} \mathbb{1}_{\left\{ \left| \vec{\mathbf{w}}_r(0)^\top \mathbf{x} \right| \leq R \right\}}$, so that $\widehat{v}_R(\mathbf{W}(0), \vec{\mathbf{x}}_i) = \left| \bar{E}_{i,R} \right| / m$. It follows from (55), (56) and the induction hypothesis that

$$\left| \mathbf{E}_i^{(1)} \right| \leq \eta c_{\mathbf{u}} \left( \frac{2R}{\sqrt{2\pi\kappa}} + C_2(m/2, R_0, \delta) \right). \tag{57}$$

It follows from (57) that $\left\| \mathbf{E}^{(1)} \right\|_2$ can be bounded by

$$\left\| \mathbf{E}^{(1)} \right\|_2 \leq \eta c_{\mathbf{u}} \sqrt{n} \left( \frac{2R}{\sqrt{2\pi\kappa}} + C_2(m/2, R_0, \delta) \right). \tag{58}$$

$\mathbf{D}_i^{(1)}$ on the RHS of (54) is expressed by

$$\mathbf{D}_i^{(1)} = \frac{1}{\sqrt{m}} \sum_{r \in E_{i,R}} a_r \left( \sigma \left( \vec{\mathbf{w}}_{\mathbf{S},r}^\top(t+1) \vec{\mathbf{x}}_i \right) - \sigma \left( \vec{\mathbf{w}}_{\mathbf{S},r}^\top(t) \vec{\mathbf{x}}_i \right) \right)$$

$$= \frac{1}{\sqrt{m}} \sum_{r \in E_{i,R}} a_r \mathbb{1}_{\left\{ \vec{\mathbf{w}}_{\mathbf{S},r}(t)^\top \vec{\mathbf{x}}_i \geq 0 \right\}} \left( \vec{\mathbf{w}}_{\mathbf{S},r}(t+1) - \vec{\mathbf{w}}_{\mathbf{S},r}(t) \right)^\top \vec{\mathbf{x}}_i$$

$$= \frac{1}{\sqrt{m}} \sum_{r=1}^{m} a_r \mathbb{1}_{\left\{ \vec{\mathbf{w}}_{\mathbf{S},r}(t)^\top \vec{\mathbf{x}}_i \geq 0 \right\}} \left( -\frac{\eta}{n} [\mathbf{Z}_{\mathbf{S}}(t)]_{[(r-1)d:rd]} (\widehat{\mathbf{y}}(t) - \mathbf{y}) \right)^\top \vec{\mathbf{x}}_i$$

$$+ \frac{1}{\sqrt{m}} \sum_{r \in \bar{E}_{i,R}} a_r \mathbb{1}_{\left\{ \vec{\mathbf{w}}_{\mathbf{S},r}(t)^\top \vec{\mathbf{x}}_i \geq 0 \right\}} \left( \frac{\eta}{n} [\mathbf{Z}_{\mathbf{S}}(t)]_{[(r-1)d:rd]} (\widehat{\mathbf{y}}(t) - \mathbf{y}) \right)^\top \vec{\mathbf{x}}_i$$

$$= \underbrace{-\frac{\eta}{n} [\mathbf{H}(t)]_i (\widehat{\mathbf{y}}(t) - \mathbf{y})}_{:=\mathbf{D}_i^{(2)}}$$

$$+ \underbrace{\frac{1}{\sqrt{m}} \sum_{r \in \bar{E}_{i,R}} a_r \mathbb{1}_{\left\{ \vec{\mathbf{w}}_{\mathbf{S},r}(t)^\top \vec{\mathbf{x}}_i \geq 0 \right\}} \left( \frac{\eta}{n} [\mathbf{Z}_{\mathbf{S}}(t)]_{[(r-1)d:rd]} (\widehat{\mathbf{y}}(t) - \mathbf{y}) \right)^\top \vec{\mathbf{x}}_i}_{:=\mathbf{E}_i^{(2)}}$$

$$= \mathbf{D}_i^{(2)} + \mathbf{E}_i^{(2)}, \tag{59}$$

where $\mathbf{H}(t) \in \mathbb{R}^{n \times n}$ is a matrix specified by

$$\mathbf{H}_{pq}(t) = \frac{\vec{\mathbf{x}}_p^\top \vec{\mathbf{x}}_q}{m} \sum_{r=1}^m \mathbb{1}_{\left\{\vec{\mathbf{w}}_{\mathbf{S},r}(t)^\top \vec{\mathbf{x}}_p \geq 0\right\}} \mathbb{1}_{\left\{\vec{\mathbf{w}}_r(0)^\top \vec{\mathbf{x}}_q \geq 0\right\}}, \quad \forall p \in [n], q \in [n].$$

Let $\mathbf{D}^{(2)}, \mathbf{E}^{(2)} \in \mathbb{R}^n$ be a vector with their $i$-the element being $\mathbf{D}_i^{(2)}$ and $\mathbf{E}_i^{(2)}$ defined on the RHS of (59). $\mathbf{E}^{(2)}$ can be expressed by $\mathbf{E}^{(2)} = \frac{\eta}{n} \tilde{\mathbf{E}}^{(2)} (\widehat{\mathbf{y}}(t) - \mathbf{y})$ with $\tilde{\mathbf{E}}^{(2)} \in \mathbb{R}^{n \times n}$ and

$$\tilde{\mathbf{E}}_{pq}^{(2)} = \frac{1}{m} \sum_{r \in \bar{E}_{i,R}} \mathbb{1}_{\left\{\vec{\mathbf{w}}_{\mathbf{S},r}(t)^\top \vec{\mathbf{x}}_p \geq 0\right\}} \mathbb{1}_{\left\{\vec{\mathbf{w}}_r(0)^\top \vec{\mathbf{q}}_q \geq 0\right\}} \vec{\mathbf{x}}_q^\top \vec{\mathbf{x}}_p \leq \frac{1}{m} \sum_{r \in \bar{E}_{i,R}} 1 = \frac{|\bar{E}_{i,R}|}{m}$$

for all $p \in [n], q \in [n]$. The spectral norm of $\tilde{\mathbf{E}}^{(2)}$ is bounded by

$$\left\|\tilde{\mathbf{E}}^{(2)}\right\|_2 \leq \left\|\tilde{\mathbf{E}}^{(2)}\right\|_{\mathrm{F}} \leq n\frac{|\bar{E}_{i,R}|}{m} \overset{\textcircled{1}}{\leq} n\left(\frac{2R}{\sqrt{2\pi}\kappa} + C_2(m/2, R_0, \delta)\right), \tag{60}$$

where $\textcircled{1}$ follows from (56). Also, $\|\mathbf{H}(t)\|_2 \leq \|\mathbf{H}(t)\|_{\mathrm{F}} \leq \sqrt{nN}$ for all $t \geq 0$. It follows from (60) that $\left\|\mathbf{E}^{(2)}\right\|_2$ can be bounded by

$$\left\|\mathbf{E}^{(2)}\right\|_2 \leq \frac{\eta}{n}\left\|\tilde{\mathbf{E}}^{(2)}\right\|_2 \|\mathbf{y}(t) - \mathbf{y}\|_2$$

$$\leq \eta c_{\mathbf{u}} \sqrt{n}\left(\frac{2R}{\sqrt{2\pi}\kappa} + C_2(m/2, R_0, \delta)\right). \tag{61}$$

$\mathbf{D}_i^{(2)}$ on the RHS of (59) is expressed by

$$\mathbf{D}^{(2)} = -\frac{\eta}{n}\mathbf{H}(t)\left(\widehat{\mathbf{y}}(t) - \mathbf{y}\right)$$

$$= \underbrace{-\frac{\eta}{n}\mathbf{K}\left(\widehat{\mathbf{y}}(t) - \mathbf{y}\right)}_{:=\mathbf{D}^{(3)}}$$

$$+ \underbrace{\frac{\eta}{n}\left(\mathbf{K} - \mathbf{H}(0)\right)\left(\widehat{\mathbf{y}}(t) - \mathbf{y}\right)}_{:=\mathbf{E}^{(3)}}$$

$$+ \underbrace{\frac{\eta}{n}\left(\mathbf{H}(0) - \mathbf{H}(t)\right)\left(\widehat{\mathbf{y}}(t) - \mathbf{y}\right)}_{:=\mathbf{E}^{(4)}}$$

$$= \mathbf{D}^{(3)} + \mathbf{E}^{(3)} + \mathbf{E}^{(4)}. \tag{62}$$

On the RHS of (62), $\mathbf{D}^{(3)}, \mathbf{E}^{(3)}, \mathbf{E}^{(4)} \in \mathbb{R}^n$ are vectors which are analyzed as follows. $\left\|\tilde{\mathbf{E}}^{(3)}\right\|_2$ is bounded by

$$\|\mathbf{K} - \mathbf{H}(0)\|_2 \leq \|\mathbf{K} - \mathbf{H}(0)\|_F \leq nC_1(m/2, d, \delta), \tag{63}$$

where the last inequality holds with probability $1 - \delta$ over $\mathbf{W}(0)$ according to Theorem C.1.

In order to bound $\mathbf{E}^{(4)}$, we first estimate the upper bound for $|\mathbf{H}_{ij}(t) - \mathbf{H}_{ij}(0)|$ for all $i, j \in [n]$. We note that

$$\mathbb{1}_{\left\{\mathbb{1}_{\left\{\vec{\mathbf{w}}_{\mathbf{S},r}(t)^\top \vec{\mathbf{x}}_i\right\}} \neq \mathbb{1}_{\left\{\mathbf{w}_r(0)^\top \vec{\mathbf{x}}_i\right\}}\right\}} \leq \mathbb{1}_{\left\{\left|\mathbf{w}_r(0)^\top \vec{\mathbf{x}}_i\right| \leq R\right\}} + \mathbb{1}_{\left\{\left\|\mathbf{w}_{\mathbf{S},r}(t) - \vec{\mathbf{w}}_r(0)\right\|_2 > R\right\}}. \tag{64}$$

It follows from (64) that

$$|\mathbf{H}_{ij}(t) - \mathbf{H}_{ij}(0)|$$

$$= \left| \frac{\vec{\mathbf{x}}_i^\top \vec{\mathbf{x}}_j}{m} \sum_{r=1}^m \left( \mathbb{1}_{\left\{ \vec{\mathbf{w}}_{\mathbf{S},r}(t)^\top \vec{\mathbf{x}}_i \geq 0 \right\}} \mathbb{1}_{\left\{ \vec{\mathbf{w}}_r(0)^\top \vec{\mathbf{x}}_j \geq 0 \right\}} - \mathbb{1}_{\left\{ \mathbf{w}_r(0)^\top \vec{\mathbf{x}}_i \geq 0 \right\}} \mathbb{1}_{\left\{ \mathbf{w}_r(0)^\top \vec{\mathbf{x}}_j \geq 0 \right\}} \right) \right|$$

$$\leq \frac{1}{m} \sum_{r=1}^m \mathbb{1}_{\left\{ \mathbb{1}_{\left\{ \vec{\mathbf{w}}_{\mathbf{S},r}(t)^\top \vec{\mathbf{x}}_i \geq 0 \right\}} \neq \mathbb{1}_{\left\{ \vec{\mathbf{w}}_r(0)^\top \vec{\mathbf{x}}_i \geq 0 \right\}} \right\}}$$

$$\leq \frac{1}{m} \sum_{r=1}^m \left( \mathbb{1}_{\left\{ \left| \vec{\mathbf{w}}_r(0)^\top \vec{\mathbf{x}}_i \right| \leq R \right\}} + \mathbb{1}_{\left\{ \left\| \mathbf{w}_{\mathbf{S},r}(t) - \vec{\mathbf{w}}_r(0) \right\|_2 > R \right\}} \right)$$

$$\leq v_R(\mathbf{W}(0), \vec{\mathbf{x}}_i) \overset{\textcircled{1}}{\leq} \frac{2R}{\sqrt{2\pi}\kappa} + C_2(m/2, R_0, \delta), \tag{65}$$

where $\textcircled{1}$ follows from (56).

It follows from (63) and (65) that $\left\| \mathbf{E}^{(3)} \right\|_2, \left\| \mathbf{E}^{(4)} \right\|_2$ are bounded by

$$\left\| \mathbf{E}^{(3)} \right\|_2 \leq \frac{\eta}{n} \| \mathbf{K} - \mathbf{H}(0) \|_2 \| \widehat{\mathbf{y}}(t) - \mathbf{y} \|_2$$

$$\leq \frac{\eta}{n} \cdot n C_1(m/2, d, \delta) \cdot \| \mathbf{y}(t) - \mathbf{y} \|_2$$

$$\leq \eta c_{\mathbf{u}} \sqrt{n} C_1(m/2, d, \delta), \tag{66}$$

$$\left\| \mathbf{E}^{(4)} \right\|_2 \leq \frac{\eta}{n} \| \mathbf{H}(0) - \mathbf{H}(t) \|_2 \| \widehat{\mathbf{y}}(t) - \mathbf{y} \|_2$$

$$\leq \frac{\eta}{n} \cdot n \left( \frac{2R}{\sqrt{2\pi}\kappa} + C_2(m/2, R_0, \delta) \right) \cdot \| \mathbf{y}(t) - \mathbf{y} \|_2$$

$$\leq \eta c_{\mathbf{u}} \sqrt{n} \left( \frac{2R}{\sqrt{2\pi}\kappa} + C_2(m/2, R_0, \delta) \right). \tag{67}$$

It follows from (59) and (62) that

$$\mathbf{D}_i^{(1)} = \mathbf{D}_i^{(3)} + \mathbf{E}_i^{(2)} + \mathbf{E}_i^{(3)} + \mathbf{E}_i^{(4)}. \tag{68}$$

It then follows from (54) that

$$\widehat{\mathbf{y}}_i(t+1) - \widehat{\mathbf{y}}_i(t) = \mathbf{D}_i^{(1)} + \mathbf{E}_i^{(1)}$$

$$= \mathbf{D}_i^{(3)} + \underbrace{\mathbf{E}_i^{(1)} + \mathbf{E}_i^{(2)} + \mathbf{E}_i^{(3)} + \mathbf{E}_i^{(4)}}_{:=\mathbf{E}_i}$$

$$= -\frac{\eta}{n} \mathbf{K} \left( \widehat{\mathbf{y}}(t) - \mathbf{y} \right) + \mathbf{E}_i, \tag{69}$$

where $\mathbf{E} \in \mathbb{R}^n$ with its $i$-th element being $\mathbf{E}_i$, and $\mathbf{E} = \mathbf{E}^{(1)} + \mathbf{E}^{(2)} + \mathbf{E}^{(3)} + \mathbf{E}^{(4)}$. It then follows from (58), (61), (66), and (67) that

$$\| \mathbf{E} \|_2 \leq \eta c_{\mathbf{u}} \sqrt{n} \left( 3 \left( \frac{2R}{\sqrt{2\pi}\kappa} + C_2(m/2, R_0, \delta) \right) + C_1(m/2, d, \delta) \right). \tag{70}$$

Finally, (69) can be rewritten as

$$\widehat{\mathbf{y}}(t+1) - \mathbf{y} = \left( \mathbf{I} - \frac{\eta}{n} \mathbf{K} \right) \left( \widehat{\mathbf{y}}(t) - \mathbf{y} \right) + \mathbf{E}(t+1),$$

which proves (51) with the upper bound for $\| \mathbf{E} \|_2$ in (70).

$\square$

**Lemma C.6.** Suppose that $t \in [0, T-1]$ for $T \geq 1$, and $\| \widehat{\mathbf{y}}(t') - \mathbf{y} \|_2 \leq \sqrt{n} c_{\mathbf{u}}$ holds for all $0 \leq t' \leq t$. Then

$$\left\| \vec{\mathbf{w}}_{\mathbf{S},r}(t') - \vec{\mathbf{w}}_r(0) \right\|_2 \leq R, \quad \forall 0 \leq t' \leq t+1. \tag{71}$$

*Proof.* Let $[\mathbf{Z}_\mathbf{S}(t)]_{[(r-1)d:rd]}$ denotes the submatrix of $\mathbf{Z}_\mathbf{S}(t)$ formed by the the rows of $\mathbf{Z}_\mathbf{Q}(t)$ with row indices in $[(r-1)d:rd]$. By the GD update rule we have for $t \in [0, T-1]$ that

$$\vec{\mathbf{w}}_{\mathbf{S},r}(t+1) - \vec{\mathbf{w}}_{\mathbf{S},r}(t)$$
$$= -\frac{\eta}{n} [\mathbf{Z}_\mathbf{S}(t)]_{[(r-1)d:rd]} (\widehat{\mathbf{y}}(t) - \mathbf{y}), \tag{72}$$

We have $\left\| [\mathbf{Z}_\mathbf{S}(t)]_{[(r-1)d:rd]} \right\|_2 \leq \sqrt{n/m}$. It then follows from (72) that

$$\left\| \vec{\mathbf{w}}_{\mathbf{S},r}(t+1) - \vec{\mathbf{w}}_{\mathbf{S},r}(t) \right\|_2 \leq \frac{\eta}{n} \|\mathbf{Z}_\mathbf{S}(t)\|_2 \|\widehat{\mathbf{y}}(t) - \mathbf{y}\|_2 \leq \frac{\eta c_\mathbf{u}}{\sqrt{m}}. \tag{73}$$

Note that (71) trivially holds for $t' = 0$. For $t' \in [1, t+1]$, it follows from (73) that

$$\left\| \vec{\mathbf{w}}_{\mathbf{S},r}(t') - \vec{\mathbf{w}}_r(0) \right\|_2 \leq \sum_{t''=0}^{t'-1} \left\| \vec{\mathbf{w}}_{\mathbf{S},r}(t''+1) - \vec{\mathbf{w}}_{\mathbf{S},r}(t'') \right\|_2$$

$$\leq \frac{\eta}{\sqrt{m}} \sum_{t''=0}^{t'-1} c_\mathbf{u}$$

$$\leq \frac{\eta c_\mathbf{u} T}{\sqrt{m}} = R, \tag{74}$$

which completes the proof. $\qquad\square$

**Lemma C.7.** Let $h(\cdot) = \sum_{t'=0}^{t-1} h(\cdot, t')$ for $t \in [T], T \leq \widehat{T}$ where

$$h(\cdot, t') = v(\cdot, t') + \widehat{e}(\cdot, t'),$$

$$v(\cdot, t') = \frac{\eta}{n} \sum_{j=1}^{n} K(\vec{\mathbf{x}}_j, \mathbf{x}) \mathbf{v}_j(t'),$$

$$\widehat{e}(\cdot, t') = \frac{\eta}{n} \sum_{j=1}^{n} K(\vec{\mathbf{x}}_j, \mathbf{x}) \vec{\mathbf{e}}_j(t'),$$

where $\mathbf{v}(t') \in \mathcal{V}_{t'}$, $\mathbf{e}(t') \in \mathcal{E}_{t',\tau}$ for all $0 \leq t' \leq t-1$. Suppose that $\tau \lesssim 1/(\eta T)$, then with probability at least $1 - \exp\left(-\Theta(n\widehat{\varepsilon}_n^2)\right)$ over the random noise $\mathbf{w}$,

$$\|h\|_{\mathcal{H}_K} \leq B_h = \mu_0 + 1 + \sqrt{2}, \tag{75}$$

and $B_h$ is also defined in (35).

*Proof.* We have $\mathbf{y} = f^*(\mathbf{S}) + \mathbf{w}$, $\mathbf{v}(t) = -(\mathbf{I} - \eta\mathbf{K}_n)^t f^*(\mathbf{S})$, $\mathbf{e}(t) = \vec{\mathbf{e}}_1(t) + \vec{\mathbf{e}}_2(t)$ with $\vec{\mathbf{e}}_1(t) = -(\mathbf{I} - \eta\mathbf{K}_n)^t \mathbf{w}$, $\left\| \vec{\mathbf{e}}_2(t) \right\|_2 \lesssim \sqrt{n}\tau$. We define

$$\widehat{e}_1(\cdot, t) = \frac{\eta}{n} \sum_{j=1}^{n} K(\vec{\mathbf{x}}_j, \mathbf{x}) \left[ \vec{\mathbf{e}}_1(t') \right]_j, \quad \widehat{e}_2(\cdot, t) = \frac{\eta}{n} \sum_{j=1}^{n} K(\vec{\mathbf{x}}_j, \mathbf{x}) \left[ \vec{\mathbf{e}}_2(t') \right]_j,$$

Let $\boldsymbol{\Sigma}$ be the diagonal matrix containing eigenvalues of $\mathbf{K}_n$, we then have

$$\sum_{t'=0}^{t-1} v(\mathbf{x}, t') = \frac{\eta}{n} \sum_{j=1}^{n} \sum_{t'=0}^{t-1} \left[ (\mathbf{I} - \eta\mathbf{K}_n)^{t'} f^*(\mathbf{S}) \right]_j K(\vec{\mathbf{x}}_j, \mathbf{x})$$

$$= \frac{\eta}{n} \sum_{j=1}^{n} \sum_{t'=0}^{t-1} \left[ \mathbf{U} (\mathbf{I} - \eta\boldsymbol{\Sigma})^{t'} \mathbf{U}^\top f^*(\mathbf{S}) \right]_j K(\vec{\mathbf{x}}_j, \mathbf{x}). \tag{76}$$

It follows from (76) that

$$\left\| \sum_{t'=0}^{t-1} v(\cdot, t') \right\|_{\mathcal{H}_K}^2$$

$$
= \frac{\eta^2}{n^2} f^*(\mathbf{S})^\top \mathbf{U} \sum_{t'=0}^{t-1} (\mathbf{I} - \eta\boldsymbol{\Sigma})^{t'} \mathbf{U}^\top \mathbf{K} \mathbf{U} \sum_{t'=0}^{t-1} (\mathbf{I} - \eta\boldsymbol{\Sigma})^{t'} \mathbf{U}^\top f^*(\mathbf{S})
$$

$$
= \frac{1}{n} \left\| \eta \left(\mathbf{K}_n\right)^{1/2} \mathbf{U} \sum_{t'=0}^{t-1} (\mathbf{I} - \eta\boldsymbol{\Sigma})^{t'} \mathbf{U}^\top f^*(\mathbf{S}) \right\|_2^2
$$

$$
\leq \frac{1}{n} \sum_{i=1}^{n} \frac{\left(1 - \left(1 - \eta\widehat{\lambda}_i\right)^t\right)^2}{\widehat{\lambda}_i} \left[\mathbf{U}^\top f^*(\mathbf{S})\right]_i^2
$$

$$
\leq \mu_0^2, \tag{77}
$$

where the last inequality follows from Lemma C.12.

Similarly, we have

$$
\left\| \sum_{t'=0}^{t-1} \widehat{e}_1(\cdot, t') \right\|_{\mathcal{H}_K}^2 \leq \frac{1}{n} \sum_{i=1}^{n} \frac{\left(1 - \left(1 - \eta\widehat{\lambda}_i\right)^t\right)^2}{\widehat{\lambda}_i} \left[\mathbf{U}^\top \mathbf{w}\right]_i^2. \tag{78}
$$

It then follows from the argument in the proof of (Raskutti et al., 2014, Lemma 9) that the RHS of (78) is bounded with high probability. We define a diagonal matrix $\mathbf{R} \in \mathbb{R}^{n \times n}$ with $\mathbf{R}_{ii} = \left(1 - (1 - \eta\widehat{\lambda}_i)^t\right)^2 / \widehat{\lambda}_i$ for $i \in [n]$. Then the RHS of (78) is $1/n \cdot \mathrm{tr}\left(\mathbf{U}\mathbf{R}\mathbf{U}^\top \mathbf{w}\mathbf{w}^\top\right)$. It follows from (Wright, 1973) that

$$
\Pr\left[1/n \cdot \mathrm{tr}\left(\mathbf{U}\mathbf{R}\mathbf{U}^\top \mathbf{w}\mathbf{w}^\top\right) - \mathbb{E}\left[1/n \cdot \mathrm{tr}\left(\mathbf{U}\mathbf{R}\mathbf{U}^\top \mathbf{w}\mathbf{w}^\top\right)\right] \geq u\right]
$$

$$
\leq \exp\left(-c \min\left\{nu/\|\mathbf{R}\|_2, n^2 u^2/\|\mathbf{R}\|_F^2\right\}\right) \tag{79}
$$

for all $u > 0$, and $c$ is a positive constant. Recall that $\eta_t = \eta t$ for all $t \geq 0$, we have

$$
\mathbb{E}\left[1/n \cdot \mathrm{tr}\left(\mathbf{U}\mathbf{R}\mathbf{U}^\top \mathbf{w}\mathbf{w}^\top\right)\right] \leq \frac{\sigma^2}{n} \sum_{i=1}^{n} \frac{\left(1 - \left(1 - \eta\widehat{\lambda}_i\right)^t\right)^2}{\widehat{\lambda}_i}
$$

$$
\overset{①}{\leq} \frac{\sigma^2}{n} \sum_{i=1}^{n} \min\left\{\frac{1}{\widehat{\lambda}_i}, \eta_t^2\widehat{\lambda}_i\right\}
$$

$$
\leq \frac{\sigma^2 \eta_t}{n} \sum_{i=1}^{n} \min\left\{\frac{1}{\eta_t\widehat{\lambda}_i}, \eta_t\widehat{\lambda}_i\right\}
$$

$$
\overset{②}{\leq} \frac{\sigma^2 \eta_t}{n} \sum_{i=1}^{n} \min\left\{1, \eta_t\widehat{\lambda}_i\right\}
$$

$$
= \frac{\sigma^2 \eta_t^2}{n} \sum_{i=1}^{n} \min\left\{\eta_t^{-1}, \widehat{\lambda}_i\right\}
$$

$$
= \sigma^2 \eta_t^2 \widehat{R}_K^2\left(\sqrt{1/\eta_t}\right) \leq 1. \tag{80}
$$

Here ① follows from the fact that $(1 - \eta\widehat{\lambda}_i)^t \geq \max\left\{0, 1 - t\eta\widehat{\lambda}_i\right\}$, and ② follows from $\min\{a, b\} \leq \sqrt{ab}$ for any nonnegative numbers $a, b$. Because $t \leq T \leq \widehat{T}$, we have $\widehat{R}_K\left(\sqrt{1/\eta_t}\right) \leq 1/(\sigma\eta_t)$, so the last inequality holds.

Moreover, we have the upper bounds for $\|\mathbf{R}\|_2$ and $\|\mathbf{R}\|_F$ as follows. First, we have

$$
\|\mathbf{R}\|_2 \leq \max_{i \in [n]} \frac{\left(1 - \left(1 - \eta\widehat{\lambda}_i\right)^t\right)^2}{\widehat{\lambda}_i}
$$

$$\leq \min\left\{\frac{1}{\widehat{\lambda}_i}, \eta_t^2 \widehat{\lambda}_i\right\} \leq \eta_t. \tag{81}$$

We also have

$$
\begin{aligned}
\frac{1}{n}\|\mathbf{R}\|_{\mathrm{F}}^2 &= \frac{1}{n}\sum_{i=1}^n \frac{\left(1 - \left(1 - \eta\widehat{\lambda}_i\right)^t\right)^4}{(\widehat{\lambda}_i)^2} \\
&\leq \frac{\eta_t^3}{n}\sum_{i=1}^n \min\left\{\frac{1}{\eta_t^3\widehat{\lambda}_i^2}, \eta_t\widehat{\lambda}_i^2\right\} \\
&\overset{③}{\leq} \frac{\eta_t^3}{n}\sum_{i=1}^n \min\left\{\widehat{\lambda}_i, \frac{1}{\eta_t}\right\} = \eta_t^3 \widehat{R}_K^2(\sqrt{1/\eta_t}) \leq \frac{\eta_t}{\sigma^2},
\end{aligned}
\tag{82}
$$

where ③ follows from

$$\min\left\{\frac{1}{\eta_t^3\widehat{\lambda}_i^2}, \eta_t\widehat{\lambda}_i^2\right\} = \widehat{\lambda}_i \min\left\{\frac{1}{\eta_t^3\widehat{\lambda}_i^3}, \eta_t\widehat{\lambda}_i\right\} \leq \widehat{\lambda}_i.$$

Combining (78)- (82) with $u = 1$ in (79), we have

$$\Pr\left[1/n \cdot \mathrm{tr}\left(\mathbf{U}\mathbf{R}\mathbf{U}^\top\mathbf{w}\mathbf{w}^\top\right) - \mathbb{E}\left[1/n \cdot \mathrm{tr}\left(\mathbf{U}\mathbf{R}\mathbf{U}^\top\mathbf{w}\mathbf{w}^\top\right)\right] \geq 1\right] \leq \exp\left(-c\min\left\{n/\eta_t, n\sigma^2/\eta_t\right\}\right)$$
$$\leq \exp\left(-nc'/\eta_t\right) \leq \exp\left(-c'n\widehat{\varepsilon}_n^2\right)$$

where $c' = c\min\left\{1, \sigma^2\right\}$, and the last inequality is due to the fact that $1/\eta_t \geq \widehat{\varepsilon}_n^2$ since $t \leq T \leq \widehat{T}$. It follows that with probability at least $1 - \exp\left(-\Theta(n\widehat{\varepsilon}_n^2)\right)$, $\left\|\sum_{t'=0}^{t-1}\widehat{e}_1(\cdot, t')\right\|_{\mathcal{H}_K}^2 \leq 2$.

We now find the upper bound for $\left\|\sum_{t'=0}^{t-1}\widehat{e}_2(\cdot, t')\right\|_{\mathcal{H}_K}$. We have

$$
\begin{aligned}
\|\widehat{e}_2(\cdot, t')\|_{\mathcal{H}_K}^2 &\leq \frac{\eta^2}{n^2}\vec{\mathbf{e}}_2^\top(t')\mathbf{K}\vec{\mathbf{e}}_2(t') \\
&\leq \eta^2\widehat{\lambda}_1\tau^2,
\end{aligned}
$$

so that

$$
\begin{aligned}
\left\|\sum_{t'=0}^{t-1}\widehat{e}_2(\cdot, t')\right\|_{\mathcal{H}_K} &\leq \sum_{t'=0}^{t-1}\|\widehat{e}_2(\cdot, t')\|_{\mathcal{H}_K} \\
&\leq T\eta\sqrt{\widehat{\lambda}_1}\tau \leq 1,
\end{aligned}
\tag{83}
$$

if $\tau \lesssim 1/(\eta T)$.

Finally, we have

$$
\begin{aligned}
\|h\|_{\mathcal{H}_K} &\leq \left\|\sum_{t'=0}^{t-1}\widehat{v}(\cdot, t')\right\|_{\mathcal{H}_K} + \left\|\sum_{t'=0}^{t-1}\widehat{e}_1(\cdot, t')\right\|_{\mathcal{H}_K} + \left\|\sum_{t'=0}^{t-1}\widehat{e}_2(\cdot, t')\right\|_{\mathcal{H}_K} \\
&\leq \mu_0 + 1 + \sqrt{2} = B_h.
\end{aligned}
$$

$\qquad\square$

**Theorem C.8.** For every $t \in [T]$, let the neural network $f(\cdot) = f(\mathbf{W}(t), \cdot)$ be trained by gradient descent with the learning rate $\eta \in (0, 1/\widehat{\lambda}_1)$ on the random initialization $\mathbf{W}(0) \in \mathcal{W}_0$ with $T \leq \widehat{T}$. Then with probability at least $1 - \exp\left(-\Theta(n)\right) - \exp\left(-\Theta(n\widehat{\varepsilon}_n^2)\right)$ over the random noise $\mathbf{w}$, $f \in \mathcal{F}_{\mathrm{NN}}(\mathbf{S}, \mathbf{W}(0), T)$, and $f$ can be decomposed by

$$f = h + e \in \mathcal{F}_{\mathrm{ext}}(w, T), \tag{84}$$

where $h \in \mathcal{H}_K(B_h)$ with $B_h$ defined in (35), $e \in L^\infty$. When

$$m \gtrsim \max \left\{ \frac{(\eta T)^4 \left(\sqrt{d}+1\right)^4}{w^4}, (\eta T)^8 d^2 \right\}, \tag{85}$$

then

$$\|e\|_\infty \le w. \tag{86}$$

In addition,

$$\|f\|_\infty \le \frac{B_h}{\sqrt{2}} + w. \tag{87}$$

**Remark.** We consider the kernel regression problem with the training loss $L(\boldsymbol{\alpha}) = 1/2 \cdot \|\mathbf{K}_n \boldsymbol{\alpha} - \mathbf{y}\|_2^2$. Letting $\boldsymbol{\beta} = \mathbf{K}_n^{1/2} \boldsymbol{\alpha}$ and then performing GD on $\boldsymbol{\beta}$ with this training loss and the learning rate $\eta$, it can be verified that the kernel regressor right after the $t$-th step of GD is

$$\widehat{f}_t^{(\mathrm{NTK})} = \frac{\eta}{n} \sum_{t'=0}^{t-1} \sum_{i=1}^{n} K(\cdot, \vec{\mathbf{x}}_i) \boldsymbol{\alpha}_i^{(t')}, \tag{88}$$

where $\boldsymbol{\alpha}^{(t')} = (\mathbf{I}_n - \eta \mathbf{K}_n)^{t'} \mathbf{y}$. Following from the proof of Lemma C.6 and Theorem C.8, under the conditions of Theorem C.8 we have

$$h_t = \widehat{f}_t^{(\mathrm{NTK})} + \widehat{e}_2(\cdot, t),$$

where $\widehat{e}_2(\cdot, t) = \frac{\eta}{n} \sum_{t'=0}^{t-1} \sum_{j=1}^{n} K(\cdot, \vec{\mathbf{x}}_j) \left[\vec{\mathbf{e}}_2(t')\right]_j$ and $\vec{\mathbf{e}}_2(t')$ appears in the definition of $\mathcal{E}_{t,\tau}$ in (31). It is remarked that in our analysis, we approximate $f_t$ by $h_t \in \mathcal{H}_K(B_h)$ with a small approximation error $w$, and we do not need to approximate $f_t$ by the kernel regressor $\widehat{f}_t^{(\mathrm{NTK})}$ with a sufficiently small approximation error which is the common strategy used in existing works (Hu et al., 2021; Suh et al., 2022; Li et al., 2024). In fact, our analysis only requires $m$ is suitably large so that the $\mathcal{H}_K$-norm of $\widehat{e}_2(\cdot, t) = h_t - \widehat{f}_t^{(\mathrm{NTK})}$ is bounded by a positive constant rather than an infinitesimal number as $m \to \infty$, that is, $\|\widehat{e}_2(\cdot, t)\|_{\mathcal{H}_K} \le 1$, which is revealed by the proof of Lemma C.7.

*Proof.* It follows from Lemma C.3 and its proof that conditioned on an event with probability at least $1 - \exp\left(-\Theta(n)\right) - \Theta\left(nN/n^{c_d \varepsilon_0^2/8}\right) - (1 + 2N)^{2d} \exp(-n^{c_x})$, $f \in \mathcal{F}_{\mathrm{NN}}(\mathbf{S}, \mathbf{W}(0), T)$ with $\mathbf{W}(0) \in \mathcal{W}_0$. Moreover, $f(\cdot) = f(\mathbf{W}, \cdot)$ with $\mathbf{W} = \left\{\vec{\mathbf{w}}_r\right\}_{r=1}^{m} \in \mathcal{W}(\mathbf{S}, \mathbf{W}(0), T)$, and $\mathrm{vec}\left(\mathbf{W}\right) = \mathrm{vec}\left(\mathbf{W_S}\right) = \mathrm{vec}\left(\mathbf{W}(0)\right) - \sum_{t'=0}^{t-1} \eta/n \cdot \mathbf{Z_S}(t') \mathbf{u}(t')$ for some $t \in [T]$, where $\mathbf{u}(t') \in \mathbb{R}^n$, $\mathbf{u}(t') = \mathbf{v}(t') + \mathbf{e}(t')$ with $\mathbf{v}(t') \in \mathcal{V}_{t'}$ and $\mathbf{e}(t') \in \mathcal{E}_{t',\tau}$ for all $t' \in [0, t-1]$.

$\vec{\mathbf{w}}_r$ is expressed as

$$\vec{\mathbf{w}}_r = \vec{\mathbf{w}}_{\mathbf{S},r}(t) = \vec{\mathbf{w}}_r(0) - \sum_{t'=0}^{t-1} \frac{\eta}{n} \left[\mathbf{Z_S}(t')\right]_{[(r-1)d:rd]} \mathbf{u}(t'), \tag{89}$$

where the notation $\vec{\mathbf{w}}_{\mathbf{S},r}$ emphasizes that $\vec{\mathbf{w}}_r$ depends on the training data $\mathbf{S}$.

We define the event

$$E_r(R) := \left\{ \left|\vec{\mathbf{w}}_r(0)^\top \mathbf{x}\right| \le R \right\}, \quad r \in [m].$$

We now approximate $f(\mathbf{W}, \mathbf{x})$ by $g(\mathbf{x}) := \frac{1}{\sqrt{m}} \sum_{r=1}^{m} a_r \mathbb{1}_{\left\{\vec{\mathbf{w}}_r(0)^\top \mathbf{x} \ge 0\right\}} \vec{\mathbf{w}}_r^\top \mathbf{x}$. We have

$$\left|f(\mathbf{W}, \mathbf{x}) - g(\mathbf{x})\right|$$

$$= \frac{1}{\sqrt{m}} \left|\sum_{r=1}^{m} a_r \sigma\left(\vec{\mathbf{w}}_r^\top \mathbf{x}\right) - \sum_{r=1}^{m} a_r \mathbb{1}_{\left\{\vec{\mathbf{w}}_r(0)^\top \mathbf{x} \ge 0\right\}} \vec{\mathbf{w}}_r^\top \mathbf{x}\right|$$

$$\leq \frac{1}{\sqrt{m}} \sum_{r=1}^{m} \left| a_r \left( \mathbb{1}_{\{E_r(R)\}} + \mathbb{1}_{\{\bar{E}_r(R)\}} \right) \left( \sigma \left( \vec{\mathbf{w}}_r^\top \mathbf{x} \right) - \mathbb{1}_{\left\{ \vec{\mathbf{w}}_r(0)^\top \mathbf{x} \geq 0 \right\}} \vec{\mathbf{w}}_r^\top \mathbf{x} \right) \right|$$

$$= \frac{1}{\sqrt{m}} \sum_{r=1}^{m} \mathbb{1}_{\{E_r(R)\}} \left| \sigma \left( \vec{\mathbf{w}}_r^\top \mathbf{x} \right) - \mathbb{1}_{\left\{ \vec{\mathbf{w}}_r(0)^\top \mathbf{x} \geq 0 \right\}} \vec{\mathbf{w}}_r^\top \mathbf{x} \right|$$

$$= \frac{1}{\sqrt{m}} \sum_{r=1}^{m} \mathbb{1}_{\{E_r(R)\}} \left| \sigma \left( \vec{\mathbf{w}}_r^\top \mathbf{x} \right) - \sigma \left( \vec{\mathbf{w}}_r(0)^\top \mathbf{x} \right) - \mathbb{1}_{\left\{ \vec{\mathbf{w}}_r(0)^\top \mathbf{x} \geq 0 \right\}} (\vec{\mathbf{w}}_r - \vec{\mathbf{w}}_r(0))^\top \mathbf{x} \right|$$

$$\leq \frac{2R}{\sqrt{m}} \sum_{r=1}^{m} \mathbb{1}_{\{E_r(R)\}}. \tag{90}$$

Plugging $R = \frac{\eta c_{\mathbf{u}} T}{\sqrt{m}}$ in (90), we have

$$|f(\mathbf{W}, \mathbf{x}) - g(\mathbf{x})| \leq \frac{2R}{\sqrt{m}} \sum_{r=1}^{m} \mathbb{1}_{\{E_r(R)\}}$$

$$= 2\eta c_{\mathbf{u}} T \cdot \frac{1}{m} \sum_{r=1}^{m} \mathbb{1}_{\{E_r(R)\}}$$

$$= 2\eta c_{\mathbf{u}} T \cdot \widehat{v}_R(\mathbf{W}(0), \mathbf{x})$$

$$\leq 2\eta c_{\mathbf{u}} T \left( \frac{2R}{\sqrt{2\pi\kappa}} + C_2(m/2, R_0, \delta) \right). \tag{91}$$

Using (89), we can express $g(\mathbf{x})$ as

$$g(\mathbf{x}) = \frac{1}{\sqrt{m}} \sum_{r=1}^{m} a_r \mathbb{1}_{\left\{ \vec{\mathbf{w}}_r(0)^\top \mathbf{x} \geq 0 \right\}} \vec{\mathbf{w}}_r(0)^\top \mathbf{x}$$

$$- \sum_{t'=0}^{t-1} \frac{1}{\sqrt{m}} \sum_{r=1}^{m} \mathbb{1}_{\left\{ \vec{\mathbf{w}}_r(0)^\top \mathbf{x} \geq 0 \right\}} \left( \frac{\eta}{n} \left[ \mathbf{Z_S}(t') \right]_{[(r-1)d:rd]} \mathbf{u}(t') \right)^\top \mathbf{x}$$

$$\stackrel{①}{=} - \sum_{t'=0}^{t-1} \underbrace{\frac{\eta}{nm} \sum_{r=1}^{m} \mathbb{1}_{\left\{ \vec{\mathbf{w}}_r(0)^\top \mathbf{x} \geq 0 \right\}} \sum_{j=1}^{n} \mathbb{1}_{\left\{ \vec{\mathbf{w}}_r(t')^\top \vec{\mathbf{x}}_j \geq 0 \right\}} \mathbf{u}_j(t') \vec{\mathbf{x}}_j^\top \mathbf{x}}_{:=G_{t'}(\mathbf{x})}, \tag{92}$$

where ① follows from the fact that $\frac{1}{\sqrt{m}} \sum_{r=1}^{m} a_r \mathbb{1}_{\left\{ \vec{\mathbf{w}}_r(0)^\top \mathbf{x} \geq 0 \right\}} \vec{\mathbf{w}}_r(0)^\top \mathbf{x} = f(\mathbf{W}(0), \mathbf{x}) = 0$ due to the particular initialization of the two-layer NN. For each $G_{t'}$ in the RHS of (92), we have

$$G_{t'}(\mathbf{x}) \stackrel{②}{=} \frac{\eta}{nm} \sum_{r=1}^{m} \mathbb{1}_{\left\{ \vec{\mathbf{w}}_r(0)^\top \mathbf{x} \geq 0 \right\}} \sum_{j=1}^{n} d_{t',r,j} \mathbf{u}_j(t') \vec{\mathbf{x}}_j^\top \mathbf{x}$$

$$+ \frac{\eta}{nm} \sum_{r=1}^{m} \mathbb{1}_{\left\{ \vec{\mathbf{w}}_r(0)^\top \mathbf{x} \geq 0 \right\}} \sum_{j=1}^{n} \mathbb{1}_{\left\{ \vec{\mathbf{w}}_r(0)^\top \vec{\mathbf{x}}_j \geq 0 \right\}} \mathbf{u}_j(t') \vec{\mathbf{x}}_j^\top \mathbf{x}$$

$$\stackrel{③}{=} \frac{\eta}{n} \sum_{j=1}^{n} K(\mathbf{x}, \vec{\mathbf{x}}_j) \mathbf{u}_j(t') + \underbrace{\frac{\eta}{n} \sum_{j=1}^{n} q_j \mathbf{u}_j(t')}_{:=E_1(\mathbf{x})}$$

$$+ \underbrace{\frac{\eta}{nm} \sum_{r=1}^{m} \mathbb{1}_{\left\{ \vec{\mathbf{w}}_r(0)^\top \mathbf{x} \geq 0 \right\}} \sum_{j=1}^{n} d_{t',r,j} \mathbf{u}_j(t') \vec{\mathbf{x}}_j^\top \mathbf{x}}_{:=E_2(\mathbf{x})}. \tag{93}$$

where

$$d_{t',r,j} := \mathbb{1}_{\left\{ \vec{\mathbf{w}}_r(t')^\top \vec{\mathbf{x}}_j \geq 0 \right\}} - \mathbb{1}_{\left\{ \vec{\mathbf{w}}_r(0)^\top \vec{\mathbf{x}}_j \geq 0 \right\}}$$

in ②, and and

$$q_j := \widehat{h}(\mathbf{W}(0), \vec{\mathbf{x}}_j, \mathbf{x}) - K(\vec{\mathbf{x}}_j, \mathbf{x})$$

for all $j \in [n]$ in ③.

We now analyze each term on the RHS of (93). Let $h(\cdot, t'): \mathcal{X} \to \mathbb{R}$ be defined by

$$h(\mathbf{x}, t') := \frac{\eta}{n} \sum_{j=1}^{n} K(\mathbf{x}, \vec{\mathbf{x}}_j) \mathbf{u}_j(t'),$$

then $h(\cdot, t')$ is an element in the RKHS $\mathcal{H}_K$ for each $t' \in [0, t-1]$. We further define

$$h(\cdot) := \sum_{t'=0}^{t-1} h(\cdot, t'), \tag{94}$$

It follows from Theorem C.1 that, with probability $1 - \delta$ over $\mathbf{W}(0)$, $q_j \leq C_1(m/2, d, \delta)$ for all $j' \in [n]$ with $C_1(m/2, d, \delta)$ defined in (38). Moreover, $\|\mathbf{K}_{\mathbf{S},\mathbf{Q}}\|_2 \leq \sqrt{nN}$, $\mathbf{u}(t') \leq c_{\mathbf{u}}\sqrt{n}$ with high probability, so that we have

$$\|E_1\|_\infty = \left\| \frac{\eta}{n} \sum_{j=1}^{n} q_j \mathbf{u}_j(t') \right\|_\infty \leq \frac{\eta}{n} \|\mathbf{u}(t')\|_2 \sqrt{n} C_1(m/2, d, \delta)$$

$$\leq \eta c_{\mathbf{u}} C_1(m/2, d, \delta). \tag{95}$$

We now bound the last term on the RHS of (93). Define $\mathbf{X}' \in \mathbb{R}^{d \times n}$ with its $j$-column being $\mathbf{X}'_j = \frac{1}{m} \sum_{r=1}^{m} \mathbb{I}_{\left\{\vec{\mathbf{w}}_r(0)^\top \mathbf{x} \geq 0\right\}} d_{t',r,j} \vec{\mathbf{x}}_j$ for all $j \in [n]$, then $E_2(\mathbf{x}) = \frac{\eta}{n} \left(\mathbf{X}' \mathbf{u}(t')\right)^\top \mathbf{x}$.

We need to derive the upper bound for $\|\mathbf{X}'\|_2$. Because $\left\|\vec{\mathbf{w}}_r - \vec{\mathbf{w}}_r(0)\right\|_2 \leq R$, it follows that $\mathbb{I}_{\left\{\vec{\mathbf{w}}_r(t')^\top \vec{\mathbf{x}}_j \geq 0\right\}} = \mathbb{I}_{\left\{\vec{\mathbf{w}}_r(0)^\top \vec{\mathbf{x}}_j \geq 0\right\}}$ when $\left|\vec{\mathbf{w}}_r(0)^\top \mathbf{x}'_{j'}\right| > R$ for all $j' \in [n]$. Therefore,

$$|d_{t',r,j'}| = \left| \mathbb{I}_{\left\{\vec{\mathbf{w}}_r(t')^\top \vec{\mathbf{x}}_j \geq 0\right\}} - \mathbb{I}_{\left\{\vec{\mathbf{w}}_r(0)^\top \vec{\mathbf{x}}_j \geq 0\right\}} \right| \leq \mathbb{I}_{\left\{\left|\vec{\mathbf{w}}_r(0)^\top \vec{\mathbf{x}}_j\right| \leq R\right\}},$$

and it follows that

$$\frac{\left| \sum_{r=1}^{m} \mathbb{I}_{\left\{\vec{\mathbf{w}}_r(0)^\top \vec{\mathbf{x}}_i \geq 0\right\}} d_{t',r,j} \right|}{m} \leq \frac{\sum_{r=1}^{m} |d_{t',r,j}|}{m} \leq \frac{\sum_{r=1}^{m} \mathbb{I}_{\left\{\left|\vec{\mathbf{w}}_r(0)^\top \vec{\mathbf{x}}_j\right| \leq R\right\}}}{m} = \widehat{v}_R(\mathbf{W}(0), \vec{\mathbf{x}}_j)$$

$$\leq \frac{2R}{\sqrt{2\pi}\kappa} + C_2(m/2, R_0, \delta), \tag{96}$$

where $\widehat{v}_R$ is defined by (15), and the last inequality follows from Theorem C.2.

It follows from (96) that $\|\mathbf{X}'\|_2 \leq \sqrt{n} \left( \frac{2R}{\sqrt{2\pi}\kappa} + C_2(m/2, R_0, \delta) \right)$, and we have

$$\|E_2(\mathbf{x})\|_\infty \leq \frac{\eta}{n} \|\mathbf{X}'\|_2 \|\mathbf{u}(t')\|_2 \|\mathbf{x}\|_2$$

$$\leq \eta c_{\mathbf{u}} \left( \frac{2R}{\sqrt{2\pi}\kappa} + C_2(m/2, R_0, \delta) \right). \tag{97}$$

Combining (93), (95), and (97), for any $t' \in [0, t-1]$,

$$\|G_{t'}(\mathbf{x}) - h(\mathbf{x}, t')\|_\infty \leq \|E_1\|_\infty + \|E_2\|_\infty$$

$$\leq \eta c_{\mathbf{u}} \left( C_1(m/2, d, \delta) + \frac{2R}{\sqrt{2\pi}\kappa} + C_2(m/2, R_0, \delta) \right). \tag{98}$$

Define $e(\cdot) = f(\mathbf{W}, \cdot) - h(\cdot)$, it then follows from (91), (92), and (98) that

$$
\begin{aligned}
\|e(\mathbf{x})\|_\infty &\le \|f(\mathbf{W}, \cdot) - g\|_\infty + \|g - h\|_\infty \\
&\le \|f(\mathbf{W}, \cdot) - g\|_\infty + \sum_{t'=0}^{t-1} \|G_{t'} - h(\cdot, t')\|_{\Omega_{\varepsilon_0}, \mathbf{Q}} \\
&\overset{\textcircled{2}}{\le} 2\eta c_\mathbf{u} T \left( \frac{2R}{\sqrt{2\pi}\kappa} + C_2(m/2, R_0, \delta) \right) \\
&\quad + \eta c_\mathbf{u} T \left( C_1(m/2, d, \delta) + \frac{2R}{\sqrt{2\pi}\kappa} + C_2(m/2, R_0, \delta) \right) \\
&\le \eta c_\mathbf{u} T \left( C_1(m/2, d, \delta) + 3 \left( \frac{2R}{\sqrt{2\pi}\kappa} + C_2(m/2, R_0, \delta) \right) \right) \\
&:= \Delta_{m,n,N,c_x,\eta,\tau,\delta}.
\end{aligned}
\tag{99}
$$

We now give estimates for $\Delta_{m,n,N,c_x,\eta,\tau,\delta}$. Since $m \ge \max\{d, n, 4\}$, we have $\sqrt{\frac{d\log m}{m}} \le \frac{\sqrt{d}}{m^{1/4}}$. As a result,

$$
\Delta_{m,n,N,c_x,\eta,\tau,\delta} \lesssim \eta T \left( \frac{\sqrt{d}}{m^{1/4}} + \frac{\eta T}{\sqrt{m}} \right).
$$

By direct calculations, for any $w > 0$, when

$$
m \gtrsim \frac{(\eta T)^4 \left( \sqrt{d} + 1 \right)^4}{w^4},
$$

we have $\Delta_{m,n,N,c_x,\eta,\tau,\delta} \le w$.

It follows from Lemma C.7 that with probability at least $1 - \exp\left(-\Theta(n\widehat{\varepsilon}_n^2)\right)$ over the random noise $\mathbf{w}$,

$$
\|h\|_{\mathcal{H}_K} \le B_h,
\tag{100}
$$

where $B_h$ is defined in (35), and $\tau$ are required to satisfy

$$
\tau \lesssim 1/(\eta T).
$$

Lemma C.3 requires that $m \gtrsim (\eta T)^4 (\sqrt{d} + \sqrt{\tau})^4 / \tau^4$. As a result, we have

$$
m \gtrsim (\eta T)^8 d^2.
$$

It also follows from the Cauchy-Schwarz inequality that $\|h\|_\infty \le B_h/\sqrt{2}$. This together with (99) proves (87). $\qquad\square$

For $B, w > 0$, we define the function class

$$
\mathcal{F}(B, w) := \{f \colon \exists h \in \mathcal{H}_K(B), \exists e \in L^\infty, \|e\|_\infty \le w \text{ s.t. } f = h + e\}.
\tag{101}
$$

**Lemma C.9.** For every $B, w > 0$ every $r > 0$,

$$
\mathfrak{R}\left(\{f \in \mathcal{F}(B, w) \colon \mathbb{E}_P\left[f^2\right] \le r\}\right) \le \varphi_{B,w}(r),
\tag{102}
$$

where

$$
\varphi_{B,w}(r) := \min_{Q \colon Q \ge 0} \left( (\sqrt{r} + w)\sqrt{\frac{Q}{n}} + B \left( \frac{\sum_{q=Q+1}^\infty \lambda_q}{n} \right)^{1/2} \right) + w.
\tag{103}
$$

*Proof.* We first decompose the Rademacher complexity of the function class $\left\{f \in \mathcal{F}(B, w) \colon \mathbb{E}_P\left[f^2\right] \leq r\right\}$ into two terms as follows:

$$\mathfrak{R}\left(\left\{f \colon f \in \mathcal{F}(B, w), \mathbb{E}_P\left[f^2\right] \leq r\right\}\right)$$

$$\leq \underbrace{\frac{1}{n}\mathbb{E}\left[\sup_{f \in \mathcal{F}(B,w)\colon \mathbb{E}_P[f^2] \leq r} \sum_{i=1}^n \sigma_i h(\vec{\mathbf{x}}_i)\right]}_{:=\mathcal{R}_1} + \underbrace{\frac{1}{n}\mathbb{E}\left[\sup_{f \in \mathcal{F}(B,w)\colon \mathbb{E}_P[f^2] \leq r} \sum_{i=1}^n \sigma_i e(\vec{\mathbf{x}}_i)\right]}_{:=\mathcal{R}_2}. \quad (104)$$

We now analyze the upper bounds for $\mathcal{R}_1, \mathcal{R}_2$ on the RHS of (104).

**Derivation for the upper bound for $\mathcal{R}_1$.**

According to Definition 101 and Theorem C.8, for any $f \in \mathcal{F}(B, w)$, we have $f = h + e$ with $h \in \mathcal{H}_K(B)$, $e \in L^\infty$, $\|e\|_\infty \leq w$.

When $\mathbb{E}_P\left[f^2\right] \leq r$, it follows from the triangle inequality that $\|h\|_{L^2} \leq \|f\|_{L^2} + \|e\|_{L^2} \leq \sqrt{r} + w := r_h$. We now consider $h \in \mathcal{H}_K(B)$ with $\|h\|_{L^2} \leq r_h$ in the remaining of this proof. We have

$$\sum_{i=1}^n \sigma_i f(\vec{\mathbf{x}}_i) = \sum_{i=1}^n \sigma_i \left(h(\vec{\mathbf{x}}_i) + e(\vec{\mathbf{x}}_i)\right)$$

$$= \left\langle h, \sum_{i=1}^n \sigma_i K(\cdot, \vec{\mathbf{x}}_i)\right\rangle_{\mathcal{H}_K} + \sum_{i=1}^n \sigma_i e(\vec{\mathbf{x}}_i). \quad (105)$$

Because $\{v_q\}_{q \geq 1}$ is an orthonormal basis of $\mathcal{H}_K$, for any $0 \leq Q \leq n$, we further express the first term on the RHS of (105) as

$$\left\langle h, \sum_{i=1}^n \sigma_i K(\cdot, \vec{\mathbf{x}}_i)\right\rangle_{\mathcal{H}_K}$$

$$= \left\langle \sum_{q=1}^Q \sqrt{\lambda_q}\langle h, v_q\rangle_{\mathcal{H}_K} v_q, \sum_{q=1}^Q \frac{1}{\sqrt{\lambda_q}}\left\langle \sum_{i=1}^n \sigma_i K(\cdot, \vec{\mathbf{x}}_i), v_q\right\rangle_{\mathcal{H}_K} v_q\right\rangle_{\mathcal{H}_K}$$

$$+ \left\langle h, \sum_{q>Q}\left\langle \sum_{i=1}^n \sigma_i K(\cdot, \vec{\mathbf{x}}_i), v_q\right\rangle_{\mathcal{H}_K} v_q\right\rangle_{\mathcal{H}_K}. \quad (106)$$

Due to the fact that $h \in \mathcal{H}_K$, $h = \sum_{q=1}^\infty \boldsymbol{\beta}_q^{(h)} v_q = \sum_{q=1}^\infty \sqrt{\lambda_q}\boldsymbol{\beta}_q^{(h)} e_q$ with $v_q = \sqrt{\lambda_q}e_q$. Therefore, $\|h\|_{L^2}^2 = \sum_{q=1}^\infty \lambda_q \boldsymbol{\beta}_q^{(h)^2}$, and

$$\left\|\sum_{q=1}^Q \sqrt{\lambda_q}\langle h, v_q\rangle_{\mathcal{H}_K} v_q\right\|_{\mathcal{H}_K} = \left\|\sum_{q=1}^Q \sqrt{\lambda_q}\boldsymbol{\beta}_q^{(h)} v_q\right\|_{\mathcal{H}_K}$$

$$= \sqrt{\sum_{q=1}^Q \lambda_q \boldsymbol{\beta}_q^{(h)^2}} \leq \|h\|_{L^2} \leq r_h. \quad (107)$$

According to Mercer's Theorem, because the kernel $K$ is continuous symmetric positive definite, it has the decomposition

$$K(\cdot, \vec{\mathbf{x}}_i) = \sum_{j=1}^\infty \lambda_j e_j(\cdot)e_j(\vec{\mathbf{x}}_i),$$

so that we have

$$\left\langle \sum_{i=1}^n \sigma_i K(\cdot, \vec{\mathbf{x}}_i), v_q\right\rangle_{\mathcal{H}_K} = \left\langle \sum_{i=1}^n \sigma_i \sum_{j=1}^\infty \lambda_j e_j e_j(\vec{\mathbf{x}}_i), v_q\right\rangle_{\mathcal{H}_K}$$

$$= \left\langle \sum_{i=1}^n \sigma_i \sum_{j=1}^\infty \sqrt{\lambda_j} e_j(\vec{\mathbf{x}}_i) \cdot v_j, v_q \right\rangle_{\mathcal{H}_K}$$

$$= \sum_{i=1}^n \sigma_i \sqrt{\lambda_q} e_q(\vec{\mathbf{x}}_i). \tag{108}$$

Combining (106), (107), and (108), we have

$$\left\langle h, \sum_{i=1}^n \sigma_i K(\cdot, \vec{\mathbf{x}}_i) \right\rangle$$

$$\overset{①}{\leq} \left\| \sum_{q=1}^Q \sqrt{\lambda_q} \langle h, v_q \rangle_{\mathcal{H}_K} v_q \right\|_{\mathcal{H}_K} \cdot \left\| \sum_{q=1}^Q \frac{1}{\sqrt{\lambda_q}} \left\langle \sum_{i=1}^n \sigma_i K(\cdot, \vec{\mathbf{x}}_i), v_q \right\rangle_{\mathcal{H}_K} v_q \right\|_{\mathcal{H}_K}$$

$$+ \|h\|_{\mathcal{H}_K} \cdot \left\| \sum_{q=Q+1}^\infty \left\langle \sum_{i=1}^n \sigma_i K(\cdot, \vec{\mathbf{x}}_i), v_q \right\rangle_{\mathcal{H}_K} v_q \right\|_{\mathcal{H}_K}$$

$$\leq \|h\|_{L^2} \left\| \sum_{q=1}^Q \sum_{i=1}^n \sigma_i e_q(\vec{\mathbf{x}}_i) v_q \right\|_{\mathcal{H}_K} + B \left\| \sum_{q=Q+1}^\infty \sum_{i=1}^n \sigma_i \sqrt{\lambda_q} e_q(\vec{\mathbf{x}}_i) v_q \right\|_{\mathcal{H}_K}$$

$$\leq r_h \sqrt{\sum_{q=1}^Q \left( \sum_{i=1}^n \sigma_i e_q(\vec{\mathbf{x}}_i) \right)^2} + B \sqrt{\sum_{q=Q+1}^\infty \left( \sum_{i=1}^n \sigma_i \sqrt{\lambda_q} e_q(\vec{\mathbf{x}}_i) \right)^2}, \tag{109}$$

where ① is due to Cauchy-Schwarz inequality. Moreover, by Jensen's inequality we have

$$\mathbb{E} \left[ \sqrt{\sum_{q=1}^Q \left( \sum_{i=1}^n \sigma_i e_q(\vec{\mathbf{x}}_i) \right)^2} \right] \leq \sqrt{\mathbb{E} \left[ \sum_{q=1}^Q \left( \sum_{i=1}^n \sigma_i e_q(\vec{\mathbf{x}}_i) \right)^2 \right]}$$

$$\leq \sqrt{\mathbb{E} \left[ \sum_{q=1}^Q \sum_{i=1}^n e_q^2(\vec{\mathbf{x}}_i) \right]} = \sqrt{nQ}. \tag{110}$$

and similarly,

$$\mathbb{E} \left[ \sqrt{\sum_{q=Q+1}^\infty \left( \sum_{i=1}^n \sigma_i \sqrt{\lambda_q} e_q(\vec{\mathbf{x}}_i) \right)^2} \right] \leq \sqrt{\mathbb{E} \left[ \sum_{q=Q+1}^\infty \lambda_q \sum_{i=1}^n e_q^2(\vec{\mathbf{x}}_i) \right]} = \sqrt{n \sum_{q=Q+1}^\infty \lambda_q}. \tag{111}$$

Since (109)-(111) hold for all $Q \geq 0$, it follows that

$$\mathbb{E} \left[ \sup_{h \in \mathcal{H}_K(B), \|h\|_{L^2} \leq r_h} \frac{1}{n} \sum_{i=1}^n \sigma_i h(\vec{\mathbf{x}}_i) \right] \leq \min_{Q : Q \geq 0} \left( r_h \sqrt{nQ} + B \sqrt{n \sum_{q=Q+1}^\infty \lambda_q} \right). \tag{112}$$

It follows from (104), (105), and (112) that

$$\mathcal{R}_1 \leq \frac{1}{n} \mathbb{E} \left[ \sup_{h \in \mathcal{H}_K(B), \|h\|_{L^2} \leq r_h} \sum_{i=1}^n \sigma_i h(\vec{\mathbf{x}}_i) \right]$$

$$\leq \min_{Q : Q \geq 0} \left( r_h \sqrt{\frac{Q}{n}} + B \left( \frac{\sum_{q=Q+1}^\infty \lambda_q}{n} \right)^{1/2} \right). \tag{113}$$

**Derivation for the upper bound for $\mathcal{R}_2$.**

Because $\left|1/n \sum_{i=1}^{n} \sigma_i e(\vec{\mathbf{x}}_i)\right| \le w$ when $\|e\|_\infty \le w$, we have

$$\mathcal{R}_2 \le \frac{1}{n} \mathbb{E} \left[ \sup_{e \in L^\infty \colon \|e\|_\infty \le w} \sum_{i=1}^{n} \sigma_i e(\vec{\mathbf{x}}_i) \right] \le w. \tag{114}$$

It follows from (113) and (114) that

$$\mathfrak{R}\left(\left\{ f \colon f \in \mathcal{F}(B, w), \mathbb{E}_P\left[f^2\right] \le r \right\}\right)$$

$$\le \min_{Q \colon Q \ge 0} \left( r_h \sqrt{\frac{Q}{n}} + B \left( \frac{\sum_{q=Q+1}^{\infty} \lambda_q}{n} \right)^{1/2} \right) + w.$$

Plugging $r_h$ in the RHS of the above inequality completes the proof. $\qquad\square$

**Theorem C.10.** Suppose $w \in (0, 1)$ and $m$ satisfy

$$m \gtrsim \max \left\{ \frac{(\eta T)^4 \left(\sqrt{d}+1\right)^4}{w^4}, (\eta T)^8 d^2 \right\}, \tag{115}$$

and the neural network $f(\mathbf{W}(t), \cdot)$ is trained by GD in Algorithm 1 with the learning rate $\eta \in (0, 1/\widehat{\lambda}_1)$ on random initialization $\mathbf{W}(0)$, and $T \le \widehat{T}$. Then for every $t \in [T]$, with probability at least $1 - \exp\left(-\Theta(n)\right) - \exp\left(-\Theta(n\widehat{\varepsilon}_n^2)\right) - \exp\left(-n\varepsilon_n^2\right) - 2/n$ over the random noise $\mathbf{w}$, the random training features $\mathbf{S}$ and the random initialization $\mathbf{W}(0)$,

$$\mathbb{E}_P\left[(f_t - f^*)^2\right] - 2\mathbb{E}_{P_n}\left[(f_t - f^*)^2\right]$$

$$\le c_0 \min_{0 \le Q \le n} \left( \frac{B_0 Q}{n} + w \left( \sqrt{\frac{Q}{n}} + 1 \right) + B_h \left( \frac{\sum_{q=Q+1}^{\infty} \lambda_q}{n} \right)^{1/2} \right)^2, \tag{116}$$

Furthermore, with probability at least $1 - \exp\left(-\Theta(n)\right) - \exp\left(-\Theta(n\widehat{\varepsilon}_n^2)\right) - \exp\left(-n\varepsilon_n^2\right) - 2/n$ over the random noise $\mathbf{w}$, the random training features $\mathbf{S}$ and the random initialization $\mathbf{W}(0)$,

$$\mathbb{E}_P\left[(f_t - f^*)^2\right] - 2\mathbb{E}_{P_n}\left[(f_t - f^*)^2\right] \le c_0'(\varepsilon_n^2 + w). \tag{117}$$

Here $B_0, c_0, c_0'$ are absolute positive constants depending on $\mu_0$, and $c_0'$ also depends on $\sigma$.

*Proof.* We first remark that the conditions on $m$, (115), is required by Lemma C.3 and Theorem C.8. It follows from Lemma C.3 and Theorem C.8 that for every $t \in [T]$, conditioned on an event $\Omega$ with probability at least $1 - \exp\left(-\Theta(n)\right) - \exp\left(-\Theta(n\widehat{\varepsilon}_n^2)\right)$ over the random noise $\mathbf{w}$, we have $\mathbf{W}(t) \in \mathcal{W}(\mathbf{S}, \mathbf{W}(0), T)$, and

$$f(\mathbf{W}(t), \cdot) = f_t \in \mathcal{F}_{\mathrm{NN}}(\mathbf{S}, \mathbf{W}(0), T).$$

Moreover, conditioned on the event $\Omega$,

$$f_t \in \mathcal{F}_{\mathrm{ext}}(\mathbf{Q}, w, T).$$

We then derive the sharp upper bound for $\mathbb{E}_P\left[(f_t - f^*)^2\right]$ by applying Theorem A.3 to the function class

$$\mathcal{F} = \left\{ F = (f - f^*)^2 \colon f \in \mathcal{F}(B_h, w) \right\}.$$

Let $B_0 := B_h/\sqrt{2} + 1 + \mu_0/\sqrt{2} \geq B_h/\sqrt{2} + w + \mu_0/\sqrt{2}$, then we have $\|F\|_\infty \leq B_0^2$ with $F \in \mathcal{F}$, so that $\mathbb{E}_P[F^2] \leq B_0^2 \mathbb{E}_P[F]$. Let $T(F) = B_0^2 \mathbb{E}_P[F]$ for $F \in \mathcal{F}$. Then $\mathrm{Var}[F] \leq \mathbb{E}_P[F^2] \leq T(F) = B_0^2 \mathbb{E}_P[F]$.

We have

$$
\begin{aligned}
&\mathfrak{R}\left(\{F \in \mathcal{F}: T(F) \leq r\}\right) \\
&= \mathfrak{R}\left(\left\{(f - f^*)^2 : f \in \mathcal{F}(B_h, w), \mathbb{E}_P\left[(f - f^*)^2\right] \leq \frac{r}{B_0^2}\right\}\right) \\
&\overset{\text{①}}{\leq} 2B_0\mathfrak{R}\left(\left\{f - f^* : f \in \mathcal{F}(B_h, w), \mathbb{E}_P\left[(f - f^*)^2\right] \leq \frac{r}{B_0^2}\right\}\right) \\
&\overset{\text{②}}{\leq} 4B_0\mathfrak{R}\left(\left\{f \in \mathcal{F}(B_h, w): \mathbb{E}_P\left[f^2\right] \leq \frac{r}{4B_0^2}\right\}\right),
\end{aligned}
\tag{118}
$$

where ① is due to the contraction property of Rademacher complexity in Theorem A.2. Since $f^* \in \mathcal{F}(B_h, w)$, $f \in \mathcal{F}(B_h, w)$, we have $\frac{f - f^*}{2} \in \mathcal{F}(B_h, w)$ due to the fact that $\mathcal{F}(B_h, w)$ is symmetric and convex, and it follows that ② holds.

It follows from (118) and Lemma C.9 that

$$
B_0^2\mathfrak{R}\left(\{F \in \mathcal{F}: T(F) \leq r\}\right) \leq 4B_0^3\mathfrak{R}\left(\left\{f: f \in \mathcal{F}(B_h, w), \mathbb{E}_P\left[f^2\right] \leq \frac{r}{4B_0^2}\right\}\right)
$$

$$
\leq 4B_0^3\varphi_{B_h, w}\left(\frac{r}{4B_0^2}\right) := \psi(r). \tag{119}
$$

$\psi$ defined as the RHS of (119) is a sub-root function since it is nonnegative, nondecreasing and $\frac{\psi(r)}{\sqrt{r}}$ is nonincreasing. Let $r^*$ be the fixed point of $\psi$, and $0 \leq r \leq r^*$. It follows from (Bartlett et al., 2005, Lemma 3.2) that $0 \leq r \leq \psi(r) = 4B_0^3\varphi\left(\frac{r}{4B_0^2}\right)$. Therefore, by the definition of $\varphi$ in (103), for every $0 \leq Q \leq n$, we have

$$
\frac{r}{4B_0^3} \leq \left(\frac{\sqrt{r}}{2B_0} + w\right)\sqrt{\frac{Q}{n}} + B_h\left(\frac{\sum\limits_{q=Q+1}^{\infty} \lambda_q}{n}\right)^{1/2} + w. \tag{120}
$$

Solving the quadratic inequality (120) for $r$, we have

$$
r \leq \frac{8B_0^4 Q}{n} + 8B_0^3\left(w\left(\sqrt{\frac{Q}{n}} + 1\right) + B_h\left(\frac{\sum\limits_{q=Q+1}^{\infty} \lambda_q}{n}\right)^{1/2}\right). \tag{121}
$$

(121) holds for every $0 \leq Q \leq n$, so we have

$$
r \leq 8B_0^3 \min_{0 \leq Q \leq n}\left(\frac{B_0 Q}{n} + w\left(\sqrt{\frac{Q}{n}} + 1\right) + B_h\left(\frac{\sum\limits_{q=Q+1}^{\infty} \lambda_q}{n}\right)^{1/2}\right). \tag{122}
$$

It then follows from (119) and Theorem A.3 that with probability at least $1 - \exp(-x)$ over the random training features $\mathbf{S}$,

$$
\mathbb{E}_P\left[(f_t - f^*)^2\right] - \frac{K_0}{K_0 - 1}\mathbb{E}_{P_n}\left[(f_t - f^*)^2\right] - \frac{x\left(11B_0^2 + 26B_0^2 K_0\right)}{n} \leq \frac{704K_0}{B_0^2}r^*, \tag{123}
$$

or

$$
\mathbb{E}_P\left[(f_t - f^*)^2\right] - 2\mathbb{E}_{P_n}\left[(f_t - f^*)^2\right] \lesssim r^* + \frac{x}{n}, \tag{124}
$$

with $K_0 = 2$ in (123).

It follows from (122) and (124) that

$$\mathbb{E}_P \left[ (f_t - f^*)^2 \right] - 2\mathbb{E}_{P_n} \left[ (f_t - f^*)^2 \right]$$

$$\lesssim \min_{0 \leq Q \leq n} \left( \frac{B_0 Q}{n} + w \left( \sqrt{\frac{Q}{n}} + 1 \right) + B_h \left( \frac{\sum\limits_{q=Q+1}^{\infty} \lambda_q}{n} \right)^{1/2} \right) + \frac{x}{n}.$$

Let $x = n\varepsilon_n^2$ in the above inequality, and we note that the above argument requires Theorem C.8 which holds with probability at least $1 - \exp\left(-\Theta(n)\right) - \exp\left(-\Theta(n\widehat{\varepsilon}_n^2)\right)$ over the random noise $\mathbf{w}$. Then (116) is proved combined with the facts that $\Pr\left[\mathcal{W}_0\right] \geq 1 - 2/n$.

We now prove (117). First, it follows from the definition of $\varphi_{B_h, w}$ in (103) that

$$\psi(r) = 4B_0^3 \varphi_{B_h, w} \left( \frac{r}{4B_0^2} \right)$$

$$= 4B_0^3 \min_{Q: Q \geq 0} \left( \left( \frac{\sqrt{r}}{2B_0} + w \right) \sqrt{\frac{Q}{n}} + B_h \left( \frac{\sum\limits_{q=Q+1}^{\infty} \lambda_q}{n} \right)^{1/2} \right) + 4B_0^3 w$$

$$\leq 4B_0^3 B_h \min_{Q: Q \geq 0} \left( \sqrt{\frac{Qr}{n}} + \left( \frac{\sum\limits_{q=Q+1}^{\infty} \lambda_q}{n} \right)^{1/2} \right) + 4B_0^3 w \left( \sqrt{\frac{Q}{n}} + 1 \right)$$

$$\leq \frac{4\sqrt{2}B_0^3 B_h}{\sigma} \cdot \sigma R_K(\sqrt{r}) + 8B_0^3 w := \psi_1(r),$$

where the last inequality follows from the Cauchy-Schwarz inequality. It can be verified that $\psi_1(r)$ is a sub-root function. Let the fixed point of $\psi_1(r)$ be $r_1^*$. Because the fixed point of $\sigma R_K(\sqrt{r})$ as a function of $r$ is $\varepsilon_n^2$, it follows from Lemma C.17 that

$$r_1^* \leq \max\left\{ \frac{32\sqrt{2}B_0^6 B_h^2}{\sigma^2}, 1 \right\} \varepsilon_n^2 + 16B_0^3 w. \tag{125}$$

It then follows from Theorem A.3 with $K_0 = 2$ that with probability at least $1 - \exp(-x)$,

$$\mathbb{E}_P \left[ (f_t - f^*)^2 \right] - 2\mathbb{E}_{P_n} \left[ (f_t - f^*)^2 \right] \lesssim r_1^* + \frac{x}{n}.$$

Letting $x = n\varepsilon_n^2$, then plugging the upper bound for $r_1^*$, (125), in the above inequality leads to

$$\mathbb{E}_P \left[ (f_t - f^*)^2 \right] - 2\mathbb{E}_{P_n} \left[ (f_t - f^*)^2 \right] \lesssim \varepsilon_n^2 + 16B_0^3 w. \tag{126}$$

Again, we note that the above argument requires Theorem C.8 which holds with probability at least $1 - \exp\left(-\Theta(n)\right) - \exp\left(-\Theta(n\widehat{\varepsilon}_n^2)\right)$ over the random noise $\mathbf{w}$. Then (117) is proved with the fact that $\Pr\left[\mathcal{W}_0\right] \geq 1 - 2/n$ and (126).

$\square$

**Theorem C.11.** Suppose the neural network trained after the $t$-th step of gradient descent, $f_t = f(\mathbf{W}(t), \cdot)$, satisfies $\mathbf{u}(t) = f_t(\mathbf{S}) - \mathbf{y} = \mathbf{v}(t) + \mathbf{e}(t)$ with $\mathbf{v}(t) \in \mathcal{V}_t$ and $\mathbf{e}(t) \in \mathcal{E}_{t,\tau}$ and $T \leq \widehat{T}$. If

$$\eta \in [1, 2), \quad \tau \leq \frac{1}{\eta T}, \tag{127}$$

then for every $t \in [T]$, with probability at least $1 - \exp\left(-\Theta(n\widehat{\varepsilon}_n^2)\right)$ over the random noise $\mathbf{w}$, we have

$$\mathbb{E}_{P_n} \left[ (f_t - f^*)^2 \right] \leq \frac{3}{\eta t} \left( \frac{\mu_0^2}{2e} + 3 \right). \tag{128}$$

*Proof.* We have

$$f_t(\mathbf{S}) = f^*(\mathbf{S}) + \mathbf{w} + \mathbf{v}(t) + \mathbf{e}(t), \tag{129}$$

where $\mathbf{v}(t) \in \mathcal{V}_t$, $\mathbf{e}(t) \in \mathcal{E}_{t,\tau}$, $\vec{\mathbf{e}}(t) = \vec{\mathbf{e}}_1(t) + \vec{\mathbf{e}}_2(t)$ with $\vec{\mathbf{e}}_1(t) = -(\mathbf{I}_n - \eta\mathbf{K}_n)^t\mathbf{w}$ and $\left\|\vec{\mathbf{e}}_2(t)\right\|_2 \leq \sqrt{n}\tau$. We have $\eta\lambda_1 \in (0,1)$ if $\eta \in [1,2)$. It follows from (129) that

$$\mathbb{E}_{P_n}\left[(f_t - f^*)^2\right] = \frac{1}{n}\|f_t(\mathbf{S}) - f^*(\mathbf{S})\|_2^2 = \frac{1}{n}\|\mathbf{v}(t) + \mathbf{w} + \mathbf{e}(t)\|_2^2$$

$$= \frac{1}{n}\left\|-(\mathbf{I} - \eta\mathbf{K}_n)^t f^*(\mathbf{S}) + \left(\mathbf{I}_n - (\mathbf{I}_n - \eta\mathbf{K}_n)^t\right)\mathbf{w} + \vec{\mathbf{e}}_2(t)\right\|_2^2$$

$$\overset{①}{\leq} \frac{3}{n}\sum_{i=1}^n\left(1 - \eta\widehat{\lambda}_i\right)^{2t}\left[\mathbf{U}^\top f^*(\mathbf{S})\right]_i^2 + \frac{3}{n}\sum_{i=1}^n\left(1 - \left(1 - \eta\widehat{\lambda}_i\right)^t\right)^2\left[\mathbf{U}^\top\mathbf{w}\right]_i^2 + \frac{3}{n}\left\|\vec{\mathbf{e}}_2(t)\right\|_2^2$$

$$\overset{②}{\leq} \frac{3\mu_0^2}{2e\eta t} + \frac{3}{n}\sum_{i=1}^n\left(1 - (1 - \eta\lambda_i)^t\right)^2\left[\mathbf{U}^\top\mathbf{w}\right]_i^2 + 3\tau^2$$

$$\leq \frac{3}{\eta t}\left(\frac{\mu_0^2}{2e} + \frac{1}{\eta}\right) + 3\cdot\underbrace{\frac{1}{n}\sum_{i=1}^n\left(1 - (1 - \eta\lambda_i)^t\right)^2\left[\mathbf{U}^\top\mathbf{w}\right]_i^2}_{:=E_\varepsilon}$$

$$\leq \frac{3}{\eta t}\left(\frac{\mu_0^2}{2e} + 2\widehat{\lambda}_1\right) + 3E_\varepsilon \leq \frac{3}{\eta t}\left(\frac{\mu_0^2}{2e} + 4\right) + 3E_\varepsilon. \tag{130}$$

Here ① follows from the Cauchy-Schwarz inequality, ② follows from (49) in the proof of Lemma C.4. We then derive the upper bound for $E_\varepsilon$ on the RHS of (130). We define the diagonal matrix $\mathbf{R} \in \mathbb{R}^{n\times n}$ with $\mathbf{R}_{ii} = \left(1 - (1 - \eta\lambda_i)^t\right)^2$. Then we have

$$E_\varepsilon = 1/n \cdot \mathrm{tr}\left(\mathbf{U}\mathbf{R}\mathbf{U}^\top\mathbf{w}\mathbf{w}^\top\right)$$

It follows from (Wright, 1973) that

$$\Pr\left[1/n \cdot \mathrm{tr}\left(\mathbf{U}\mathbf{R}\mathbf{U}^\top\mathbf{w}\mathbf{w}^\top\right) - \mathbb{E}\left[1/n \cdot \mathrm{tr}\left(\mathbf{U}\mathbf{R}\mathbf{U}^\top\mathbf{w}\mathbf{w}^\top\right)\right] \geq u\right]$$
$$\leq \exp\left(-c\min\left\{nu/\|\mathbf{R}\|_2, n^2u^2/\|\mathbf{R}\|_F^2\right\}\right). \tag{131}$$

for all $u > 0$, and $c$ is a positive constant. With $\eta_t = \eta t$ for all $t \geq 0$, we have

$$\mathbb{E}\left[1/n \cdot \mathrm{tr}\left(\mathbf{U}\mathbf{R}\mathbf{U}^\top\mathbf{w}\mathbf{w}^\top\right)\right] \leq \frac{\sigma^2}{n}\sum_{i=1}^n\left(1 - \left(1 - \eta\widehat{\lambda}_i\right)^t\right)^2$$

$$\overset{①}{\leq} \frac{\sigma^2}{n}\sum_{i=1}^n\min\left\{1, \eta_t^2\widehat{\lambda}_i^2\right\}$$

$$\leq \frac{\sigma^2\eta_t}{n}\sum_{i=1}^n\min\left\{\frac{1}{\eta_t}, \eta_t\widehat{\lambda}_i^2\right\}$$

$$\overset{②}{\leq} \frac{\sigma^2\eta_t}{n}\sum_{i=1}^n\min\left\{\frac{1}{\eta_t}, \widehat{\lambda}_i\right\}$$

$$= \sigma^2\eta_t\widehat{R}_K^2(\sqrt{1/\eta_t}) \leq \frac{1}{\eta_t}. \tag{132}$$

Here ① follows from the fact that $(1 - \eta\widehat{\lambda}_i)^t \geq \max\left\{0, 1 - t\eta\widehat{\lambda}_i\right\}$, and ② follows from $\min\{a,b\} \leq \sqrt{ab}$ for any nonnegative numbers $a, b$. Because $t \leq T \leq \widehat{T}$, we have $R_K(\sqrt{1/\eta_t}) \leq 1/(\sigma\eta_t)$, so the last inequality holds.

Moreover, we have the upper bounds for $\|\mathbf{R}\|_2$ and $\|\mathbf{R}\|_\mathrm{F}$ as follows. First, we have

$$\|\mathbf{R}\|_2 \leq \max_{i \in [n]} \left( 1 - \left( 1 - \eta \widehat{\lambda}_i \right)^t \right)^2$$
$$\leq \min \left\{ 1, \eta_t^2 \widehat{\lambda}_i^2 \right\} \leq 1. \tag{133}$$

We also have

$$\frac{1}{n} \|\mathbf{R}\|_\mathrm{F}^2 = \frac{1}{n} \sum_{i=1}^n \left( 1 - \left( 1 - \eta \widehat{\lambda}_i \right)^t \right)^4$$
$$\leq \frac{\eta_t}{n} \sum_{i=1}^n \min \left\{ \frac{1}{\eta_t}, \eta_t^3 \widehat{\lambda}_i^4 \right\}$$
$$\overset{③}{\leq} \frac{\eta_t}{n} \sum_{i=1}^n \min \left\{ \widehat{\lambda}_i, \frac{1}{\eta_t} \right\} = \eta_t \widehat{R}_K^2(\sqrt{1/\eta_t}) \leq \frac{1}{\sigma^2 \eta_t}. \tag{134}$$

If $1/\eta_t \leq \eta_t^3 (\widehat{\lambda}_i)^4$, then $\min \left\{ 1/\eta_t, \eta_t^3 (\widehat{\lambda}_i)^4 \right\} = 1/\eta_t$. Otherwise, we have $\eta_t^4 \widehat{\lambda}_i^4 < 1$, so that $\eta_t \widehat{\lambda}_i < 1$ and it follows that $\min \left\{ 1/\eta_t, \eta_t^3 (\widehat{\lambda}_i)^4 \right\} \leq \eta_t^3 \widehat{\lambda}_i^4 \leq \widehat{\lambda}_i$. As a result, ③ holds.

Combining (131)- (134), we have

$$\Pr \left[ 1/n \cdot \mathrm{tr} \left( \mathbf{U} \mathbf{R} \mathbf{U}^\top \mathbf{w} \mathbf{w}^\top \right) - \mathbb{E} \left[ 1/n \cdot \mathrm{tr} \left( \mathbf{U} \mathbf{R} \mathbf{U}^\top \mathbf{w} \mathbf{w}^\top \right) \right] \geq u \right] \leq \exp \left( -cn \min \left\{ u, u^2 \sigma^2 \eta_t \right\} \right).$$

Let $u = 1/\eta_t$ in the above inequality, we have

$$\exp \left( -cn \min \left\{ u, u^2 \sigma^2 \eta_t \right\} \right) = \exp \left( -c'n/\eta_t \right) \leq \exp \left( -c'n \widehat{\varepsilon}_n^2 \right)$$

where $c' = c \min \left\{ 1, \sigma^2 \right\}$, and the last inequality is due to the fact that $1/\eta_t \geq \widehat{\varepsilon}_n^2$ since $t \leq T \leq \widehat{T}$. It follows that with probability at least $1 - \exp \left( -\Theta(n \widehat{\varepsilon}_n^2) \right)$,

$$E_\varepsilon \leq u + \frac{1}{\eta_t} = \frac{2}{\eta_t}. \tag{135}$$

It then follows from (130), (131)-(135) that

$$\mathbb{E}_{P_n} \left[ (f_t - f^*)^2 \right] \leq \frac{3}{\eta t} \left( \frac{\mu_0^2}{2e} + 6 \right)$$

holds with probability at least $1 - \exp \left( -c'n \widehat{\varepsilon}_n^2 \right)$.

$\square$

### C.3 AUXILIARY RESULTS ABOUT REPRODUCING KERNEL HILBERT SPACES

**Lemma C.12** (In the proof of (Raskutti et al., 2014, Lemma 8)). For any $f \in \mathcal{H}_K(\mu_0)$, we have

$$\frac{1}{n} \sum_{i=1}^n \frac{\left[ \mathbf{U}^\top f(\mathbf{S}') \right]_i^2}{\widehat{\lambda}_i} \leq \mu_0^2. \tag{136}$$

Similarly, for $f \in \mathcal{H}_K(\mu_0)$, we have $\frac{1}{n} \sum_{i=1}^n \frac{\left[ \mathbf{U}^\top f(\mathbf{S}') \right]_i^2}{\lambda_i} \leq \mu_0^2$.

**Lemma C.13.** For any positive real number $a \in (0, 1)$ and natural number $t$, we have

$$(1 - a)^t \leq e^{-ta} \leq \frac{1}{eta}. \tag{137}$$

*Proof.* The result follows from the facts that $\log(1 - a) \leq a$ for $a \in (0, 1)$ and $\sup_{u \in \mathbb{R}} u e^{-u} \leq 1/e$.

$\square$

**Lemma C.14.** (([Rosasco et al., 2010](#), Proposition 10)) With probability $1 - \delta$ over the training data **S**, for all $j \in [n]$,

$$\left| \lambda_j - \widehat{\lambda}_j \right| \leq \sqrt{\frac{2 \log \frac{2}{\delta}}{n}}. \tag{138}$$

**Lemma C.15.** With probability at least $1 - 2 \exp(-\Theta(n \varepsilon_n^2))$,

$$\varepsilon_n^2 \leq c_1 \widehat{\varepsilon}_n^2. \tag{139}$$

Furthermore, with probability at least $1 - 2 \exp(-\Theta(n \varepsilon_n^2))$,

$$\widehat{\varepsilon}_n^2 \leq c_1 \varepsilon_n^2. \tag{140}$$

Here $c_1$ is an absolute positive constant depending on $\sigma$.

**Remark.** Lemma [C.15](#) shows that with probability at least $1 - 4 \exp(-\Theta(n \varepsilon_n^2))$, $\varepsilon_n^2 \asymp \widehat{\varepsilon}_n^2$, which is also a fact used in kernel complexity or local Rademacher based analysis for kernel regression in the statistical learning literature. We herein provide a detailed proof to ensure the mathematical rigor of this paper.

*Proof.* Define function classes

$$\mathcal{F}_t := \left\{ f \in \mathcal{H}_K \colon \|f\|_{\mathcal{H}_K} \leq 1, \|f\|_{L^2} \leq t \right\}, \quad \widehat{\mathcal{F}}_t := \left\{ f \in \mathcal{H}_K \colon \|f\|_{\mathcal{H}_K} \leq 1, \|f\|_n \leq t \right\},$$

where $\|f\|_n^2 := 1/n \cdot \sum_{i=1}^n f^2(\vec{\mathbf{x}}_i)$. Let $\mathcal{R}(t)$ be the Rademacher complexity of $\mathcal{F}_t$, that is,

$$\mathcal{R}(t) = \mathfrak{R}\left( \mathcal{F}_t \right) = \mathbb{E}_{\left\{ \vec{\mathbf{x}}_i \right\}, \{\sigma_i\}} \left[ \sup_{f \in \mathcal{F}_t} \frac{1}{n} \sum_{i=1}^n \sigma_i f(\vec{\mathbf{x}}_i), \right]$$

and we will also write $\mathcal{R}(t) = \mathbb{E} \left[ \sup_{f \in \mathcal{F}_t} \frac{1}{n} \sum_{i=1}^n \sigma_i f(\vec{\mathbf{x}}_i), \right]$ for simplicity of notations. We let $\widehat{\mathcal{R}}(t)$ be the empirical Rademacher complexity of $\mathcal{F}_t$, that is,

$$\widehat{\mathcal{R}}(t) = \mathbb{E}_\sigma \left[ \sup_{f \in \widehat{\mathcal{F}}_t} \frac{1}{n} \sum_{i=1}^n \sigma_i f(\vec{\mathbf{x}}_i). \right]$$

By results of ([Mendelson, 2002](#)), there are universal constants $c_\ell$ and $C_u$ with $0 < c_\ell < C_u$ such that when $t^2 \geq 1/n$, we have

$$c_\ell R_K(t) \leq \mathcal{R}(t) \leq C_u R_K(t), \quad c_\ell \widehat{R}_K(t) \leq \widehat{\mathcal{R}}(t) \leq C_u \widehat{R}_K(t). \tag{141}$$

When $f \in \mathcal{F}_t$, $\|f\|_\infty \leq \tau_0 = \frac{1}{\sqrt{2}}$. It follows from Lemma [C.16](#) that with probability at least $1 - \exp(-n \varepsilon_n^2)$,

$$\mathcal{F}_t \subseteq \left\{ f \in \mathcal{H}_K \colon \|f\|_{\mathcal{H}_K} \leq 1, \|f\|_n \leq \sqrt{c_2 t^2 + c_3 \varepsilon_n^2} \right\} := \widehat{\mathcal{F}}_{\sqrt{c_2 t^2 + c_3 \varepsilon_n^2}}. \tag{142}$$

Moreover, by the relation between Rademacher complexity and its empirical version in ([Bartlett et al., 2005](#), Lemma A.4), for every $x > 0$, with probability at least $1 - \exp(-x)$,

$$\mathbb{E} \left[ \sup_{f \in \widehat{\mathcal{F}}_{\sqrt{c_2 t^2 + c_3 \varepsilon_n^2}}} \frac{1}{n} \sum_{i=1}^n \sigma_i f(\vec{\mathbf{x}}_i) \right] \leq 2 \mathbb{E}_\sigma \left[ \sup_{f \in \widehat{\mathcal{F}}_{\sqrt{c_2 t^2 + c_3 \varepsilon_n^2}}} \frac{1}{n} \sum_{i=1}^n \sigma_i f(\vec{\mathbf{x}}_i) \right] + \frac{2 \tau_0 x}{n}. \tag{143}$$

As a result,

$$\mathcal{R}(t) \overset{\textcircled{1}}{\leq} \mathbb{E} \left[ \sup_{f \in \widehat{\mathcal{F}}_{\sqrt{c_2 t^2 + c_3 \varepsilon_n^2}}} \frac{1}{n} \sum_{i=1}^n \sigma_i f(\vec{\mathbf{x}}_i) \right]$$

$$\overset{②}{\leq} 2\mathbb{E}_\sigma\left[\sup_{f\in\widehat{\mathcal{F}}_{\sqrt{c_2t^2+c_3\varepsilon_n^2}}} \frac{1}{n}\sum_{i=1}^n \sigma_i f(\vec{\mathbf{x}}_i)\right] + \frac{2\tau_0 x}{n}$$

$$= 2\widehat{\mathcal{R}}(\sqrt{c_2t^2+c_3\varepsilon_n^2}) + \frac{2\tau_0 x}{n}.$$

Here ① follows from (142), and ② follows from (143). It follows from (141) and the above inequality that

$$c_\ell/\sigma \cdot \sigma R_K(t) \leq 2C_u/\sigma \cdot \sigma\widehat{R}_K(\sqrt{c_2t^2+c_3\varepsilon_n^2}) + \frac{2\tau_0 x}{n}, \forall t^2 \geq 1/n.$$

Rewrite $R_K(t)$ as a function of $r = t^2$ as $R_K(t) = F_K(r)$. Similarly, $\widehat{R}_K(t) = \widehat{F}_K(r)$ with $r = t^2$. Then we have

$$\sigma F_K(r) \leq \max\{2C_u/c_\ell, 1\}\cdot \sigma\widehat{F}_K(c_2r+c_3\varepsilon_n^2) + \frac{2\sigma\tau_0 x}{nc_\ell} := G(r), \forall r \geq 1/n. \tag{144}$$

It can be verified that $G(r)$ is a sub-root function, and let $r_G^*$ be the fixed point of $G$. Let $x \geq c_\ell/(2\sigma\tau_0)$, then $r_G^* \geq 1/n$. Moreover, $\sigma F_K(r)$ and $\sigma\widehat{F}_K(r)$ are sub-root functions, and they have fixed points $\varepsilon_n^2$ and $\widehat{\varepsilon}_n^2$, respectively. Set $r = r_G^* \geq 1/n$ in (144), we have

$$\sigma F_K(r_G^*) \leq r_G^*,$$

and it follows from the above inequality and (Bartlett et al., 2005, Lemma 3.2) that $\varepsilon_n^2 \leq r_G^*$. Since $c_2 > 1$, it then follows from the properties about the fixed point of a sub-root function in Lemma C.17 that

$$\varepsilon_n^2 \leq r_G^* \leq \max\{2C_u/c_\ell, 1\}^2\left(c_2\widehat{\varepsilon}_n^2 + \frac{2c_3\varepsilon_n^2}{c_2}\right) + \frac{4\sigma\tau_0 x}{nc_\ell}.$$

We can choose $c_2$ such that $c_2 > 2c_3\max\{2C_u/c_\ell, 1\}^2$, then the above inequality indicates that

$$\varepsilon_n^2 \leq c_{u,\ell}\widehat{\varepsilon}_n^2 + \frac{4\sigma\tau_0 x}{nc_\ell},$$

where $c_{u,\ell}$ is a constant depending on $c_\ell$, $C_u$, $c_2$, $c_3$, and (139) is proved with $x = c'n\varepsilon_n^2$ where $c' > 0$ is a positive constant which is chosen such that $4c'\sigma\tau_0/c_\ell < 1$.

Similarly, it follows from Lemma C.16 that with probability at least $1 - \exp(-n\varepsilon_n^2)$,

$$\widehat{\mathcal{F}}_t \subseteq \left\{f \in \mathcal{H}_K : \|f\|_{\mathcal{H}_K} \leq 1, \|f\|_{L^2} \leq \sqrt{c_2t^2+c_3\varepsilon_n^2}\right\} = \mathcal{F}_{\sqrt{c_2t^2+c_3\varepsilon_n^2}}. \tag{145}$$

It follows from (Bartlett et al., 2005, Lemma A.4) again that for every $x > 0$, with probability at least $1 - \exp(-x)$,

$$\mathbb{E}_\sigma\left[\sup_{f\in\mathcal{F}_{\sqrt{c_2t^2+c_3\varepsilon_n^2}}} \frac{1}{n}\sum_{i=1}^n \sigma_i f(\vec{\mathbf{x}}_i)\right] \leq 2\mathbb{E}\left[\sup_{f\in\mathcal{F}_{\sqrt{c_2t^2+c_3\varepsilon_n^2}}} \frac{1}{n}\sum_{i=1}^n \sigma_i f(\vec{\mathbf{x}}_i)\right] + \frac{10\tau_0 x}{12n}. \tag{146}$$

As a result, we have

$$\widehat{\mathcal{R}}(t) \overset{①}{\leq} \mathbb{E}_\sigma\left[\sup_{f\in\mathcal{F}_{\sqrt{c_2t^2+c_3\varepsilon_n^2}}} \frac{1}{n}\sum_{i=1}^n \sigma_i f(\vec{\mathbf{x}}_i)\right]$$

$$\overset{②}{\leq} 2\mathbb{E}\left[\sup_{f\in\mathcal{F}_{\sqrt{c_2t^2+c_3\varepsilon_n^2}}} \frac{1}{n}\sum_{i=1}^n \sigma_i f(\vec{\mathbf{x}}_i)\right] + \frac{10\tau_0 x}{12n}$$

$$= 2\mathcal{R}(\sqrt{c_2t^2+c_3\varepsilon_n^2}) + \frac{5\sqrt{2}x}{12n} \leq 2C_u R_K(\sqrt{c_2t^2+c_3\varepsilon_n^2}) + \frac{10\tau_0 x}{12n},$$

where ① follows from (145), and ② follows from (146). Using a similar argument for the proof of the first inequality in (139), we have

$$\widehat{\varepsilon}_n^2 \le r_G^* \le \max\left\{2C_u/c_\ell, 1\right\}^2 \left(c_2 + \frac{2c_3}{c_2}\right)\varepsilon_n^2 + \frac{10\tau_0 x}{12n},$$

and the second inequality in (140) is approved with $x = \Theta\left(n\varepsilon_n^2\right)$.

$\square$

**Lemma C.16.** Let $K$ be a PSD kernel, then with probability at least $1 - \exp\left(-n\varepsilon_n^2\right)$,

$$\|g\|_{L^2}^2 \le c_2\|g\|_n^2 + c_3\varepsilon_n^2, \quad \forall g \in \mathcal{H}_K(1). \tag{147}$$

Furthermore, with probability at least $1 - \exp\left(-n\varepsilon_n^2\right)$,

$$\|g\|_n^2 \le c_2\|g\|_{L^2}^2 + c_3\varepsilon_n^2, \quad \forall g \in \mathcal{H}_K(1). \tag{148}$$

Here $c_2, c_3$ are positive constants with $c_2 > 1$.

*Proof.* The results follow from Theorem A.1. $\square$

**Lemma C.17.** Suppose $\psi\colon [0, \infty) \to [0, \infty)$ is a sub-root function with the unique fixed point $r^*$. Then the following properties hold.

(1) Let $a \ge 0$, then $\psi(r) + a$ as a function of $r$ is also a sub-root function with fixed point $r_a^*$, and $r^* \le r_a^* \le r^* + 2a$.

(2) Let $b \ge 1$, $c \ge 0$ then $\psi(br + c)$ as a function of $r$ is also a sub-root function with fixed point $r_b^*$, and $r_b^* \le br^* + 2c/b$.

(3) Let $b \ge 1$, then $\psi_b(r) = b\psi(r)$ is also a sub-root function with fixed point $r_b^*$, and $r_b^* \le b^2 r^*$.

*Proof.* (1). Let $\psi_a(r) = \psi(r) + a$. It can be verified that $\psi_a(r)$ is a sub-root function because its nonnegative, nondecreasing and $\psi_a(r)/\sqrt{r}$ is nonincreasing. It follows from (Bartlett et al., 2005, Lemma 3.2) that $\psi_a$ has unique fixed point denoted by $r_a^*$. Because $r^* = \psi(r^*) \le \psi(r^*) + a = \psi_a(r^*)$, it follows from (Bartlett et al., 2005, Lemma 3.2) that $r^* \le r_a^*$. Furthermore, since

$$\psi_a(r^* + 2a) = \psi(r^* + 2a) + a \le \psi(r^*)\sqrt{\frac{r^* + 2a}{r^*}} + a \le \sqrt{r^*(r^* + 2a)} + a \le r^* + 2a,$$

it follows from (Bartlett et al., 2005, Lemma 3.2) again that $r_a^* \le r^* + 2a$.

(2). Let $\psi_b(r) = \psi(br + c)$. It can be verified that $\psi_b(r)$ a sub-root function by checking the definition. Also, we have $\psi(b(br^* + 2c/b) + c)/\sqrt{b(br^* + 2c/b) + c} \le \psi(r^*)/\sqrt{r^*}$. It follows that

$$\psi_b\left(br^* + \frac{2c}{b}\right) = \psi\left(b\left(br^* + \frac{2c}{b}\right) + c\right) \le b\sqrt{\left(r^* + \frac{3c}{b^2}\right)r^*}$$

$$\le b\left(r^* + \frac{3c}{2b^2}\right) \le br^* + \frac{2c}{b}.$$

Then it follows from (Bartlett et al., 2005, Lemma 3.2) that $r_b^* \le br^* + 2c/b$.

(3). Let $\psi_b(r) = b\psi(r)$. It can be verified that $\psi_b(r)$ a sub-root function by checking the definition. Also, we have $\psi(b^2 r^*)/\sqrt{b^2 r^*} \le \psi(r^*)/\sqrt{r^*}$, so $\psi(b^2 r^*) \le br^*$ and $\psi_b(b^2 r^*) = b\psi(b^2 r^*) \le b^2 r^*$. Then it follows from (Bartlett et al., 2005, Lemma 3.2) that $r_b^* \le b^2 r^*$. $\square$

### C.4 Proofs of Theorem C.1 and Theorem C.2

We need the following definition of $\varepsilon$-net for the proof of Theorem C.1 and Theorem C.2.

*Definition* C.1. ($\varepsilon$-net) Let $(X, d)$ be a metric space and let $\varepsilon > 0$. A subset $N_\varepsilon(X, d)$ is called an $\varepsilon$-net of $X$ if for every point $x \in X$, there exists some point $y \in N_\varepsilon(X, d)$ such that $d(x, y) \leq \varepsilon$. The minimal cardinality of an $\varepsilon$-net of $X$, if finite, is denoted by $N(X, d, \varepsilon)$ and is called the covering number of $X$ at scale $\varepsilon$.

**Proof of Theorem C.1.** First, we have $\mathbb{E}_{\mathbf{w} \sim \mathcal{N}(\mathbf{0}, \kappa^2 \mathbf{I}_d)}[h(\mathbf{w}, \mathbf{x}, \mathbf{y})] = K(\mathbf{x}, \mathbf{y})$. For any $\mathbf{x} \in \mathcal{X}$ and $s > 0$, define function class

$$\mathcal{H}_{\mathbf{x}, s} := \left\{ h(\cdot, \mathbf{x}', \mathbf{y}) \colon \mathbb{R}^d \to \mathbb{R} \colon \mathbf{x}' \in \mathbf{B}(\mathbf{x}; s) \cap \mathcal{X}, \mathbf{y} \in \mathcal{X} \right\}. \tag{149}$$

We first build an $s$-net for the unit sphere $\mathcal{X}$. By (Vershynin, 2012, Lemma 5.2), there exists an $s$-net $N_s(\mathcal{X}, \|\cdot\|_2)$ of $\mathcal{X}$ such that $N(\mathcal{X}, \|\cdot\|_2, s) \leq \left(1 + \frac{2}{s}\right)^d$.

In the sequel, a function in the class $\mathcal{H}_{\mathbf{x}, s}$ is also denoted as $h(\mathbf{w})$, omitting the presence of variables $\mathbf{x}'$ and $\mathbf{y}$ when no confusion arises. Let $P_m$ be the empirical distribution over $\left\{ \vec{\mathbf{w}}_r(0) \right\}$ so that $\mathbb{E}_{\mathbf{w} \sim P_m}[h(\mathbf{w})] = \widehat{h}(\mathbf{W}(0), \mathbf{x}, \mathbf{y})$. Given $\mathbf{x} \in N(\mathcal{X}, s)$, we aim to estimate the upper bound for the supremum of empirical process $\mathbb{E}_{\mathbf{w} \sim \mathcal{N}(\mathbf{0}, \kappa^2 \mathbf{I}_d)}[h(\mathbf{w})] - \mathbb{E}_{\mathbf{w} \sim P_m}[h(\mathbf{w})]$ when function $h$ ranges over the function class $\mathcal{H}_{\mathbf{x}, s}$. To this end, we apply Theorem A.1 to the function class $\mathcal{H}_{\mathbf{x}, s}$ with $\mathbf{W}(0) = \left\{ \vec{\mathbf{w}}_r(0) \right\}_{r=1}^m$. It can be verified that $h \in [0, 1]$ for any $h \in \mathcal{H}_{\mathbf{x}, s}$. It follows that we can set $a = 0, b = 1$ in Theorem A.1. With probability at least $1 - 2e^{-x}$ over the random initialization $\mathbf{W}(0)$,

$$\sup_{h \in \mathcal{H}_{\mathbf{x}, s}} \left| \mathbb{E}_{\mathbf{w} \sim \mathcal{N}(\mathbf{0}, \kappa^2 \mathbf{I}_d)}[h(\mathbf{w})] - \mathbb{E}_{\mathbf{w} \sim P_m}[h(\mathbf{w})] \right|$$

$$\leq \inf_{\alpha \in (0,1)} \left( 2(1 + \alpha) \mathbb{E}_{\mathbf{W}(0), \{\sigma_r\}_{r=1}^m} \left[ \sup_{h \in \mathcal{H}_{\mathbf{x}, s}} \frac{1}{m} \sum_{r=1}^m \sigma_r h(\vec{\mathbf{w}}_r(0)) \right] + \sqrt{\frac{2rx}{m}} + (b - a) \left( \frac{1}{3} + \frac{1}{\alpha} \right) \frac{x}{m} \right), \tag{150}$$

where $\{\sigma\}_{r=1}^m$ are i.i.d. Rademacher random variables taking values of $\pm 1$ with equal probability.

It can be verified that $\mathrm{Var}[h] \leq \mathbb{E}_{\mathbf{w}}[h(\mathbf{w}, \mathbf{x}', \mathbf{y})^2] \leq 1$. Setting $\alpha = \frac{1}{2}$ in (150), it follows that with probability at least $1 - \delta$,

$$\sup_{\mathbf{x}' \in \mathbf{B}(\mathbf{x}; s) \cap \mathcal{X}, \mathbf{y} \in \mathcal{X}} \left| K(\mathbf{x}', \mathbf{y}) - \widehat{h}(\mathbf{W}(0), \mathbf{x}', \mathbf{y}) \right| \leq 3\mathcal{R}(\mathcal{H}_{\mathbf{x}, s}) + \sqrt{\frac{2 \log \frac{2}{\delta}}{m}} + \frac{7 \log \frac{2}{\delta}}{3m}. \tag{151}$$

Here $\mathcal{R}(\mathcal{H}_{\mathbf{x}, s}) = \mathbb{E}_{\mathbf{W}(0), \{\sigma_r\}_{r=1}^m} \left[ \sup_{h \in \mathcal{H}_{\mathbf{x}, s}} \frac{1}{m} \sum_{r=1}^m \sigma_r h(\vec{\mathbf{w}}_r(0)) \right]$ is the Rademacher complexity of the function class $\mathcal{H}_{\mathbf{x}, s}$. By Lemma C.18, $\mathcal{R}(\mathcal{H}_{\mathbf{x}, s}) \leq \frac{1}{\sqrt{m}} + B\sqrt{ds}(s + 1) + \sqrt{s} + s$. Plugging such upper bound for $\mathcal{R}(\mathcal{H}_{\mathbf{x}, s})$ in (151), we have

$$\sup_{\mathbf{x}' \in \mathbf{B}(\mathbf{x}; s) \cap \mathcal{X}, \mathbf{y} \in \mathcal{X}} \left| K(\mathbf{x}', \mathbf{y}) - \widehat{h}(\mathbf{W}(0), \mathbf{x}', \mathbf{y}) \right|$$

$$\leq 3 \left( \frac{1}{\sqrt{m}} + B\sqrt{ds}(s + 1) + \sqrt{s} + s \right) + \sqrt{\frac{2 \log \frac{2}{\delta}}{m}} + \frac{7 \log \frac{2}{\delta}}{3m}. \tag{152}$$

Setting $s = \frac{1}{m}$, we have

$$\sup_{\mathbf{x}' \in \mathbf{B}(\mathbf{x};s) \cap \mathcal{X}, \mathbf{y} \in \mathcal{X}} \left| K(\mathbf{x}', \mathbf{y}) - \widehat{h}(\mathbf{W}(0), \mathbf{x}', \mathbf{y}) \right|$$

$$\leq 3 \left( \frac{1}{\sqrt{m}} + \frac{B\sqrt{d}\left(1 + \frac{1}{m}\right)}{\sqrt{m}} + \frac{1}{\sqrt{m}} + \frac{1}{m} \right) + \sqrt{\frac{2\log\frac{2}{\delta}}{m}} + \frac{7\log\frac{2}{\delta}}{3m}$$

$$\leq \frac{1}{\sqrt{m}} \left( 6(1 + B\sqrt{d}) + \sqrt{2\log\frac{2}{\delta}} \right) + \frac{1}{m} \left( 3 + \frac{7\log\frac{2}{\delta}}{3} \right). \tag{153}$$

By union bound, with probability at least $1 - (1 + 2m)^d \delta$ over $\mathbf{W}(0)$, (153) holds for arbitrary $\mathbf{x} \in N(\mathcal{X}, s)$. In this case, for any $\mathbf{x}' \in \mathcal{X}, \mathbf{y} \in \mathcal{X}$, there exists $\mathbf{x} \in N_s(\mathcal{X}, \|\cdot\|_2)$ such that $\|\mathbf{x}' - \mathbf{x}\|_2 \leq s$, so that $\mathbf{x}' \in \mathbf{B}(\mathbf{x}; s) \cap \mathcal{X}$, and (153) holds. Changing the notation $\mathbf{x}'$ to $\mathbf{x}$, the conclusion is proved.

$\square$

**Lemma C.18.** Let $\mathcal{R}(\mathcal{H}_{\mathbf{x},s}) := \mathbb{E}_{\mathbf{W}(0), \{\sigma_r\}_{r=1}^m} \left[ \sup_{h \in \mathcal{H}_{\mathbf{x},s}} \frac{1}{m} \sum_{r=1}^m \sigma_r h(\vec{\mathbf{w}}_r(0)) \right]$ be the Rademacher complexity of the function class $\mathcal{H}_{\mathbf{x},s}$, $B$ is a positive constant. Then

$$\mathcal{R}(\mathcal{H}_{\mathbf{x},s}) \leq \frac{1}{\sqrt{m}} + B\sqrt{ds}(s + 1) + \sqrt{s} + s. \tag{154}$$

*Proof.* We have

$$\mathcal{R}(\mathcal{H}_{\mathbf{x},s}) = \mathbb{E}_{\mathbf{W}(0), \{\sigma_r\}_{r=1}^m} \left[ \sup_{\mathbf{x}' \in \mathbf{B}(\mathbf{x};s), \mathbf{y} \in \mathcal{X}} \frac{1}{m} \sum_{r=1}^m \sigma_r h(\vec{\mathbf{w}}_r(0), \mathbf{x}', \mathbf{y}) \right]$$

$$\leq \mathcal{R}_1 + \mathcal{R}_2, \tag{155}$$

where

$$\mathcal{R}_1 = \mathbb{E}_{\mathbf{W}(0), \{\sigma_r\}_{r=1}^m} \left[ \sup_{\mathbf{x}' \in \mathbf{B}(\mathbf{x};s), \mathbf{y} \in \mathcal{X}} \frac{1}{m} \sum_{r=1}^m \sigma_r h(\vec{\mathbf{w}}_r(0), \mathbf{x}, \mathbf{y}) \right],$$

$$\mathcal{R}_2 = \mathbb{E}_{\mathbf{W}(0), \{\sigma_r\}_{r=1}^m} \left[ \sup_{\mathbf{x}' \in \mathbf{B}(\mathbf{x};s), \mathbf{y} \in \mathcal{X}} \frac{1}{m} \sum_{r=1}^m \sigma_r \left( h(\vec{\mathbf{w}}_r(0), \mathbf{x}', \mathbf{y}) - h(\vec{\mathbf{w}}_r(0), \mathbf{x}, \mathbf{y}) \right) \right]. \tag{156}$$

Here (155) follows from the subadditivity of superemum and the fact that $\sum_{r=1}^m \sigma_r h(\vec{\mathbf{w}}_r(0), \mathbf{x}', \mathbf{y}) = \sum_{r=1}^m \sigma_r h(\vec{\mathbf{w}}_r(0), \mathbf{x}, \mathbf{y}) + \sum_{r=1}^m \sigma_r \left( h(\vec{\mathbf{w}}_r(0), \mathbf{x}', \mathbf{y}) - h(\vec{\mathbf{w}}_r(0), \mathbf{x}, \mathbf{y}) \right)$.

Now we bound $\mathcal{R}_1$ and $\mathcal{R}_2$ separately. For $\mathcal{R}_1$, we have

$$\mathcal{R}_1 = \mathbb{E}_{\mathbf{W}(0), \{\sigma_r\}_{r=1}^m} \left[ \sup_{\mathbf{x}' \in \mathbf{B}(\mathbf{x};s), \mathbf{y} \in \mathcal{X}} \frac{1}{m} \sum_{r=1}^m \sigma_r h(\vec{\mathbf{w}}_r(0), \mathbf{x}, \mathbf{y}) \right]$$

$$\overset{\textcircled{1}}{=} \mathbb{E}_{\mathbf{W}(0)} \left[ \mathbb{E}_{\{\sigma_r\}_{r=1}^m} \left[ \sup_{\mathbf{y} \in \mathcal{X}} \frac{1}{m} \sum_{r=1}^m \sigma_r \mathbf{x}^\top \mathbf{y} \mathbb{1}_{\left\{ \vec{\mathbf{w}}_r(0)^\top \mathbf{x} \geq 0 \right\}} \mathbb{1}_{\left\{ \vec{\mathbf{w}}_r(0)^\top \mathbf{y} \geq 0 \right\}} \right] \right]$$

$$= \mathbb{E}_{\mathbf{W}(0)} \left[ \mathbb{E}_{\{\sigma_r\}_{r=1}^m} \left[ \sup_{\mathbf{y} \in \mathcal{X}} \frac{1}{m} \mathbf{y}^\top \mathbb{1}_{\left\{ \vec{\mathbf{w}}_r(0)^\top \mathbf{y} \geq 0 \right\}} \left( \sum_{r=1}^m \sigma_r \mathbf{x} \mathbb{1}_{\left\{ \vec{\mathbf{w}}_r(0)^\top \mathbf{x} \geq 0 \right\}} \right) \right] \right]$$

$$\overset{\textcircled{2}}{\leq} \mathbb{E}_{\mathbf{W}(0)} \left[ \mathbb{E}_{\{\sigma_r\}_{r=1}^m} \left[ \sup_{\mathbf{y} \in \mathcal{X}} \frac{1}{m} \|\mathbf{y}\|_2 \left| \mathbb{1}_{\left\{ \vec{\mathbf{w}}_r(0)^\top \mathbf{y} \geq 0 \right\}} \right| \left\| \sum_{r=1}^m \sigma_r \mathbf{x} \mathbb{1}_{\left\{ \vec{\mathbf{w}}_r(0)^\top \mathbf{x} \geq 0 \right\}} \right\|_2 \right] \right]$$

$$
\overset{③}{\leq} \mathbb{E}_{\mathbf{W}(0)} \left[ \mathbb{E}_{\{\sigma_r\}_{r=1}^m} \left[ \frac{1}{m} \left\| \sum_{r=1}^m \sigma_r \mathbf{x} \mathbb{1}_{\left\{ \overrightarrow{\mathbf{w}}_r(0)^\top \mathbf{x} \geq 0 \right\}} \right\|_2 \right] \right]
$$

$$
\overset{④}{=} \mathbb{E}_{\mathbf{W}(0)} \left[ \mathbb{E}_{\{\sigma_r\}_{r=1}^m} \left[ \frac{1}{m} \sqrt{ \left( \sum_{r=1}^m \sigma_r \mathbb{1}_{\left\{ \overrightarrow{\mathbf{w}}_r(0)^\top \mathbf{x} \geq 0 \right\}} \right)^2 } \right] \right]
$$

$$
\overset{⑤}{\leq} \mathbb{E}_{\mathbf{W}(0)} \left[ \frac{1}{m} \sqrt{ \mathbb{E}_{\{\sigma_r\}_{r=1}^m} \left[ \left( \sum_{r=1}^m \sigma_r \mathbb{1}_{\left\{ \overrightarrow{\mathbf{w}}_r(0)^\top \mathbf{x} \geq 0 \right\}} \right)^2 \right] } \right]
$$

$$
= \mathbb{E}_{\mathbf{W}(0)} \left[ \frac{1}{m} \sqrt{ \mathbb{E}_{\{\sigma_r\}_{r=1}^m} \left[ \left( \sum_{r \in [m], r' \in [m]} \sigma_r \sigma_{r'} \mathbb{1}_{\left\{ \overrightarrow{\mathbf{w}}_r(0)^\top \mathbf{x} \geq 0 \right\}} \mathbb{1}_{\left\{ \overrightarrow{\mathbf{w}}_{r'}(0)^\top \mathbf{x} \geq 0 \right\}} \right) \right] } \right]
$$

$$
\overset{⑥}{\leq} \mathbb{E}_{\mathbf{W}(0)} \left[ \frac{1}{m} \cdot \sqrt{m} \right] = \frac{1}{\sqrt{m}}. \tag{157}
$$

In (157), ① is due to the fact that the operand of the supremum operator does not depend on $\mathbf{x}'$ and the Fubini Theorem. ② follows from the Cauchy-Schwarz inequality. ③ is due to the fact that $\|\mathbf{y}\|_2 = 1$, $\mathbb{1}_{\left\{ \overrightarrow{\mathbf{w}}_r(0)^\top \mathbf{y} \geq 0 \right\}} \in \{0, 1\}$. ④ follows from $\|\mathbf{x}\|_2 = 1$, and ⑤ is due to the Jensen's inequality. ⑥ follows from the property of Rademacher variable, that is, $\mathbb{E}_{\{\sigma_r\}_{r=1}^m} [\sigma_r \sigma_{r'}] = \mathbb{1}_{\{r=r'\}}$, and the fact that $\mathbb{1}_{\left\{ \overrightarrow{\mathbf{w}}_r(0)^\top \mathbf{x} \geq 0 \right\}} \mathbb{1}_{\left\{ \overrightarrow{\mathbf{w}}_{r'}(0)^\top \mathbf{x} \geq 0 \right\}} \in \{0, 1\}$.

For $\mathcal{R}_2$, we first define

$$
Q := \frac{1}{m} \sum_{r=1}^m \mathbb{1}_{\left\{ \mathbb{1}_{\left\{ \mathbf{x}'^\top \overrightarrow{\mathbf{w}}_r(0) \geq 0 \right\}} \neq \mathbb{1}_{\left\{ \mathbf{x}^\top \overrightarrow{\mathbf{w}}_r(0) \geq 0 \right\}} \right\}},
$$

which is the average number of weights in $\mathbf{W}(0)$ whose inner products with $\mathbf{x}$ and $\mathbf{x}'$ have different signs. Our observation is that, if $\left| \mathbf{x}^\top \overrightarrow{\mathbf{w}}_r(0) \right| > s \left\| \overrightarrow{\mathbf{w}}_r(0) \right\|_2$, then $\mathbf{x}^\top \overrightarrow{\mathbf{w}}_r(0)$ has the same sign as $\mathbf{x}'^\top \overrightarrow{\mathbf{w}}_r(0)$. To see this, by the Cauchy-Schwarz inequality,

$$
\left| \mathbf{x}'^\top \overrightarrow{\mathbf{w}}_r(0) - \mathbf{x}^\top \overrightarrow{\mathbf{w}}_r(0) \right| \leq \|\mathbf{x}' - \mathbf{x}\|_2 \left\| \overrightarrow{\mathbf{w}}_r(0) \right\|_2 \leq s \left\| \overrightarrow{\mathbf{w}}_r(0) \right\|_2, \tag{158}
$$

then we have $\mathbf{x}^\top \overrightarrow{\mathbf{w}}_r(0) > s \left\| \overrightarrow{\mathbf{w}}_r(0) \right\|_2 \Rightarrow \mathbf{x}'^\top \overrightarrow{\mathbf{w}}_r(0) \geq \mathbf{x}^\top \overrightarrow{\mathbf{w}}_r(0) - s \left\| \overrightarrow{\mathbf{w}}_r(0) \right\|_2 > 0$, and $\mathbf{x}^\top \overrightarrow{\mathbf{w}}_r(0) < -s \left\| \overrightarrow{\mathbf{w}}_r(0) \right\|_2 \Rightarrow \mathbf{x}'^\top \overrightarrow{\mathbf{w}}_r(0) \leq \mathbf{x}^\top \overrightarrow{\mathbf{w}}_r(0) + s \left\| \overrightarrow{\mathbf{w}}_r(0) \right\|_2 < 0$.

As a result, $Q \leq \frac{1}{m} \sum_{r=1}^m \mathbb{1}_{\left\{ \left| \mathbf{x}^\top \overrightarrow{\mathbf{w}}_r(0) \right| \leq s \left\| \overrightarrow{\mathbf{w}}_r(0) \right\|_2 \right\}}$, and it follows that

$$
\mathbb{E}_{\mathbf{W}(0)} [Q] \leq \mathbb{E}_{\mathbf{W}(0)} \left[ \frac{1}{m} \sum_{r=1}^m \mathbb{1}_{\left\{ \left| \mathbf{x}^\top \overrightarrow{\mathbf{w}}_r(0) \right| \leq s \left\| \overrightarrow{\mathbf{w}}_r(0) \right\|_2 \right\}} \right] = \Pr \left[ \left| \mathbf{x}^\top \overrightarrow{\mathbf{w}}_r(0) \right| \leq s \left\| \overrightarrow{\mathbf{w}}_r(0) \right\|_2 \right]
$$

$$
= \Pr \left[ \frac{\left| \mathbf{x}^\top \overrightarrow{\mathbf{w}}_r(0) \right|}{\left\| \overrightarrow{\mathbf{w}}_r(0) \right\|_2} \leq s \right], \tag{159}
$$

where the last equality holds because each $\overrightarrow{\mathbf{w}}_r(0), r \in [m]$, follows a continuous Gaussian distribution. By Lemma C.19, $\Pr \left[ \frac{\left| \mathbf{x}^\top \overrightarrow{\mathbf{w}}_r(0) \right|}{\left\| \overrightarrow{\mathbf{w}}_r(0) \right\|_2} \leq s \right] \leq B\sqrt{d}s$ for an absolute positive constant $B$. According to this inequality and (159), it follows that

$$\mathbb{E}_{\mathbf{W}(0)}[Q] \le B\sqrt{d}s. \tag{160}$$

By Markov's inequality, we have

$$\Pr\left[Q \ge \sqrt{s}\right] \le B\sqrt{d}s, \tag{161}$$

where the probability is with respect to the probability measure space of $\mathbf{W}(0)$. Let $A$ be the event that $Q \ge \sqrt{s}$. We denote by $\Omega_s$ the subset of the probability measure space of $\mathbf{W}(0)$ such that $A$ happens, then $\Pr\left[\Omega_s\right] \le B\sqrt{d}s$. Now we aim to bound $\mathcal{R}_2$ by estimating its bound on $\Omega_s$ and its complement. First, we have

$$
\begin{aligned}
\mathcal{R}_2 &= \mathbb{E}_{\mathbf{W}(0),\{\sigma_r\}_{r=1}^m}\left[\sup_{\mathbf{x}'\in\mathbf{B}(\mathbf{x};s),\mathbf{y}\in\mathcal{X}}\frac{1}{m}\sum_{r=1}^m \sigma_r\Big(h(\vec{\mathbf{w}}_r(0),\mathbf{x}',\mathbf{y}) - h(\vec{\mathbf{w}}_r(0),\mathbf{x},\mathbf{y})\Big)\right] \\
&= \underbrace{\mathbb{E}_{\mathbf{W}(0)\in\Omega_s,\{\sigma_r\}_{r=1}^m}\left[\sup_{\mathbf{x}'\in\mathbf{B}(\mathbf{x};s),\mathbf{y}\in\mathcal{X}}\frac{1}{m}\sum_{r=1}^m \sigma_r\Big(h(\vec{\mathbf{w}}_r(0),\mathbf{x}',\mathbf{y}) - h(\vec{\mathbf{w}}_r(0),\mathbf{x},\mathbf{y})\Big)\right]}_{\mathcal{R}_{21}} \\
&\quad + \underbrace{\mathbb{E}_{\mathbf{W}(0)\notin\Omega_s,\{\sigma_r\}_{r=1}^m}\left[\sup_{\mathbf{x}'\in\mathbf{B}(\mathbf{x};s),\mathbf{y}\in\mathcal{X}}\frac{1}{m}\sum_{r=1}^m \sigma_r\Big(h(\vec{\mathbf{w}}_r(0),\mathbf{x}',\mathbf{y}) - h(\vec{\mathbf{w}}_r(0),\mathbf{x},\mathbf{y})\Big)\right]}_{\mathcal{R}_{22}},
\end{aligned}
\tag{162}
$$

where we used the convention that $\mathbb{E}_{\mathbf{W}(0)\in A}[\cdot] = \mathbb{E}_{\mathbf{W}(0)}\left[\mathbb{1}_{\{\mathbf{W}(0)\in A\}} \times \cdot\right]$. Now we estimate the upper bound for $\mathcal{R}_{21}$ and $\mathcal{R}_{22}$ separately. Let $I = \left\{r \in [m]\colon \mathbb{1}_{\left\{\mathbf{x}'^\top\vec{\mathbf{w}}_r(0)\ge 0\right\}} \ne \mathbb{1}_{\left\{\mathbf{x}^\top\vec{\mathbf{w}}_r(0)\ge 0\right\}}\right\}$. When $\mathbf{W}(0) \notin \Omega_s$, we have $Q < \sqrt{s}$. In this case, it follows that $|I| \le m\sqrt{s}$. Moreover, when $r \in I$, either $\mathbb{1}_{\left\{\mathbf{x}'^\top\vec{\mathbf{w}}_r(0)\ge 0\right\}} = 0$ or $\mathbb{1}_{\left\{\mathbf{x}^\top\vec{\mathbf{w}}_r(0)\ge 0\right\}} = 0$. As a result,

$$
\begin{aligned}
&\left|h(\vec{\mathbf{w}}_r(0),\mathbf{x}',\mathbf{y}) - h(\vec{\mathbf{w}}_r(0),\mathbf{x},\mathbf{y})\right| \\
&= \left|\mathbf{x}'^\top\mathbf{y}\,\mathbb{1}_{\left\{\mathbf{x}'^\top\vec{\mathbf{w}}_r(0)\ge 0\right\}}\mathbb{1}_{\left\{\mathbf{y}^\top\vec{\mathbf{w}}_r(0)\ge 0\right\}} - \mathbf{x}^\top\mathbf{y}\,\mathbb{1}_{\left\{\mathbf{x}^\top\vec{\mathbf{w}}_r(0)\ge 0\right\}}\mathbb{1}_{\left\{\mathbf{y}^\top\vec{\mathbf{w}}_r(0)\ge 0\right\}}\right| \\
&\le \max\left\{\mathbf{x}'^\top\mathbf{y}\,\mathbb{1}_{\left\{\mathbf{x}'^\top\vec{\mathbf{w}}_r(0)\ge 0\right\}}\mathbb{1}_{\left\{\mathbf{y}^\top\vec{\mathbf{w}}_r(0)\ge 0\right\}}, \mathbf{x}^\top\mathbf{y}\,\mathbb{1}_{\left\{\mathbf{x}^\top\vec{\mathbf{w}}_r(0)\ge 0\right\}}\mathbb{1}_{\left\{\mathbf{y}^\top\vec{\mathbf{w}}_r(0)\ge 0\right\}}\right\} \\
&\le \max\left\{\mathbf{x}'^\top\mathbf{y}, \mathbf{x}^\top\mathbf{y}\right\} \le 1. \tag{163}
\end{aligned}
$$

When $r \in [m] \setminus I$, we have

$$
\begin{aligned}
&\left|h(\vec{\mathbf{w}}_r(0),\mathbf{x}',\mathbf{y}) - h(\vec{\mathbf{w}}_r(0),\mathbf{x},\mathbf{y})\right| \\
&= \left|\mathbf{x}'^\top\mathbf{y}\,\mathbb{1}_{\left\{\mathbf{x}'^\top\vec{\mathbf{w}}_r(0)\ge 0\right\}}\mathbb{1}_{\left\{\mathbf{y}^\top\vec{\mathbf{w}}_r(0)\ge 0\right\}} - \mathbf{x}^\top\mathbf{y}\,\mathbb{1}_{\left\{\mathbf{x}^\top\vec{\mathbf{w}}_r(0)\ge 0\right\}}\mathbb{1}_{\left\{\mathbf{y}^\top\vec{\mathbf{w}}_r(0)\ge 0\right\}}\right| \\
&= \left|\left(\mathbf{x}'\mathbb{1}_{\left\{\mathbf{x}'^\top\vec{\mathbf{w}}_r(0)\ge 0\right\}} - \mathbf{x}\mathbb{1}_{\left\{\mathbf{x}^\top\vec{\mathbf{w}}_r(0)\ge 0\right\}}\right)^\top \mathbf{y}\,\mathbb{1}_{\left\{\mathbf{y}^\top\vec{\mathbf{w}}_r(0)\ge 0\right\}}\right| \\
&\overset{①}{\le} \left\|\mathbf{x}'\mathbb{1}_{\left\{\mathbf{x}'^\top\vec{\mathbf{w}}_r(0)\ge 0\right\}} - \mathbf{x}\mathbb{1}_{\left\{\mathbf{x}^\top\vec{\mathbf{w}}_r(0)\ge 0\right\}}\right\|_2 \|\mathbf{y}\|_2 \left|\mathbb{1}_{\left\{\mathbf{y}^\top\vec{\mathbf{w}}_r(0)\ge 0\right\}}\right| \\
&\overset{②}{\le} \left\|\mathbf{x}'\mathbb{1}_{\left\{\mathbf{x}'^\top\vec{\mathbf{w}}_r(0)\ge 0\right\}} - \mathbf{x}\mathbb{1}_{\left\{\mathbf{x}'^\top\vec{\mathbf{w}}_r(0)\ge 0\right\}} + \mathbf{x}\mathbb{1}_{\left\{\mathbf{x}'^\top\vec{\mathbf{w}}_r(0)\ge 0\right\}} - \mathbf{x}\mathbb{1}_{\left\{\mathbf{x}^\top\vec{\mathbf{w}}_r(0)\ge 0\right\}}\right\|_2
\end{aligned}
$$

$$\leq \|\mathbf{x}' - \mathbf{x}\|_2 \left| \mathbb{I}_{\left\{\mathbf{x}'^\top \vec{\mathbf{w}}_r(0) \geq 0\right\}} \right| + \|\mathbf{x}\|_2 \left| \mathbb{I}_{\left\{\mathbf{x}'^\top \vec{\mathbf{w}}_r(0) \geq 0\right\}} - \mathbb{I}_{\left\{\mathbf{x}^\top \vec{\mathbf{w}}_r(0) \geq 0\right\}} \right|$$

$$\overset{\text{③}}{\leq} s, \tag{164}$$

where ① follows from the Cauchy-Schwarz inequality, ② is due to the fact that $\left| \mathbb{I}_{\left\{\mathbf{y}^\top \vec{\mathbf{w}}_r(0) \geq 0\right\}} \right| \in \{0,1\}$ and $\|\mathbf{y}\|_2 = 1$. ③ follows from $\mathbf{x}' \in \mathbf{B}(\mathbf{x};s)$, $\left| \mathbb{I}_{\left\{\mathbf{x}'^\top \vec{\mathbf{w}}_r(0) \geq 0\right\}} \right| \in \{0,1\}$, and $\mathbb{I}_{\left\{\mathbf{x}'^\top \vec{\mathbf{w}}_r(0) \geq 0\right\}} = \mathbb{I}_{\left\{\mathbf{x}^\top \vec{\mathbf{w}}_r(0) \geq 0\right\}}$ because $r \notin I$.

By (163) and (164), we have

$$\frac{1}{m} \sum_{r=1}^m \sigma_r \left( h(\vec{\mathbf{w}}_r(0), \mathbf{x}', \mathbf{y}) - h(\vec{\mathbf{w}}_r(0), \mathbf{x}, \mathbf{y}) \right)$$

$$= \frac{1}{m} \sum_{r \in I} \sigma_r \left( h(\vec{\mathbf{w}}_r(0), \mathbf{x}', \mathbf{y}) - h(\vec{\mathbf{w}}_r(0), \mathbf{x}, \mathbf{y}) \right) + \frac{1}{m} \sum_{r \in [m] \setminus I} \sigma_r \left( h(\vec{\mathbf{w}}_r(0), \mathbf{x}', \mathbf{y}) - h(\vec{\mathbf{w}}_r(0), \mathbf{x}, \mathbf{y}) \right)$$

$$\leq \frac{1}{m} \sum_{r \in I} \left| h(\vec{\mathbf{w}}_r(0), \mathbf{x}', \mathbf{y}) - h(\vec{\mathbf{w}}_r(0), \mathbf{x}, \mathbf{y}) \right| + \frac{1}{m} \sum_{r \in [m] \setminus I} \left| h(\vec{\mathbf{w}}_r(0), \mathbf{x}', \mathbf{y}) - h(\vec{\mathbf{w}}_r(0), \mathbf{x}, \mathbf{y}) \right|$$

$$\overset{\text{①}}{\leq} \frac{m\sqrt{s}}{m} + \frac{m - m\sqrt{s}}{m} s \leq \sqrt{s} + s, \tag{165}$$

where ① uses the bounds in (163) and (164).

Using (165), we now estimate the upper bound for $\mathcal{R}_{22}$ by

$$\mathcal{R}_{22} = \mathbb{E}_{\mathbf{W}(0) \notin \Omega_s, \{\sigma_r\}_{r=1}^m} \left[ \sup_{\mathbf{x}' \in \mathbf{B}(\mathbf{x};s), \mathbf{y} \in \mathcal{X}} \frac{1}{m} \sum_{r=1}^m \sigma_r \left( h(\vec{\mathbf{w}}_r(0), \mathbf{x}', \mathbf{y}) - h(\vec{\mathbf{w}}_r(0), \mathbf{x}, \mathbf{y}) \right) \right]$$

$$\leq \mathbb{E}_{\mathbf{W}(0) \notin \Omega_s, \{\sigma_r\}_{r=1}^m} \left[ \sqrt{s} + s \right] \leq \sqrt{s} + s. \tag{166}$$

When $\mathbf{W}(0) \in \Omega_s$, by the second last inequality of (164), we have

$$\left| h(\vec{\mathbf{w}}_r(0), \mathbf{x}', \mathbf{y}) - h(\vec{\mathbf{w}}_r(0), \mathbf{x}, \mathbf{y}) \right|$$

$$\leq \|\mathbf{x}' - \mathbf{x}\|_2 \left| \mathbb{I}_{\left\{\mathbf{x}'^\top \vec{\mathbf{w}}_r(0) \geq 0\right\}} \right| + \|\mathbf{x}\|_2 \left| \mathbb{I}_{\left\{\mathbf{x}'^\top \vec{\mathbf{w}}_r(0) \geq 0\right\}} - \mathbb{I}_{\left\{\mathbf{x}^\top \vec{\mathbf{w}}_r(0) \geq 0\right\}} \right| \leq s + 1. \tag{167}$$

According to (167), for $\mathcal{R}_{21}$, we have

$$\mathcal{R}_{21} = \mathbb{E}_{\mathbf{W}(0) \in \Omega_s, \{\sigma_r\}_{r=1}^m} \left[ \sup_{\mathbf{x}' \in \mathbf{B}(\mathbf{x};s), \mathbf{y} \in \mathcal{X}} \frac{1}{m} \sum_{r=1}^m \sigma_r \left( h(\vec{\mathbf{w}}_r(0), \mathbf{x}', \mathbf{y}) - h(\vec{\mathbf{w}}_r(0), \mathbf{x}, \mathbf{y}) \right) \right]$$

$$\leq \mathbb{E}_{\mathbf{W}(0) \in \Omega_s, \{\sigma_r\}_{r=1}^m} \left[ \sup_{\mathbf{x}' \in \mathbf{B}(\mathbf{x};s), \mathbf{y} \in \mathcal{X}} \frac{1}{m} \sum_{r=1}^m \left| \sigma_r \left( h(\vec{\mathbf{w}}_r(0), \mathbf{x}', \mathbf{y}) - h(\vec{\mathbf{w}}_r(0), \mathbf{x}, \mathbf{y}) \right) \right| \right]$$

$$\overset{\text{①}}{\leq} \mathbb{E}_{\mathbf{W}(0) \in \Omega_s, \{\sigma_r\}_{r=1}^m} \left[ s + 1 \right] = (s+1) \Pr\left[\Omega_s\right] \leq B\sqrt{ds}(s+1) \tag{168}$$

Combining (162), (166), and (168), we have the upper bound for $\mathcal{R}_2$ as

$$\mathcal{R}_2 = \mathcal{R}_{21} + \mathcal{R}_{22} \leq B\sqrt{ds}(s+1) + \sqrt{s} + s. \tag{169}$$

Plugging (157) and (169) in (155), we have

$$\mathcal{R}(\mathcal{H}_{\mathbf{x},s}) \leq \mathcal{R}_1 + \mathcal{R}_2 \leq \frac{1}{\sqrt{m}} + B\sqrt{ds}(s+1) + \sqrt{s} + s. \tag{170}$$

$\square$

**Lemma C.19.** Let $\mathbf{w} \sim \mathcal{N}(\mathbf{0}, \kappa^2 \mathbf{I}_d)$ with $\kappa > 0$. Then for any $\varepsilon \in (0,1)$ and $\mathbf{x} \in \mathcal{X}$, $\Pr\left[\frac{|\mathbf{x}^\top \mathbf{w}|}{\|\mathbf{w}\|_2} \leq \varepsilon\right] \leq B\sqrt{d}\varepsilon$ where $B$ is an absolute positive constant.

**Remark.** In fact, $B$ can be set to $2(2\pi)^{-1/2}$ when $d \to \infty$.

*Proof.* Let $z = \frac{\mathbf{x}^\top \mathbf{w}}{\|\mathbf{w}\|_2}$. It can be verified that $z^2 \sim z_1$ where $z_1$ is a random variable following the Beta distribution $\mathrm{Beta}(\frac{1}{2}, \frac{d-1}{2})$. Therefore, the distribution of $z$ has the following continuous probability density function $p_z$ with respect to the Lebesgue measure,

$$p_z(x) = (1 - x^2)^{\frac{d-3}{2}} \mathbb{I}_{\{|x| \leq 1\}} / B', \tag{171}$$

where $B' = \int_{-1}^1 (1 - x^2)^{\frac{d-3}{2}} \mathrm{d}x$ is the normalization factor. It can be verified by standard calculation that $1/B' \leq \frac{B\sqrt{d}}{2}$ for an absolute positive constant $B$.

Because $1 - x^2 \leq 1$ over $x \in [-1, 1]$, we have $B' \leq 1$. In addition,

$$\Pr\left[\frac{|\mathbf{x}^\top \mathbf{w}|}{\|\mathbf{w}\|_2} \leq \varepsilon\right] = \Pr\left[-\varepsilon \leq z \leq \varepsilon\right] = \frac{1}{B'} \int_{-\varepsilon}^{\varepsilon} (1 - x^2)^{\frac{d-3}{2}} \mathrm{d}x \leq B\sqrt{d}\varepsilon, \tag{172}$$

where the last inequality is due to the fact that $1 - x^2 \leq 1$ for $x \in [-\varepsilon, \varepsilon]$ with $\varepsilon \in (0,1)$. $\square$

**Proof of Theorem C.2.** We follow the same proof strategy as that for Theorem C.1.

First, we have $\mathbb{E}_{\mathbf{w} \sim \mathcal{N}(\mathbf{0}, \kappa^2 \mathbf{I}_d)}[v_R(\mathbf{w}, \mathbf{x})] = \Pr\left[|\mathbf{w}^\top \mathbf{x}| \leq R\right]$. For any $\mathbf{x} \in \mathcal{X}$ and $s > 0$, define function class

$$\mathcal{V}_{\mathbf{x},s} \coloneqq \left\{ v_R(\cdot, \mathbf{x}') \colon \mathbb{R}^d \to \mathbb{R} \colon \mathbf{x}' \in \mathbf{B}(\mathbf{x}; s) \cap \in \mathcal{X} \right\}. \tag{173}$$

We first build an $s$-net for the unit sphere $\mathcal{X}$. By (Vershynin, 2012, Lemma 5.2), there exists an $s$-net $N_s(\mathcal{X}, \|\cdot\|_2)$ of $\mathcal{X}$ such that $N(\mathcal{X}, \|\cdot\|_2, s) \leq \left(1 + \frac{2}{s}\right)^d$.

In the sequel, a function in the class $\mathcal{V}_{\mathbf{x}}$ is also denoted as $v_R(\mathbf{w})$, omitting the presence of $\mathbf{x}$ when no confusion arises. Let $P_m$ be the empirical distribution over $\left\{\vec{\mathbf{w}}_r(0)\right\}$ and $\mathbb{E}_{\mathbf{w} \sim P_m}[v_R(\mathbf{w})] = \widehat{v}_R(\mathbf{W}(0), \mathbf{x})$.

Given $\mathbf{x} \in N_s(\mathcal{X}, \|\cdot\|_2)$, we aim to estimate the upper bound for the supremum of empirical process $\mathbb{E}_{\mathbf{w} \sim \mathcal{N}(\mathbf{0}, \kappa^2 \mathbf{I}_d)}[v_R(\mathbf{w})] - \mathbb{E}_{\mathbf{w} \sim P_m}[v_R(\mathbf{w})]$ when function $v_R$ ranges over the function class $\mathcal{V}_{\mathbf{x},s}$. To this end, we apply Theorem A.1 to the function class $\mathcal{V}_{\mathbf{x},s}$ with $\mathbf{W}(0) = \left\{\vec{\mathbf{w}}_r(0)\right\}_{r=1}^m$. It can be verified that $v_R \in [0, 1]$ for any $v_R \in \mathcal{V}_{\mathbf{x},s}$. It follows that we can set $a = 0, b = 1$ in Theorem A.1. Setting $\alpha = \frac{1}{2}$ in Theorem A.1, then with probability at least $1 - 2e^{-x}$ over the random initialization $\mathbf{W}(0)$,

$$\sup_{v_R \in \mathcal{V}_{\mathbf{x},s}} \left|\mathbb{E}_{\mathbf{w} \sim \mathcal{N}(\mathbf{0}, \kappa^2 \mathbf{I}_d)}[v_R(\mathbf{w})] - \mathbb{E}_{\mathbf{w} \sim P_m}[v_R(\mathbf{w})]\right|$$

$$\leq \inf_{\alpha \in (0,1)} \left(3\mathbb{E}_{\mathbf{W}(0),\{\sigma_r\}_{r=1}^m}\left[\sup_{v_R \in \mathcal{V}_{\mathbf{x},s}} \frac{1}{m} \sum_{r=1}^m \sigma_r v_R(\vec{\mathbf{w}}_r(0))\right] + \sqrt{\frac{2rx}{m}} + \frac{7(b-a)x}{3m}\right), \tag{174}$$

where $\{\sigma\}_{r=1}^m$ are i.i.d. Rademacher random variables taking values of $\pm 1$ with equal probability.

It can be verified that $\mathrm{Var}\,[v_R] \leq \mathbb{E}_{\mathbf{w}}\left[v_R(\mathbf{w}, \mathbf{x})^2\right] \leq 1$, so $r$ can be set to 1. It follows that with probability at least $1 - \delta$,

$$\sup_{\mathbf{x}' \in \mathbf{B}(\mathbf{x};s) \cap \mathcal{X}} \left|\widehat{v}_R(\mathbf{W}(0), \mathbf{x}') - \Pr\left[\left|\mathbf{w}^\top \mathbf{x}'\right| \leq R\right]\right| \leq 3\mathcal{R}(\mathcal{H}_{\mathbf{x},s}) + \sqrt{\frac{2\log \frac{2}{\delta}}{m}} + \frac{7\log \frac{2}{\delta}}{3m}. \tag{175}$$

Here $\mathcal{R}(\mathcal{V}_{\mathbf{x},s}) = \mathbb{E}_{\mathbf{W}(0),\{\sigma_r\}_{r=1}^m}\left[\sup_{v_R \in \mathcal{V}_{\mathbf{x},s}} \frac{1}{m} \sum_{r=1}^m \sigma_r v_R(\vec{\mathbf{w}}_r(0))\right]$ is the Rademacher complexity

of the function class $\mathcal{V}_{\mathbf{x},s}$. By Lemma C.21, $\mathcal{R}(\mathcal{V}_{\mathbf{x},s}) \leq (B\sqrt{d}+1)\sqrt{m^{-\frac{1}{2}}+s} + \frac{\exp\left(-\frac{(\kappa^2-R_0^2)^2}{4\kappa^4}m\right)}{\sqrt{m^{-\frac{1}{2}}+s}}$.

Plugging such upper bound for $\mathcal{R}(\mathcal{V}_{\mathbf{x},s})$ in (175), we have

$$\sup_{\mathbf{x}' \in \mathbf{B}(\mathbf{x};s) \cap \mathcal{X}} \left|\widehat{v}_R(\mathbf{W}(0), \mathbf{x}') - \Pr\left[\left|\mathbf{w}^\top \mathbf{x}'\right| \leq R\right]\right|$$
$$\leq 3\left((B\sqrt{d}+1)\sqrt{m^{-\frac{1}{2}}+s} + \frac{\exp\left(-\frac{(\kappa^2-R_0^2)^2}{4\kappa^4}m\right)}{\sqrt{m^{-\frac{1}{2}}+s}}\right) + \sqrt{\frac{2\log \frac{2}{\delta}}{m}} + \frac{7\log \frac{2}{\delta}}{3m}. \tag{176}$$

Setting $s = \frac{1}{m}$, we have

$$\sup_{\mathbf{x}' \in \mathbf{B}(\mathbf{x};s) \cap \mathcal{X}} \left|\widehat{v}_R(\mathbf{W}(0), \mathbf{x}') - \Pr\left[\left|\mathbf{w}^\top \mathbf{x}'\right| \leq R\right]\right|$$
$$\leq 3\left((B\sqrt{d}+1)\sqrt{m^{-\frac{1}{2}}+\frac{1}{m}} + \frac{\exp\left(-\frac{(\kappa^2-R_0^2)^2}{4\kappa^4}m\right)}{\sqrt{m^{-\frac{1}{2}}+\frac{1}{m}}}\right) + \sqrt{\frac{2\log \frac{2}{\delta}}{m}} + \frac{7\log \frac{2}{\delta}}{3m}$$

$$\tag{177}$$

By union bound, with probability at least $1 - (1+2m)^d \delta$ over $\mathbf{W}(0)$, (177) holds for arbitrary $\mathbf{x} \in N(\mathcal{X}, s)$. In this case, for any $\mathbf{x}' \in \mathcal{X}$, there exists $\mathbf{x} \in N(\mathcal{X}, s)$ such that $\|\mathbf{x}' - \mathbf{x}\|_2 \leq s$, so that $\mathbf{x}' \in \mathbf{B}(\mathbf{x};s) \cap \mathcal{X}$, and (177) holds.

Note that $\Pr\left[\left|\mathbf{w}^\top \mathbf{x}'\right| \leq R\right] \leq \frac{2R}{\sqrt{2\pi}\kappa}$ for any $\mathbf{x}' \in \mathcal{X}$, changing the notation $\mathbf{x}'$ to $\mathbf{x}$ completes the proof.

$\square$

**Lemma C.20.** Let $\mathbf{w} \in \mathbb{R}^d$ be a Gaussian random vector distribute according to $\mathbf{w} \sim \mathcal{N}(\mathbf{0}, \kappa^2 \mathbf{I}_d)$. Then $\Pr\left[\|\mathbf{w}\|_2 \geq R'\right] \geq 1 - \exp\left(-\left(\frac{\sqrt{m}}{2} - \frac{R'^2}{2\sqrt{m}\kappa^2}\right)^2\right)$ for any $R' > 0$.

*Proof.* Let $X = \frac{\|\mathbf{w}\|_2^2}{\kappa^2}$, then $X$ follows the chi-square distribution with $m$ degrees of freedom, that is, $X \sim \chi^2(m)$. By (Laurent & Massart, 2000, Lemma 1), we have the following concentration inequalities for any $x > 0$,

$$\Pr\left[X - m \geq 2\sqrt{mx} + 2x\right] \leq \exp(-x), \Pr\left[m - X \geq 2\sqrt{mx}\right] \leq \exp(-x). \tag{178}$$

Setting $x = \left(\frac{\sqrt{m}}{2} - \frac{R'^2}{2\sqrt{m}\kappa^2}\right)^2$ in the second inequality in (178), we have $m - 2\sqrt{mx} = \frac{R'^2}{\kappa^2}$ and

$$\Pr\left[X \geq \frac{R'^2}{\kappa^2}\right] \geq 1 - \exp(-x). \tag{179}$$

It follows from (179) that

$$\Pr\left[\|\mathbf{w}\|_2 \geq R'\right] \geq 1 - \exp(-x) = 1 - \exp\left(-\left(\frac{\sqrt{m}}{2} - \frac{R'^2}{2\sqrt{m}\kappa^2}\right)^2\right), \tag{180}$$

which completes the proof.

$\square$

**Lemma C.21.** Suppose $R \leq R_0$ for an absolute positive constant $R_0 < \kappa$. Let $\mathcal{R}(\mathcal{V}_{\mathbf{x},s}) :=$ $\mathbb{E}_{\mathbf{W}(0),\{\sigma_r\}_{r=1}^m}\left[\sup_{v_R \in \mathcal{V}_{\mathbf{x},s}} \frac{1}{m}\sum_{r=1}^m \sigma_r v_R(\vec{\mathbf{w}}_r(0))\right]$ be the Rademacher complexity of the function class $\mathcal{V}_{\mathbf{x},s}$. Then

$$\mathcal{R}(\mathcal{V}_{\mathbf{x},s}) \leq (B\sqrt{d}+1)\sqrt{m^{-\frac{1}{2}}+s} + \frac{\exp\left(-\frac{(\kappa^2-R_0^2)^2}{4\kappa^4}m\right)}{\sqrt{m^{-\frac{1}{2}}+s}}, \tag{181}$$

where $B$ is a positive constant.

*Proof.* We have

$$\mathcal{R}(\mathcal{V}_{\mathbf{x},s}) = \mathbb{E}_{\mathbf{W}(0),\{\sigma_r\}_{r=1}^m}\left[\sup_{\mathbf{x}' \in \mathbf{B}(\mathbf{x};s)} \frac{1}{m}\sum_{r=1}^m \sigma_r v_R(\vec{\mathbf{w}}_r(0), \mathbf{x}')\right]$$

$$\leq \mathcal{R}_1 + \mathcal{R}_2, \tag{182}$$

where

$$\mathcal{R}_1 = \mathbb{E}_{\mathbf{W}(0),\{\sigma_r\}_{r=1}^m}\left[\sup_{\mathbf{x}' \in \mathbf{B}(\mathbf{x};s)} \frac{1}{m}\sum_{r=1}^m \sigma_r v_R(\vec{\mathbf{w}}_r(0), \mathbf{x})\right],$$

$$\mathcal{R}_2 = \mathbb{E}_{\mathbf{W}(0),\{\sigma_r\}_{r=1}^m}\left[\sup_{\mathbf{x}' \in \mathbf{B}(\mathbf{x};s)} \frac{1}{m}\sum_{r=1}^m \sigma_r \left(v_R(\vec{\mathbf{w}}_r(0), \mathbf{x}') - v_R(\vec{\mathbf{w}}_r(0), \mathbf{x})\right)\right]. \tag{183}$$

Here (182) follows from the subadditivity of superemum and the fact that $\sum_{r=1}^m \sigma_r v_R(\vec{\mathbf{w}}_r(0), \mathbf{x}') = \sum_{r=1}^m \sigma_r v_R(\vec{\mathbf{w}}_r(0), \mathbf{x}) + \sum_{r=1}^m \sigma_r \left(v_R(\vec{\mathbf{w}}_r(0), \mathbf{x}') - v_R(\vec{\mathbf{w}}_r(0), \mathbf{x})\right)$.

Now we bound $\mathcal{R}_1$ and $\mathcal{R}_2$ separately. For $\mathcal{R}_1$, we have

$$\mathcal{R}_1 = \mathbb{E}_{\mathbf{W}(0),\{\sigma_r\}_{r=1}^m}\left[\sup_{\mathbf{x}' \in \mathbf{B}(\mathbf{x};s)} \frac{1}{m}\sum_{r=1}^m \sigma_r v_R(\vec{\mathbf{w}}_r(0), \mathbf{x})\right] = 0. \tag{184}$$

For $\mathcal{R}_2$, we first define

$$Q = \frac{1}{m}\sum_{r=1}^m \mathbb{I}_{\left\{\mathbb{I}_{\left\{\left|\mathbf{x}'^\top \vec{\mathbf{w}}_r(0)\right| \leq R\right\}} \neq \mathbb{I}_{\left\{\left|\mathbf{x}^\top \vec{\mathbf{w}}_r(0)\right| \leq R\right\}}\right\}},$$

which is the number of weights in $\mathbf{W}(0)$ whose inner products with $\mathbf{x}$ and $\mathbf{x}'$ have different signs. Note that if $\left|\left|\mathbf{x}^\top \vec{\mathbf{w}}_r(0)\right| - R\right| > s\left\|\vec{\mathbf{w}}_r(0)\right\|_2$, then $\mathbb{I}_{\left\{\left|\mathbf{x}^\top \vec{\mathbf{w}}_r(0)\right| \leq R\right\}} = \mathbb{I}_{\left\{\left|\mathbf{x}'^\top \vec{\mathbf{w}}_r(0)\right| \leq R\right\}}$. To see this, by the Cauchy-Schwarz inequality,

$$\left|\mathbf{x}'^\top \vec{\mathbf{w}}_r(0) - \mathbf{x}^\top \vec{\mathbf{w}}_r(0)\right| \leq \|\mathbf{x}' - \mathbf{x}\|_2 \left\|\vec{\mathbf{w}}_r(0)\right\|_2 \leq s \left\|\vec{\mathbf{w}}_r(0)\right\|_2, \tag{185}$$

then we have $\left|\mathbf{x}^\top \vec{\mathbf{w}}_r(0)\right| - R > s\left\|\vec{\mathbf{w}}_r(0)\right\|_2 \Rightarrow \left|\mathbf{x}'^\top \vec{\mathbf{w}}_r(0)\right| - R \geq \left|\mathbf{x}^\top \vec{\mathbf{w}}_r(0)\right| - s\left\|\vec{\mathbf{w}}_r(0)\right\|_2 - R > 0$, and $\left|\mathbf{x}^\top \vec{\mathbf{w}}_r(0)\right| - R < -s\left\|\vec{\mathbf{w}}_r(0)\right\|_2 \Rightarrow \left|\mathbf{x}'^\top \vec{\mathbf{w}}_r(0)\right| - R \leq \left|\mathbf{x}^\top \vec{\mathbf{w}}_r(0)\right| + s\left\|\vec{\mathbf{w}}_r(0)\right\|_2 - R < 0$.

As a result, $Q \leq \frac{1}{m} \sum_{r=1}^m \mathbb{1}_{\left\{\left|\left|\mathbf{x}^\top \vec{\mathbf{w}}_r(0)\right| - R\right| \leq s\left\|\vec{\mathbf{w}}_r(0)\right\|_2\right\}}$. For any fixed $r \in [m]$, by Lemma C.20,

$\Pr\left[\left\|\vec{\mathbf{w}}_r(0)\right\|_2 \geq R'\right] \geq 1 - \exp\left(-\left(\frac{\sqrt{m}}{2} - \frac{R'^2}{2\sqrt{m}\kappa^2}\right)^2\right)$ holds for any $R' \geq 0$. Set $R' = \sqrt{m}R_0$ for the constant $R_0 < \kappa$. Because $R \leq R_0$, it follows that

$$\Pr\left[\frac{R}{\left\|\vec{\mathbf{w}}_r(0)\right\|_2} \leq m^{-\frac{1}{2}}\right] \geq \Pr\left[\frac{R_0}{\left\|\vec{\mathbf{w}}_r(0)\right\|_2} \leq m^{-\frac{1}{2}}\right] \geq 1 - \exp\left(-\frac{(\kappa^2 - R_0^2)^2}{4\kappa^4}m\right). \tag{186}$$

Due to the fact that $\mathbb{1}_{\left\{\left|\left|\mathbf{x}^\top \vec{\mathbf{w}}_r(0)\right| - R\right| \leq s\left\|\vec{\mathbf{w}}_r(0)\right\|_2\right\}} \leq \mathbb{1}_{\left\{\left|\mathbf{x}^\top \vec{\mathbf{w}}_r(0)\right| \leq R+s\left\|\vec{\mathbf{w}}_r(0)\right\|_2\right\}}$, we have

$$\mathbb{E}_{\mathbf{W}(0)}\left[\mathbb{1}_{\left\{\left|\left|\mathbf{x}^\top \vec{\mathbf{w}}_r(0)\right| - R\right| \leq s\left\|\vec{\mathbf{w}}_r(0)\right\|_2\right\}}\right]$$

$$\leq \mathbb{E}_{\vec{\mathbf{w}}_r(0)}\left[\mathbb{1}_{\left\{\left|\mathbf{x}^\top \vec{\mathbf{w}}_r(0)\right| \leq R+s\left\|\vec{\mathbf{w}}_r(0)\right\|_2\right\}}\right]$$

$$\overset{①}{\leq} \mathbb{E}_{\vec{\mathbf{w}}_r(0): \frac{R}{\left\|\vec{\mathbf{w}}_r(0)\right\|_2} \leq m^{-\frac{1}{2}}}\left[\mathbb{1}_{\left\{\left|\mathbf{x}^\top \vec{\mathbf{w}}_r(0)\right| \leq R+s\left\|\vec{\mathbf{w}}_r(0)\right\|_2\right\}}\right] + \mathbb{E}_{\vec{\mathbf{w}}_r(0): \frac{R}{\left\|\vec{\mathbf{w}}_r(0)\right\|_2} > m^{-\frac{1}{2}}}\left[\mathbb{1}_{\left\{\left|\mathbf{x}^\top \vec{\mathbf{w}}_r(0)\right| \leq R+s\left\|\vec{\mathbf{w}}_r(0)\right\|_2\right\}}\right]$$

$$\overset{②}{\leq} \mathbb{E}_{\vec{\mathbf{w}}_r(0): \frac{R}{\left\|\vec{\mathbf{w}}_r(0)\right\|_2} \leq m^{-\frac{1}{2}}}\left[\mathbb{1}_{\left\{\left|\mathbf{x}^\top \vec{\mathbf{w}}_r(0)\right| \leq (m^{-\frac{1}{2}}+s)\left\|\vec{\mathbf{w}}_r(0)\right\|_2\right\}}\right] + \exp\left(-\frac{(\kappa^2 - R_0^2)^2}{4\kappa^4}m\right)$$

$$\leq \Pr\left[\frac{\left|\mathbf{x}^\top \vec{\mathbf{w}}_r(0)\right|}{\left\|\vec{\mathbf{w}}_r(0)\right\|_2} \leq m^{-\frac{1}{2}} + s\right] + \exp\left(-\frac{(\kappa^2 - R_0^2)^2}{4\kappa^4}m\right)$$

$$\overset{③}{\leq} B\sqrt{d}(m^{-\frac{1}{2}} + s) + \exp\left(-\frac{(\kappa^2 - R_0^2)^2}{4\kappa^4}m\right), \tag{187}$$

where we used the convention that $\mathbb{E}_{\vec{\mathbf{w}}_r(0) \in A}[\cdot] = \mathbb{E}_{\vec{\mathbf{w}}_r(0)}\left[\mathbb{1}_{\{A\}} \times \cdot\right]$ in ① with $A$ being an event. ② is due to (186). By Lemma C.19, $\Pr\left[\frac{\left|\mathbf{x}^\top \vec{\mathbf{w}}_r(0)\right|}{\left\|\vec{\mathbf{w}}_r(0)\right\|_2} \leq m^{-\frac{1}{2}} + s\right] \leq B\sqrt{d}(m^{-\frac{1}{2}} + s)$ for an absolute constant $B$, so ③ holds.

According to (187), we have

$$\mathbb{E}_{\mathbf{W}(0)}[Q] \leq \mathbb{E}_{\mathbf{W}(0)}\left[\frac{1}{m} \sum_{r=1}^m \mathbb{1}_{\left\{\left|\left|\mathbf{x}^\top \vec{\mathbf{w}}_r(0)\right| - R\right| \leq s\left\|\vec{\mathbf{w}}_r(0)\right\|_2\right\}}\right]$$

$$\leq B\sqrt{d}(m^{-\frac{1}{2}} + s) + \exp\left(-\frac{(\kappa^2 - R_0^2)^2}{4\kappa^4}m\right). \tag{188}$$

Define $s' := m^{-\frac{1}{2}} + s$. By Markov's inequality, we have

$$\Pr\left[Q \geq \sqrt{s'}\right] \leq B\sqrt{d s'} + \frac{\exp\left(-\frac{(\kappa^2 - R_0^2)^2}{4\kappa^4}m\right)}{\sqrt{s'}}, \tag{189}$$

where the probability is with respect to the probability measure space of $\mathbf{W}(0)$. Now we aim to bound $\mathcal{R}_2$ by estimating its bound on $\Omega_s$ and its complement. First, we have

$$\mathcal{R}_2 = \mathbb{E}_{\mathbf{W}(0),\{\sigma_r\}_{r=1}^m} \left[ \sup_{\mathbf{x}' \in \mathbf{B}(\mathbf{x};s)} \frac{1}{m} \sum_{r=1}^m \sigma_r \left( v_R(\vec{\mathbf{w}}_r(0),\mathbf{x}') - v_R(\vec{\mathbf{w}}_r(0),\mathbf{x}) \right) \right]$$

$$= \underbrace{\mathbb{E}_{\mathbf{W}(0)\colon Q \geq \sqrt{s'},\{\sigma_r\}_{r=1}^m} \left[ \sup_{\mathbf{x}' \in \mathbf{B}(\mathbf{x};s)} \frac{1}{m} \sum_{r=1}^m \sigma_r \left( v_R(\vec{\mathbf{w}}_r(0),\mathbf{x}') - v_R(\vec{\mathbf{w}}_r(0),\mathbf{x}) \right) \right]}_{\mathcal{R}_{21}}$$

$$+ \underbrace{\mathbb{E}_{\mathbf{W}(0)\colon Q < \sqrt{s'},\{\sigma_r\}_{r=1}^m} \left[ \sup_{\mathbf{x}' \in \mathbf{B}(\mathbf{x};s)} \frac{1}{m} \sum_{r=1}^m \sigma_r \left( v_R(\vec{\mathbf{w}}_r(0),\mathbf{x}') - v_R(\vec{\mathbf{w}}_r(0),\mathbf{x}) \right) \right]}_{\mathcal{R}_{22}}, \quad (190)$$

Now we estimate the upper bound for $\mathcal{R}_{22}$ and $\mathcal{R}_{21}$ separately. Let

$$I = \left\{ r \in [m]\colon \mathbb{1}_{\left\{ \left| \mathbf{x}'^\top \vec{\mathbf{w}}_r(0) \right| \leq R \right\}} \neq \mathbb{1}_{\left\{ \left| \mathbf{x}^\top \vec{\mathbf{w}}_r(0) \right| \leq R \right\}} \right\}.$$

When $Q < \sqrt{s'}$, $|I| \leq m\sqrt{s'}$. Moreover, when $r \in I$, either $\mathbb{1}_{\left\{ \left| \mathbf{x}'^\top \vec{\mathbf{w}}_r(0) \right| \leq R \right\}} = 0$ or $\mathbb{1}_{\left\{ \left| \mathbf{x}^\top \vec{\mathbf{w}}_r(0) \right| \leq R \right\}} = 0$. As a result,

$$\left| v_R(\vec{\mathbf{w}}_r(0),\mathbf{x}') - v_R(\vec{\mathbf{w}}_r(0),\mathbf{x}) \right| = \left| \mathbb{1}_{\left\{ \left| \mathbf{x}'^\top \vec{\mathbf{w}}_r(0) \right| \leq R \right\}} - \mathbb{1}_{\left\{ \left| \mathbf{x}^\top \vec{\mathbf{w}}_r(0) \right| \leq R \right\}} \right|$$

$$\leq \max \left\{ \mathbb{1}_{\left\{ \left| \mathbf{x}'^\top \vec{\mathbf{w}}_r(0) \right| \leq R \right\}}, \mathbb{1}_{\left\{ \left| \mathbf{x}^\top \vec{\mathbf{w}}_r(0) \right| \leq R \right\}} \right\}$$

$$\leq 1. \quad (191)$$

When $r \in [m] \setminus I$, we have

$$\left| v_R(\vec{\mathbf{w}}_r(0),\mathbf{x}') - v_R(\vec{\mathbf{w}}_r(0),\mathbf{x}) \right| = \left| \mathbb{1}_{\left\{ \left| \mathbf{x}'^\top \vec{\mathbf{w}}_r(0) \right| \leq R \right\}} - \mathbb{1}_{\left\{ \left| \mathbf{x}^\top \vec{\mathbf{w}}_r(0) \right| \leq R \right\}} \right| = 0. \quad (192)$$

By (191) and (192), we have

$$\frac{1}{m} \sum_{r=1}^m \sigma_r \left( v_R(\vec{\mathbf{w}}_r(0),\mathbf{x}') - v_R(\vec{\mathbf{w}}_r(0),\mathbf{x}) \right)$$

$$= \frac{1}{m} \sum_{r \in I} \sigma_r \left( v_R(\vec{\mathbf{w}}_r(0),\mathbf{x}') - v_R(\vec{\mathbf{w}}_r(0),\mathbf{x}) \right) + \frac{1}{m} \sum_{r \in [m] \setminus I} \sigma_r \left( v_R(\vec{\mathbf{w}}_r(0),\mathbf{x}') - v_R(\vec{\mathbf{w}}_r(0),\mathbf{x}) \right)$$

$$\leq \frac{1}{m} \sum_{r \in I} \left| v_R(\vec{\mathbf{w}}_r(0),\mathbf{x}') - v_R(\vec{\mathbf{w}}_r(0),\mathbf{x}) \right| + \frac{1}{m} \sum_{r \in [m] \setminus I} \left| v_R(\vec{\mathbf{w}}_r(0),\mathbf{x}') - v_R(\vec{\mathbf{w}}_r(0),\mathbf{x}) \right|$$

$$\overset{①}{\leq} \frac{m\sqrt{s'}}{m} \leq \sqrt{s'}, \quad (193)$$

where ① uses the bounds in (191) and (192).

Using (193), we now estimate the upper bound for $\mathcal{R}_{22}$ by

$$\mathcal{R}_{22} = \mathbb{E}_{\mathbf{W}(0)\colon Q < \sqrt{s'},\{\sigma_r\}_{r=1}^m} \left[ \sup_{\mathbf{x}' \in \mathbf{B}(\mathbf{x};s)} \frac{1}{m} \sum_{r=1}^m \sigma_r \left( v_R(\vec{\mathbf{w}}_r(0),\mathbf{x}') - v_R(\vec{\mathbf{w}}_r(0),\mathbf{x}) \right) \right]$$

$$\leq \mathbb{E}_{\mathbf{W}(0)\colon Q<\sqrt{s'},\{\sigma_r\}_{r=1}^m} \left[ \sqrt{s'} \right] = \sqrt{s'}. \tag{194}$$

When $Q \geq \sqrt{s'}$, by (191), we still have $\left| v_R(\vec{\mathbf{w}}_r(0),\mathbf{x}') - v_R(\vec{\mathbf{w}}_r(0),\mathbf{x}) \right| \leq 1$. For $\mathcal{R}_{21}$, we have

$$\mathcal{R}_{21} = \mathbb{E}_{\mathbf{W}(0)\colon Q\geq\sqrt{s'},\{\sigma_r\}_{r=1}^m} \left[ \sup_{\mathbf{x}'\in\mathbf{B}(\mathbf{x};s)} \frac{1}{m}\sum_{r=1}^m \sigma_r \Big( v_R(\vec{\mathbf{w}}_r(0),\mathbf{x}') - v_R(\vec{\mathbf{w}}_r(0),\mathbf{x}) \Big) \right]$$

$$\leq \mathbb{E}_{\mathbf{W}(0)\colon Q\geq\sqrt{s'},\{\sigma_r\}_{r=1}^m} \left[ \sup_{\mathbf{x}'\in\mathbf{B}(\mathbf{x};s)} \frac{1}{m}\sum_{r=1}^m \left| \sigma_r \Big( v_R(\vec{\mathbf{w}}_r(0),\mathbf{x}') - v_R(\vec{\mathbf{w}}_r(0),\mathbf{x}) \Big) \right| \right]$$

$$\leq \mathbb{E}_{\mathbf{W}(0)\colon Q\geq\sqrt{s'},\{\sigma_r\}_{r=1}^m} \left[ 1 \right] = \Pr\left[ Q \geq \sqrt{s'} \right] \leq B\sqrt{ds'} + \frac{\exp\left( -\frac{(\kappa^2-R_0^2)^2}{4\kappa^4}m \right)}{\sqrt{s'}}, \tag{195}$$

where the last inequality is due to (189). Combining (190), (194), and (195), we have the upper bound for $\mathcal{R}_2$ as

$$\mathcal{R}_2 = \mathcal{R}_{21} + \mathcal{R}_{22} \leq (B\sqrt{d}+1)\sqrt{s'} + \frac{\exp\left( -\frac{(\kappa^2-R_0^2)^2}{4\kappa^4}m \right)}{\sqrt{s'}}. \tag{196}$$

Plugging (184) and (196) in (182), we have

$$\mathcal{R}(\mathcal{V}_{\mathbf{x},s}) \leq \mathcal{R}_1 + \mathcal{R}_2 \leq (B\sqrt{d}+1)\sqrt{s'} + \frac{\exp\left( -\frac{(\kappa^2-R_0^2)^2}{4\kappa^4}m \right)}{\sqrt{s'}}, \tag{197}$$

which completes the proof.

$\square$

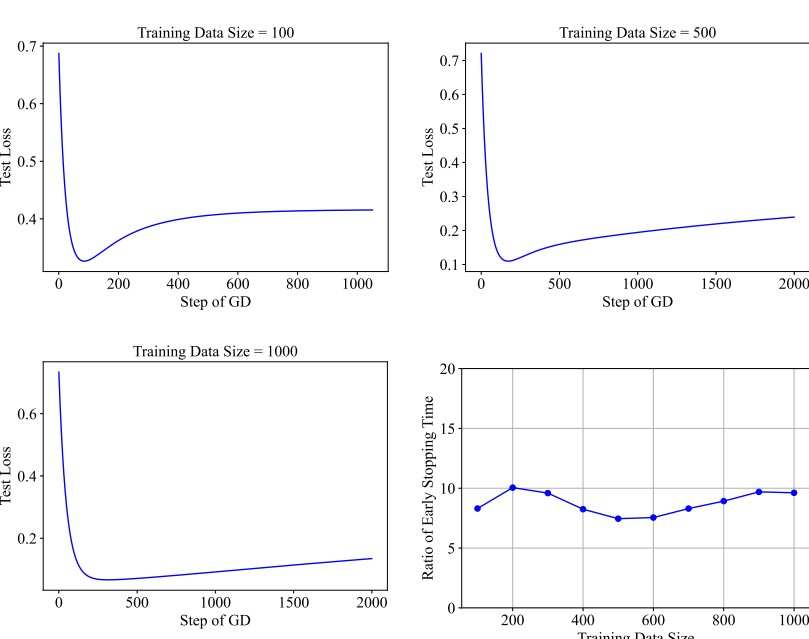

Figure 2: Illustration of the test loss by GD

# D  SIMULATION STUDY

We present simulation results for GD in this section. We randomly sample $n$ points $\left\{ \vec{\mathbf{x}}_i \right\}_{i=1}^{n}$ as a i.i.d. sample of random variables distributed uniformly on the unit sphere in $\mathbb{R}^{50}$. $n$ ranges within $[100, 1000]$ with a step size of 100. We set the target function to $f^*(\mathbf{x}) = \mathbf{s}^\top \mathbf{x}$ where $\mathbf{s} \sim \text{Unif}\,(\mathcal{X})$ is randomly sampled. We also uniformly and independenly sample 1000 points on the unit sphere in $\mathbb{R}^{50}$ as the test data. We train the two-layer NN (1) using either GD by Algoirthm 1 or GD by Algoirthm 1 with $m \asymp n^2$ on a NVIDIA A100 GPU card with a learning rate $\eta = 0.1$, and report the test loss in Figure 2. It can be observed that early-stopping is always helpful in training neural networks with better generalization, as the test loss initially decreases and then increases with over-training. Figure 2 illustrates the test loss with respect to the steps (or epochs) of GD for $n = 100, 500, 1000$. For each $n$ in $[100, 1000]$ with a step size of 100, we find the step of GD where minimum test loss is achieved, denoted by $\widehat{t}_n$ which is the empirical early stopping time. We note that the theoretically predicted early stopping time is $\widehat{\varepsilon}_n = n^{-d/(2d-1)}$, and we compute the ratio of early stopping time for each $n$ by $\widehat{t}_n / \widehat{\varepsilon}_n$. Such ratios for different values of $n$ are illustrated in the bottom right figure of Figure 2. It is observed that the ratio of early stopping time is roughly stable and distributed between $[8, 10]$, suggesting that predicted early stopping time is empirically proportional to the empirical early stopping time.

