# OpenReview forum: "Sharp Generalization for Nonparametric Regression by Over-Parameterized Neural Networks: A Distribution-Free Analysis"
_ICLR.cc/2025/Conference — ICLR 2025 Conference Withdrawn Submission_

### Official Review · Reviewer_agPR · 2024-10-23

**Soundness:** 2
**Presentation:** 2
**Contribution:** 2
**Rating:** 5
**Confidence:** 3

**Summary:**

This paper provides generalization bounds for overparameterized two-layer neural networks (NNs) with early stopping without making assumptions on the covariate distribution, except being supported on the unit sphere. The proof shows uniform convergence to the NTK and bounds the Rademacher complexity of the function class that is learned by the NNs.

**Strengths:**

This paper poses weaker assumptions on the covariate distribution, in particular no assumptions on the NTK’s eigenvalue decay rate, than previous works like Suh et al. (2022) or Li et al. (2024), which is a significant contribution.

The results suggest that the required network width can be estimated from data by estimating the NTK’s eigenvalue decay rate (EDR). However, a successful estimation of the EDR or of the required network width is not shown, neither empirically nor theoretically.

**Weaknesses:**

Overall, I believe this paper requires a significant revision and should be rejected, due to the following severe weaknesses:

- **Overstatement.** One of the main strengths that the authors claim is that their analysis is “distribution-free”.  This is an overstatement. While the assumptions on the covariate distribution are weak (that it should be supported on $\mathcal{X}=S^{d-1}$, in particular this implies that the covariates have compact, bounded support), the assumptions on the conditional distribution y|x are quite strong. In particular, the Bayes optimal function should lie in the NTK’s RKHS with low norm and the label noise is iid Gaussian. At no point in the paper is the claim of “distribution-free” qualified. For example, in the Abstract, the authors write “our result does not require distributional assumptions on the training data”, which is just wrong.
- **Artificial Setting.** Only first-layer weights of a 2-layer NN without biases are trained with GD and the anti-symmetric initialization trick is used. The comparable papers Suh et al. (2021) and Li et al. (2024) treat training all layers of L-hidden layer MLPs. Du et al. (2019) also discusses training both layers. The authors do not make sufficiently transparent in which ways this paper poses stronger assumptions or analyses more artificial settings than previous work that trains all layers, uses SGD (Allen-Zhu et al.,2019) or that poses weaker assumptions on y|x (Haas et al., 2024).
- **Related work** has not been sufficiently acknowledged. In particular:
    - The antisymmetric initialization trick used in this work has been previously used in the literature, e.g. Zhang et al. (2020).
    - It is never mentioned that uniform convergence to the NTK has previously been shown.
    - The statement in lines 50-53 is imprecise. It is only correct under the neural tangent parameterization. Yang and Hu (2021) have shown that there are parameterizations such that NNs do learn features even in the infinite-width limit.
    - In lines 69-71, it is claimed that previous studies that achieve minimax optimal rates with overparameterized NNs impose strong distributional assumptions. Theorem G.5 from Haas et al. (2024) achieves minimax optimal rates up to log terms under a weak covariate assumption in the overparameterized limit. In particular, there $f^*$ may also lie outside of the NTK’s RKHS, which is much less restrictive than the boundedness assumption on the RKHS norm here, and label noise does not have to be Gaussian.
- **Missing experiments.** While not necessary, experiments to quantify finite-width and finite-sample relevance of the derived bounds would be a valuable addition. E.g. Du et al. (2019) provide minimal experiments. Useful experiments could evaluate for increasing n:
    1. What is empirically the optimal stopping time $\hat T_{emp}$? Does the theoretically predicted $\hat T$ coincide at least in the rate?
    2. Does the generalization error decay with the predicted rate, for one typical covariate distribution, like uniform on the sphere, and one which does not satisfy the stronger assumptions in other works?
    3. Is the predicted necessary scaling of $m$ tight or can it be chosen even smaller?
- **Unpolished writing.** Throughout the paper, some descriptions are lengthy, repetitive or little informative. For example, the description in Section 6.2 could be shortened, and instead of only describing what has been done in the proofs, it could shortly be explained *how* it was possible to show uniform convergence to the NTK and to bound the local Rademacher complexity. Figure 1 would only be helpful, if it also contained what the respective results in the appendix say.

**Things to improve the paper that did not impact the score:**

- The notation $f_0$ is confusing. It would usually denote a function, not a constant.

**Typos:** 264: first, 363: small t in the minimum?, 379-380: several mistakes.

**Questions:**

- Can you provide a short sketch or a short overview how it was possible to show uniform convergence to the NTK and to bound the local Rademacher complexity?
- In which ways does the analysis differ from previous analyses?

---

> ### Author Response · Authors · 2024-11-13
> **Request for clarification about the reference (Haas et al., 2024)**
>
> Dear Reviewer agPR,
>
> Thank you for your comments. We would like to let you know that the raised issues are being addressed and the summary of our revisions will be ready for you to review very soon along with a revised paper. We would like to respectfully point out that we will revise the claims about "distribution-free" to "distribution-free in the spherical covariates" specifying that the covariates are only supposed to be distributed on the unit sphere for arbitrary spherical distribution, which does not change all the results/contributions of this paper and addresses your concern about overstatement. Please kindly note that we can always normalize the covariates so they lie on the unit sphere. We are also revising the paper with a more detailed discussion about the literature as you mentioned.
>
> At this point, could you provide the detailed reference information for (Haas et al., 2024) mentioned in your review? Thank you!
>
> Regards,
>
> The Authors

---

> > ### Comment · Reviewer_agPR · 2024-11-15
> >
> > Dear Authors,
> >
> > thank you for your comment. I am looking forward to your revision. I am sorry for not providing the references; below you can find the full list. My concern is less about the results in Haas et al. (2024) in particular, but rather about clearly acknowledging assumptions and limitations, for example the strong assumptions on y|x. If you prefer another example of work in the overparameterized regime that allows regression functions outside the RKHS you could consider Bordelon et al. (2024).
> >
> > **References:**
> >
> > Allen-Zhu, Zeyuan, Yuanzhi Li, and Yingyu Liang. *Learning and
> > generalization in overparameterized neural networks, going beyond two
> > layers.* Advances in neural information processing systems 32 (2019).
> >
> > Bordelon, Blake, Alexander Atanasov, and Cengiz Pehlevan. *How Feature Learning Can Improve Neural Scaling Laws.* arXiv:2409.17858 (2024).
> >
> > Du, Simon S., Xiyu Zhai, Barnabas Poczos, and Aarti Singh. *Gradient
> > descent provably optimizes over-parameterized neural networks.* arXiv:1810.02054 (2018).
> >
> > Haas, M., Holzmüller, D., Luxburg, U. and Steinwart, I., 2024. *Mind the
> > spikes: Benign overfitting of kernels and neural networks in fixed
> > dimension.* Advances in Neural Information Processing Systems, 36.
> >
> > Li, Yicheng, Zixiong Yu, Guhan Chen, and Qian Lin. *On the Eigenvalue
> > Decay Rates of a Class of Neural-Network Related Kernel Functions
> > Defined on General Domains.* Journal of Machine Learning Research 25, no. 82 (2024): 1-47.
> >
> > Suh, Namjoon, Hyunouk Ko, and Xiaoming Huo. *A non-parametric regression
> >  viewpoint: Generalization of overparametrized deep ReLU network under
> > noisy observations.* In International Conference on Learning Representations. 2022.
> >
> > Yang, G. and Hu, E.J., 2020. *Feature learning in infinite-width neural networks.* arXiv:2011.14522.
> >
> > Yaoyu Zhang, Zhi-Qin John Xu, Tao Luo, and Zheng Ma. *A type of generalization error induced by
> > initialization in deep neural networks.* In Proceedings of The First Mathematical and Scientific
> > Machine Learning Conference, 2020.

---

> ### Author Response · Authors · 2024-11-28
> **Response to Reviewer agPR Part 1**
>
> We appreciate the review and the suggestions in this review. The raised issues are addressed below. In the following text, the line numbers are for the revised paper.
>
> **(1) More precise description of our contribution by changing “distribution-free” to “distribution-free in spherical covariate”, and improved technical results allowing for sub-Gaussian noise**
>
> Addressing the **raised issue about Overstatement** that our result is not distribution-free such as that from the perspective of $P(y|x)$, we have revised our paper and now we clearly specify that our result is “distribution-free in spherical covariate”, which means that “there are no distributional assumptions about the covariate as long as the covariate lies on the unit sphere.” (in line 83-84). All the claims are revised in this manner in the revised paper.  We have also extend our results with Gaussian noise to those with sub-Gaussian noise (line 133-134) with only minor change of the proofs.
>
> **(2) More discussion about this work and the relevant literature**
>
> We herein provide more discussion about the results of this work and comparison to the existing relevant works with sharp rates for nonparametric regression, addressing the **raised issues about Artificial Setting, Related work** (summarized in line 306-312, detailed in line 781-793), which also acknowledge the assumptions and limitations of this work.
>
> While this paper establishes sharp rate which is distribution-free in spherical covariate, such rate still depends on bounded input space (${\mathcal X} = {\mathbb S}^{d-1}$) and the condition that the target function $f^* \in \mathcal H_K(\mu_0)$. Some other existing works consider target function $f^*$ not belonging to the RKHS ball centered at the origin with constant or low radius, such as [HaasHLS23,Bordelon2024]. However, the target functions in [HaasHLS23,Bordelon2024] escape the finite norm or low-norm regime of RKHS at the cost of either restriction condition on the density function of the covariate distribution or the training process. In particular, Theorem G.5 in [HaasHLS23] requires the condition for bounded density function (in its condition (D3) ) of the distribution $P$, which is not required by our result. Moreover, the training process of the model in [Bordelon2024] requires information about the target function (in its Eq. (4)) and certain distribution $P$ which admits certain polynomial EDR, that is, $\lambda_j \asymp j^{-\alpha}$ with $\alpha > 1$, which happens under certain restrictive conditions on $P$.
>
> We also note that in this work, only the first layer of an over-parameterized two-layer neural network is trained, while the weights of the second layer are randomly initialized and then fixed in the training process. In existing works such as [HuWLC21,SuhKH22,Allen-ZhuLL19],  all the layers of a deep neural networks with more than two layers are trained by GD or its variants. However, this work shows that only training the first layer still leads to  sharp rate for nonparametric regression, which supports the claim in [BiettiB21] that a shallow over-parameterized neural networks with ReLU activations exhibit the same approximation properties as its deeper counterpart.
>
> The full version of the discussion above will be added to the final version of this paper.
>
> **Regarding existing result about uniform convergence to NTK**. We have added the following discussion before Theorem 6.1 of the revised paper: “While existing works such as [Li2024] also have uniform convergence results for over-parameterized neural network, our result does not depend on the Holder continuity of the NTK.”
>
> **Regarding existing works about antisymmetric initialization**.  References for such initialization trick are added to line 340-341.
>
> **"The statement in lines 50-53 is imprecise…"** The corrections are made to line 57-58.
>
> **(3) Added Experimental Results**
>
> Section D of the revised paper contains experimental results for $P$ as the uniform spherical distribution with the network width $m = n^2$. For each training data size $n$, the empirical early stopping time $\hat t_n$ is the step of GD where minimum test loss is achieved, and the theoretically predicted early stopping time is $1/\hat \epsilon^2_n \asymp n^{d/(2d-1)}$. We compute the ratio of early stopping time for each $n$ by $\hat t_n/n^{d/(2d-1)}$. Such ratios for different values of $n$ are illustrated in the bottom right figure of Figure 2. It is observed that the ratio of early stopping time is roughly stable and distributed between $[8,10]$ for all $n$ within $[100,1000]$ with a step size of $100$, suggesting that predicted early stopping time is empirically proportional to the empirical early stopping time. Some notations in Section D will be updated with the notations in this response, and we will perform additional empirical study for other types of distribution $P$ in the final version of this paper.

---

> ### Author Response · Authors · 2024-11-28
> **Response to Reviewer agPR Part 2**
>
> **(3) Added Experimental Results (Cont'd)**. $m$ used in the experiment is smaller than the derived lower bound for it, so finding sharper lower bound for $m$ can be a future work.
>
> **(4) Improved Presentation detailing the novelty of our results and the novel proof strategy of this work (including how our proof strategy/analysis differs from existing works)**
>
> The pointed issues about the typos and the notation $f_0$ have been fixed in the revised paper.
>
> In Section 6.2 of the revised paper,  we present a new **Summary of the technical approaches and novel results in the proofs** and **Novel proof strategy of this work**, addressing the concerns of unpolished writing, and they are copied below for your convenience.
>
> **Summary of the technical approaches and novel results in the proofs**. Theorem C.8 is the first novel result in the proofs of this work, showing that with high probability, the neural network function $f(\mathbf W(t),\cdot)$ at step $t$ of GD can be decomposed into two functions by $f(\mathbf W(t),\cdot) = f_t = h  + e$, where $h \in \mathcal H_{K}$ is a function in the RKHS associated with $K$ with bounded $\mathcal H_{K}$-norm. The error function $e$ has a small $L^{\infty}$-norm, that is, $|| e || _ {\infty} \le w$ with $w$ being a small number controlled by the network width $m$, that is, larger $m$ leads to smaller $w$. Theorem C.10 is the second novel result in the proofs, where we derive sharp and novel bound for the nonparametric regression risk of the neural network function $f(\mathbf W(t),\cdot)$, that is, $E_P (f_t-f^*)^2 - 2 E_{P_n}(f_t-f^*)^2 \le \Theta(\epsilon_n^2 + w)$. To the best of our knowledge, Theorem C.10 is among the first in the literature to employ local Rademacher complexity so as to obtain sharp rate for the risk of nonparametric regression which is distribution-free in spherical covariate,  and local Rademacher complexity is employed to tightly bound the Rademacher complexity of the function class comprising all the possible neural network functions obtained by GD.
>
> **Novel proof strategy of this work**. We remark that the proof strategy of our main result,
> Theorem 5.1, is significantly novel and different from the existing works in training over-parameterized neural networks for nonparametric regression with minimax rates [HuWLC21,SuhKH22,Li2024]. In particular, the common proof strategy in these works uses the decomposition $f_t-f^* = (f_t - \hat f_t^{\textup{(NTK)}}) + (\hat f_t^{\textup{(NTK)}} - f^*)$ and then show that both $|| f_t - \hat f_t^{\textup{(NTK)}} || _ {L^2}$ and $|| \hat f_t^{\textup{(NTK)}}- f^*|| _ {L^2}$ are bounded by certain minimax optimal rate, where $\hat f_t^{\textup{(NTK)}}$ is the kernel regressor obtained by either kernel ridge regression [HuWLC21,SuhKH22 or gradient-flow based GD with early stopping [Li2024]. The remark after Theorem C.8 details a formulation of $\hat f_t^{\textup{(NTK)}}$.
> $||\hat f_t^{\textup{(NTK)}}- f^*\| _ {L^2}$ is bounded by the minimax optimal rate under certain distributional assumptions in the covariate, and this is one reason for the distributional assumptions about the covariate in existing works such as [HuWLC21,SuhKH22,Li2024]. In a strong contrast, our analysis does not rely on such decomposition of $f_t-f^*$. Instead of approximating $f_t$ by $\hat f_t^{\textup{(NTK)}}$, we have a new decomposition of $f_t$ by $f_t = h_t  + e_t$ where $f_t$ is approximated by $h_t$ with $e_t$ being the approximation error. As suggested by the remark after Theorem C.8, we have $h_t = \hat f_t^{\textup{(NTK)}}  + \hat e_2(\cdot,t)$ so that $f_t = \hat f_t^{\textup{(NTK)}}  + \hat e_2(\cdot,t)+ e_t$. Our analysis only requires the network width
> $m$ to be suitably large so that the $\mathcal H_K$-norm of $\hat e_2(\cdot,t)$ is bounded by a positive constant and $|| e_t || _ {\infty} \le w$, while the common proof strategy in [HuWLC21,SuhKH22,Li2024] needs $m$ to be sufficiently large so that both $||\hat e_2(\cdot,t)|| _ {\infty}$ and $||e_t|| _ {\infty}$ are bounded by an infinitesimal number (a minimax optimal rate such as $\mathcal O(n^{-\frac d{2d-1}})$ and then $||f_t - \hat f_t^{\textup{(NTK)}}|| _ {L^2}$ is bounded by such minimax optimal rate. Detailed in Section 3, such novel proof strategy leads to our sharp analysis, rendering a smaller lower bound for $m$ in our main result compared to some existing works.

---

> ### Author Response · Authors · 2024-11-28
> **Response to Reviewer agPR Part 3: References**
>
> **References**
>
> [HaasHLS23] Haas et al. Mind the spikes: Benign overfitting of kernels and neural networks in fixed dimension, Advances in Neural Information Processing Systems 2023 (and its most recent arXiv version).
>
> [Bordelon2024] Bordelon et al. How Feature Learning Can Improve Neural Scaling Laws. arXiv:2409.17858, 2024.
>
> [HuWLC21] Hu et al. Regularization matters: A nonparametric perspective on overparametrized neural network. International Conference on Artificial Intelligence and Statistics, 2021.
>
> [SuhKH22] Suh et al. A non-parametric regression viewpoint : Generalization of overparametrized deep RELU network under noisy observations. International Conference on Learning Representations, 2022.
>
> [Allen-ZhuLL19] Allen-Zhu et al. Learning and generalization in overparameterized neural networks, going beyond two layers. Advances in Neural Information Processing Systems,  2019.
>
> [BiettiB21] Bietti et al. Deep equals shallow for relu networks in kernel regimes. International Conference on Learning Representations, 2021.
>
> [Li2024] Li et al. On the eigenvalue decay rates of a class of neural-network related kernel functions defined on general domains. Journal of Machine Learning Research, 2024.

---

> ### Author Response · Authors · 2024-12-01
> **Additional Response regarding the EDR**
>
> We would also respectfully provide more clarification regarding the following comment in this review: **"...However, a successful estimation of the EDR or of the required network width is not shown, neither empirically nor theoretically".** This work does not need any assumption about the EDR (eigenvalue decay rate) of the NTK and the main result (Theorem 5.1) works for arbitrary EDR. It is also emphasized that the derived risk in Theorem 5.1 (which is $\mathcal O(\epsilon_n^2)$) is proportional to a quantity that can be estimated from the training data, that is, $\epsilon_n^2 \asymp \hat \epsilon_n^2$ and $\hat \epsilon_n^2$, as the fixed point of the empirical kernel complexity $\hat R$ defined in Eq. (7), can be estimated from the training data.

---

> ### Author Response · Authors · 2024-12-02
> **Reminder of Feedback**
>
> Dear Reviewer agPR,
>
> This is a gentle reminder of your feedback. All the concerns in your original review have been addressed. Since today is the last day for your feedback, we really look forward to your feedback. We will clarify your further doubts/concerns if there are any. Thank you for your time!
>
> Best Regards,
>
> Authors

---

> ### Author Response · Authors · 2024-12-03
> **Inquiry about the remaining concerns**
>
> Dear Reviewer agPR,
>
> We noticed that the your rating of this paper was changed to 5, but we have not received any comments from you. We would really appreciate it if you could leave a comment in your main review regarding the remaining issues you have with this paper. Thank you for your time!
>
> Best Regards,
>
> The Authors

---

### Official Review · Reviewer_G9HA · 2024-11-04

**Soundness:** 3
**Presentation:** 4
**Contribution:** 3
**Rating:** 6
**Confidence:** 3

**Summary:**

This paper investigates the excess risk of two-layer neural networks trained in the so-called NTK regime. While existing works impose some restrictive distributional assumptions, this paper derives a bound that is distribution-free and admissible for any eigenvalue decay rate. The derived bound includes an existing result on the minimax optimality of the neural networks in the NTK regime with a slight improvement of the bound on the network width.

**Strengths:**

As distributional assumptions are often imposed in deep learning theory literature, including the NTK-type analysis, the distribution-free analysis is of significant interest, and this paper provides a novel result. The writing of this paper provides a detailed explanation of theoretical outcomes and comparisons to existing literature, which makes the paper more accessible for readers to follow.

**Weaknesses:**

While the distribution-free analysis exhibited in this paper has a novelty in the NTK-type analysis literature, I have several concerns about the theoretical results as follows:

- While the authors relax the distributional assumption, the domain is still limited to $\mathbb{S}^{d-1}$, a unit sphere in $\mathbb{R}^d$, which still can be seen as a distributional assumption. Could the authors relax this condition to any (bounded) domain?

- The constant learning rate, which the authors refer to as the advantage of their analysis, is also obtained in [1] while they treat the stochastic gradient descent. The authors may need to refer to the comparison with this paper.

	[1] Atsushi Nitanda and Taiji Suzuki, “Optimal Rates for Averaged Stochastic Gradient Descent under Neural Tangent Kernel Regime”, ICLR2021.

- (minor) While I imagine the authors treating ReLU activation throughout the analysis, section 2 does not refer to that.

**Questions:**

Besides the questions listed in the **weakness** part, I have the following question:

- The authors derive the result about the minimax optimality of the NTK regime shown in (Hu et al., 2021; Suh et al., 2022; Li et al., 2024) as an example. Could the authors provide other suggestive examples derived from Theorem 5.1?

---

> ### Author Response · Authors · 2024-11-30
> **Response to Reviewer G9HA**
>
> We appreciate the review and the suggestions in this review. The raised issues are addressed below. In the following text, the line numbers are for the revised paper without special notes.
>
> **(1) "Could the authors relax this condition to any (bounded) domain?"**
>
> We would respectfully mention that the spherical covariate (with the input space $\mathcal X = \mathcal S^{d-1}$) is widely considered in the nonparametric regression literature such as
> [HuWLC21,SuhKH22]. We will extend our analysis to arbitrary bounded input space $\mathcal X$ as a future work, and please kindly note that we can always normalize the covariate so that it lies on the unit sphere $\mathcal S^{d-1}$.
>
> We have also extend our results with Gaussian noise to those with sub-Gaussian noise (mentioned in line 133-134) with only minor change of the proofs.
>
> **(2) Updated Reference**
>
> The suggested relevant work has been cited in line 304-305.
>
> **(3) Definition of the ReLU Activation Function**
>
> The ReLU activation function is now defined in line 144-145.
>
> **"Could the authors provide other suggestive examples derived from Theorem 5.1?"**
>
> We will provide more examples derived from Theorem 5.1 along with the extension of the input space to arbitrary bounded domains as the future work, and thank you for your suggestion.
>
> **References**
>
> [HuWLC21] Hu et al. Regularization matters: A nonparametric perspective on overparametrized neural network. International Conference on Artificial Intelligence and Statistics, 2021.
>
> [SuhKH22] Suh et al. A non-parametric regression viewpoint : Generalization of overparametrized deep RELU network under noisy observations. International Conference on Learning Representations, 2022.

---

> ### Author Response · Authors · 2024-12-02
> **Reminder of Feedback**
>
> Dear Reviewer G9HA,
>
> This is a gentle reminder of your feedback. All the concerns in your original review have been addressed. Since today is the last day for your feedback, we really look forward to your feedback. We will clarify your further doubts/concerns if there are any. Thank you for your time!
>
> Best Regards,
>
> Authors

---

### Official Review · Reviewer_sDwU · 2024-11-04

**Soundness:** 3
**Presentation:** 2
**Contribution:** 2
**Rating:** 3
**Confidence:** 3

**Summary:**

This paper considers the generalization of a 2-layer NN trained with early stopping in the NTK (kernel) regime for *arbitrary* distributions. It is known (Raskutti et al. 2014) that a kernel model trained with GD with early stopping on $n$ samples from an arbitrary distribution can achieve the risk $\epsilon_n^2$, where $\epsilon_n$ is the "critical population rate" which relates to the local Rademacher complexity.  The novelty in this paper is showing this for finite-width 2-layer NNs, where the width is only required to grow polynomially in the ambient dimension $d$ and in $\epsilon_n$.

This result is then applied to yield a risk bound of $n^{-d/(2d - 2)}$ for uniformly distributed covariates --- which are known to yeild a certain polynomial eigenvalue decay rate (EDR)  of the Kernel --- recovering the rate achieved in several other works. The advantage of this work is that it holds if this EDR holds regardless of distributional requirements. The uniformly distributed covariate case, along with another distributional assumptions priorly studied, lead to the EDR holding.

**Strengths:**

This paper gives a cleaner and more unified analysis of early stopping for the NTK which works for arbitrary distributions. In comparison to other results that study the same polynomial EDR, this result on requires a *weaker* assumption on the width.

**Weaknesses:**

The contribution in this work seems marginal, because the convergence of 2-layer NNs in some regimes to the NTK is well-studied, and thus it is unsurpring given the early stopping result for kernels from Raskutti et al. 2014 that this same result could be achieved for NNs.
- I am not sure what the technical difficulty or novelty is of proving the convergence to the NTK in this particular regime is --- can Theorem 6.1 not be proven using existing results on the converge of NNs to the NTK kernel?
-  Further, for the second half of the proof involving generalization given the uniform convergence to the kernel, can the results of Raskutti et al. 2014 not be used as a black box? If not, what new technical tools are required to prove generalization?

Better comparison to related work needed.
- There are many comparisons to work specifically on the EDR results, but there is no discussion on general results on convergence to the NTK and how this results compares or improves upon those results.

The paper is not written clearly
- Many details are undefined/left out:
  - The fact that the network activations are relu was never mentioned
  - When the kernel K is defined in equation (2), it is unclear what this Kernel is associated with. I would have expected it to associated with the finite width network, but I think it is actually an infinite width relu network?
  - When "critical population rate" and "critical radius" are first mentioned (page 4), they should be defined (or there at least should be a link to the definition later in the paper).
  - In link 180 $f_0$ is not defined.
  - Line 232 what is s?
  - Line 462 what is $\kappa$?
- The theorems are not clearly written. It would be much clearer to say something like: "Let $\hat{T} := BLAH$. Then for any $t \in [c\hat{T}, C\hat{T}]$, the following holds...". It is hard to parse what is currently written.
- The roadmap of the proof given in Figure 1 is very had to make sense of, becaue not all of the Lemmas/theorems in the diagram are explained and stated. Even if they are mentioned in the text, it is very breif, and it is not clear if they are technically novel. And where does Theorem 6.1 appear in the the diagram? The authors should choose a few of their key lemmas and theorems, state them, and explain how they are stichted together. The current "summary" just briefly mentions (some of) the lemmas, but too many lemmas are mentioned, and not enough details are given.

**Questions:**

See above

---

> ### Author Response · Authors · 2024-11-29
> **Response to Reviewer sDwU Part 1**
>
> We appreciate the review and the suggestions in this review. The raised issues are addressed below. In the following text, the line numbers are for the revised paper without special notes.
>
> **(1) The Novelty of Our Results, the New Theoretical Framework and the Mathematical Difficulty Overcome**
>
> We respectfully point out that there is a factual misunderstanding in the comment that “…because the convergence of 2-layer NNs in some regimes to the NTK is well-studied, and thus it is unsurpring given the early stopping result for kernels from Raskutti et al. 2014 that this same result could be achieved for NNs”.
>
> The novelty of our results with a new theoretical framework and their significant difference from existing works and mathematical difficulty overcome are explained below. Although the neural network (NN) function $f_t$ can approximate the usual kernel regressor by uniform convergence, it never trivially indicates that $f_t$ can achieve the same rate for nonparametric regression, $\mathcal O(\epsilon_n^2)$, as that of the usual kernel regressor. It is remarked that [Raskutti 2014] uses the off-the-shelf local Rademacher complexity (LRC) based generalization bound [Bartlett2005] to derive the risk of the order $\mathcal O(\epsilon_n^2)$ for the usual kernel regressor (e.g., in its Lemma 10). For the NN function $f_t$ that uniformly converges to the usual kernel regressor, this work establishes a new theoretical framework where any $f_t$ is decomposed into a function in the RKHS $\mathcal H_K$ and an error function, and then the tight bound  is derived for the nonparametric regression risk by a tight bound for the Rademacher complexity of a localized function class of all the NN functions trained by GD (Lemma C.9, Theorem C.10). This theoretical framework, with its novelty and significant difference from existing results, are detailed in **Summary of the technical approaches and novel results in the proofs** and **Novel proof strategy of this work** In Section 6.2 of the revised paper, which are copied below for your convenience.
>
> **Summary of the technical approaches and novel results in the proofs**. Theorem C.8 is the first novel result in the proofs of this work, showing that with high probability, the neural network function $f(\mathbf W(t),\cdot)$ at step $t$ of GD can be decomposed into two functions by $f(\mathbf W(t),\cdot) = f_t = h  + e$, where $h \in \mathcal H_{K}$ is a function in the RKHS associated with $K$ with bounded $\mathcal H_{K}$-norm. The error function $e$ has a small $L^{\infty}$-norm, that is, $|| e || _ {\infty} \le w$ with $w$ being a small number controlled by the network width $m$, that is, larger $m$ leads to smaller $w$. Theorem C.10 is the second novel result in the proofs, where we derive sharp and novel bound for the nonparametric regression risk of the neural network function $f(\mathbf W(t),\cdot)$, that is, $E_P (f_t-f^*)^2 - 2 E_{P_n}(f_t-f^*)^2 \le \Theta(\epsilon_n^2 + w)$. To the best of our knowledge, Theorem C.10 is among the first in the literature to employ local Rademacher complexity so as to obtain sharp rate for the risk of nonparametric regression which is distribution-free in spherical covariate,  and local Rademacher complexity is employed to tightly bound the Rademacher complexity of the function class comprising all the possible neural network functions obtained by GD.
>
> **Novel proof strategy of this work**. We remark that the proof strategy of our main result,
> Theorem 5.1, is significantly novel and different from the existing works in training over-parameterized neural networks for nonparametric regression with minimax rates [HuWLC21,SuhKH22,Li2024]. In particular, the common proof strategy in these works uses the decomposition $f_t-f^* = (f_t - \hat f_t^{\textup{(NTK)}}) + (\hat f_t^{\textup{(NTK)}} - f^*)$ and then show that both $|| f_t - \hat f_t^{\textup{(NTK)}} || _ {L^2}$ and $|| \hat f_t^{\textup{(NTK)}}- f^*|| _ {L^2}$ are bounded by certain minimax optimal rate, where $\hat f_t^{\textup{(NTK)}}$ is the kernel regressor obtained by either kernel ridge regression [HuWLC21,SuhKH22 or gradient-flow based GD with early stopping [Li2024]. The remark after Theorem C.8 details a formulation of $\hat f_t^{\textup{(NTK)}}$.
> $||\hat f_t^{\textup{(NTK)}}- f^*\| _ {L^2}$ is bounded by the minimax optimal rate under certain distributional assumptions in the covariate, and this is one reason for the distributional assumptions about the covariate in existing works such as [HuWLC21,SuhKH22,Li2024]. In a strong contrast, our analysis does not rely on such decomposition of $f_t-f^*$. Instead of approximating $f_t$ by $\hat f_t^{\textup{(NTK)}}$, we have a new decomposition of $f_t$ by $f_t = h_t  + e_t$ where $f_t$ is approximated by $h_t$ with $e_t$ being the approximation error.

---

> ### Author Response · Authors · 2024-11-29
> **Response to Reviewer sDwU Part 2**
>
> **Novel proof strategy of this work** (Cont'd).  As suggested by the remark after Theorem C.8, we have $h_t = \hat f_t^{\textup{(NTK)}}  + \hat e_2(\cdot,t)$ so that $f_t = \hat f_t^{\textup{(NTK)}}  + \hat e_2(\cdot,t)+ e_t$. Our analysis only requires the network width $m$ to be suitably large so that the $\mathcal H_K$-norm of $\hat e_2(\cdot,t)$ is bounded by a positive constant and $|| e_t || _ {\infty} \le w$, while the common proof strategy in [HuWLC21,SuhKH22,Li2024] needs $m$ to be sufficiently large so that both $||\hat e_2(\cdot,t)|| _ {\infty}$ and $||e_t|| _ {\infty}$ are bounded by an infinitesimal number (a minimax optimal rate such as $\mathcal O(n^{-\frac d{2d-1}})$ and then $||f_t - \hat f_t^{\textup{(NTK)}}|| _ {L^2}$ is bounded by such minimax optimal rate. Detailed in Section 3, such novel proof strategy leads to our sharp analysis, rendering a smaller lower bound for $m$ in our main result compared to some existing works.
>
> **(2) More Detailed Comparison to Existing Works**
>
> We have provided a more discussion about the results of this work and comparison to the existing relevant works with sharp rates for nonparametric regression. Such discussion is summarized in line 306-312 and detailed in line 781-793 of the revised paper.
>
> **Regarding existing results about uniform convergence to NTK**.  We have added the following discussion before Theorem 6.1 of the revised paper: “While existing works such as [Li2024] also have uniform convergence results for over-parameterized neural network, our result does not depend on the Holder continuity of the NTK.”  Our Theorem 6.1 is completely different from existing works about the uniform convergence to NTK. Theorem 6.1 is derived by a novel analysis for the convergence to NTK in local function classes (Lemma C. 18 and Lemma C. 21), and the uniform convergence is established by the convergence for the local function classes and the union bound.
>
> **(3) Improved Presentation**
>
> We have moved Figure 1 to the appendix, and the new **Summary of the technical approaches and novel results in the proofs** in Section 6.2 of the revised paper emphasizes the novelty and the details of two key results, Theorem C.8 and Theorem C.10. The ReLU activation function $\sigma(\cdot)$ is defined in line 144-145. In line 232 of the original paper it is already indicated that “s = 1 in [Li2024, Proposition 13]”, and $s$ is defined in  [Li2024, Proposition 13] as the smoothing parameter for a Hilbert space such that the Hilbert space is a RKHS considered in this work when $s = 1$.
>
> We respectfully point out that either the other raised presentation issues do not exist or the suggested change is not appropriate from the perspective of professional mathematical writing. In line 147-148 of the original paper it is clearly indicated that the NTK $K$ is associated with the two-layer NN in Eq. (1) with finite neurons. The original paper already give a reference (Wainwright, 2019) for "critical population rate" or the "critical radius" at its first occurrence, and it is not a good practice to refer to an equation defined much later. $f_0$ is already defined in line 162-163 of the original paper. $\kappa$ is already defined in line 329-320 of the original paper.
>
> **Regarding the presentation style of the Theorems**.  We respectfully disagree that $\hat T$ should be defined within the theorem, such as Theorem 5.1 as this reviewer suggested, since it would introduce too many complicated notations in a theorem. Please note that $\hat T$ is already defined in line 362-363 of the original paper, only within 10 lines before its occurrence in Theorem 5.1. On the other hand, we will add the reference to the definition of $\hat T$ in Theorem 5.1 in the final version of this paper.

---

> ### Author Response · Authors · 2024-11-29
> **Response to Reviewer sDwU Part 3: References**
>
> **References**
>
> [Raskutti 2014] Raskutti et al. Early stopping and non-parametric regression: an optimal data-dependent stopping rule. Journal of Machine Learning Research, 2014.
>
> [Bartlett2005] Bartlett et al. Local rademacher complexities. Annals of Statistics, 2005.
>
> [HuWLC21] Hu et al. Regularization matters: A nonparametric perspective on overparametrized neural network. International Conference on Artificial Intelligence and Statistics, 2021.
>
> [SuhKH22] Suh et al. A non-parametric regression viewpoint : Generalization of overparametrized deep RELU network under noisy observations. International Conference on Learning Representations, 2022.
>
> [Li2024] Li et al. On the eigenvalue decay rates of a class of neural-network related kernel functions defined on general domains. Journal of Machine Learning Research, 2024.

---

> ### Author Response · Authors · 2024-12-02
> **Reminder of Feedback**
>
> Dear Reviewer sDwU,
>
> This is a gentle reminder of your feedback. All the concerns in your original review have been addressed. Since today is the last day for your feedback, we really look forward to your feedback. We will clarify your further doubts/concerns if there are any. Thank you for your time!
>
> Best Regards,
>
> Authors

---

> > ### Comment · Reviewer_sDwU · 2024-12-03
> >
> > Thank you for your response. I have read it and taken a look at the updated paper. I will leave my score.

---

### Official Review · Reviewer_CtXq · 2024-11-10

**Soundness:** 3
**Presentation:** 3
**Contribution:** 2
**Rating:** 6
**Confidence:** 3

**Summary:**

This paper studies nonparametric regression using overparameterized ReLU feed-forward neural network with one hidden layer in a neural tangent kernel (NTK) regime.

The input features $(\vec{x_i})_{i=1}^n$ are assumed to come from a $d$-dimensional sphere.

The labels of the regression task $(y_i)_{i=1}^n \in \mathbb{R}$ are generated using a ground truth function that lies in a RKHS associated with the NTK, perturbed with zero-mean Gaussian noise. This paper proves that when the neural network is trained with GD with respect to square loss with early stopping, it achieves a convergence rate of loss of order $\mathcal{O}(\varepsilon_n^2)$, where $\varepsilon_n$ is the critical population
rate of the NTK associated with the network, and $n$ is the
number of the training data. Remarkably, such risk bound is attained without any distributional assumption of data. The authors show that the risk bound is minimax optimal under some special cases and recovered results obtained by previous papers (Hu et al., 2021; Suh et al., 2022; Li et al., 2024).

**Strengths:**

- Overall, this paper is well-written, has a good flow, and I believe it presents very solid theoretical results.
-  This paper contributes by providing a more general result in nonparametric regression by neural networks trained with GD with early stopping. It improves the prior results in the literature in the sense that (1) they present a distribution-free risk bound while existing results require at least a uniform distribution assumption on the data; (2) they give an explicit characterization of the distribution-free stopping  $\widehat{T}$, which is of order $\widehat{T} \asymp \varepsilon_n^{-2}$; (3) they present a lower bound of network width which depends only on $d$ and $\varepsilon_n$; (4) their theory holds for simply employing constant learning rate in GD.

**Weaknesses:**

I am not fully convinced of the novelty of the results. To better emphasize their contributions, the authors should consider elaborating on the technical methods used to derive the risk bound, particularly in terms of how they relax the uniform distributional assumptions on the data. While the roadmap of proofs in Section 6 provides a good overview of the proof procedures, the authors should further highlight the novelty of their proof by answering the question, ``Why are our methods able to yield a distribution-free risk bound where others (Hu et al., 2021; Suh et al., 2022) have not?"

**Questions:**

See Weaknesses

---

> ### Author Response · Authors · 2024-11-30
> **Response to Reviewer CtXq Part 1**
>
> We appreciate the review and the suggestions in this review. We answer the raised question "Why are our methods able to yield a distribution-free risk bound where others (Hu et al., 2021; Suh et al., 2022) have not" by a new **Summary of the technical approaches and novel results in the proofs** and **Novel proof strategy of this work** in Section 6.2 of the revised paper, which are copied below for your convenience.
>
>
> **The Novelty of Our Results, the New Theoretical Framework and the Mathematical Difficulty Overcome**
>
> The novelty of our results with a new theoretical framework and their significant difference from existing works and mathematical difficulty overcome are explained below. Although the neural network (NN) function $f_t$ can approximate the usual kernel regressor by uniform convergence, it never trivially indicates that $f_t$ can achieve the same rate for nonparametric regression, $\mathcal O(\epsilon_n^2)$, as that of the usual kernel regressor. It is remarked that [Raskutti 2014] uses the off-the-shelf local Rademacher complexity (LRC) based generalization bound [Bartlett2005] to derive the risk of the order $\mathcal O(\epsilon_n^2)$ for the usual kernel regressor (e.g., in its Lemma 10). For the NN function $f_t$ that uniformly converges to the usual kernel regressor, this work establishes a new theoretical framework where any $f_t$ is decomposed into a function in the RKHS $\mathcal H_K$ and an error function, and then the tight bound  is derived for the nonparametric regression risk by a tight bound for the Rademacher complexity of a localized function class of all the NN functions trained by GD (Lemma C.9, Theorem C.10). This theoretical framework, with its novelty and significant difference from existing results, are detailed in **Summary of the technical approaches and novel results in the proofs** and **Novel proof strategy of this work** In Section 6.2 of the revised paper.
>
> **Summary of the technical approaches and novel results in the proofs**. Theorem C.8 is the first novel result in the proofs of this work, showing that with high probability, the neural network function $f(\mathbf W(t),\cdot)$ at step $t$ of GD can be decomposed into two functions by $f(\mathbf W(t),\cdot) = f_t = h  + e$, where $h \in \mathcal H_{K}$ is a function in the RKHS associated with $K$ with bounded $\mathcal H_{K}$-norm. The error function $e$ has a small $L^{\infty}$-norm, that is, $|| e || _ {\infty} \le w$ with $w$ being a small number controlled by the network width $m$, that is, larger $m$ leads to smaller $w$. Theorem C.10 is the second novel result in the proofs, where we derive sharp and novel bound for the nonparametric regression risk of the neural network function $f(\mathbf W(t),\cdot)$, that is, $E_P (f_t-f^*)^2 - 2 E_{P_n}(f_t-f^*)^2 \le \Theta(\epsilon_n^2 + w)$. To the best of our knowledge, Theorem C.10 is among the first in the literature to employ local Rademacher complexity so as to obtain sharp rate for the risk of nonparametric regression which is distribution-free in spherical covariate, and local Rademacher complexity is employed to tightly bound the Rademacher complexity of the function class comprising all the possible neural network functions obtained by GD.
>
> **Novel proof strategy of this work**. We remark that the proof strategy of our main result,
> Theorem 5.1, is significantly novel and different from the existing works in training over-parameterized neural networks for nonparametric regression with minimax rates [HuWLC21,SuhKH22,Li2024]. In particular, the common proof strategy in these works uses the decomposition $f_t-f^* = (f_t - \hat f_t^{\textup{(NTK)}}) + (\hat f_t^{\textup{(NTK)}} - f^*)$ and then show that both $|| f_t - \hat f_t^{\textup{(NTK)}} || _ {L^2}$ and $|| \hat f_t^{\textup{(NTK)}}- f^*|| _ {L^2}$ are bounded by certain minimax optimal rate, where $\hat f_t^{\textup{(NTK)}}$ is the kernel regressor obtained by either kernel ridge regression [HuWLC21,SuhKH22 or gradient-flow based GD with early stopping [Li2024]. The remark after Theorem C.8 details a formulation of $\hat f_t^{\textup{(NTK)}}$.
> $||\hat f_t^{\textup{(NTK)}}- f^*|| _ {L^2}$ is bounded by the minimax optimal rate under certain distributional assumptions in the covariate, and this is one reason for the distributional assumptions about the covariate in existing works such as [HuWLC21,SuhKH22,Li2024]. In a strong contrast, our analysis does not rely on such decomposition of $f_t-f^*$. Instead of approximating $f_t$ by $\hat f_t^{\textup{(NTK)}}$, we have a new decomposition of $f_t$ by $f_t = h_t  + e_t$ where $f_t$ is approximated by $h_t$ with $e_t$ being the approximation error.

---

> ### Author Response · Authors · 2024-11-30
> **Response to Reviewer CtXq Part 2**
>
> **Novel proof strategy of this work** (Cont'd).  As suggested by the remark after Theorem C.8, we have $h_t = \hat f_t^{\textup{(NTK)}}  + \hat e_2(\cdot,t)$ so that $f_t = \hat f_t^{\textup{(NTK)}}  + \hat e_2(\cdot,t)+ e_t$. Our analysis only requires the network width $m$ to be suitably large so that the $\mathcal H_K$-norm of $\hat e_2(\cdot,t)$ is bounded by a positive constant and $|| e_t || _ {\infty} \le w$, while the common proof strategy in [HuWLC21,SuhKH22,Li2024] needs $m$ to be sufficiently large so that both $||\hat e_2(\cdot,t)|| _ {\infty}$ and $||e_t|| _ {\infty}$ are bounded by an infinitesimal number (a minimax optimal rate such as $\mathcal O(n^{-\frac d{2d-1}})$ and then $||f_t - \hat f_t^{\textup{(NTK)}}|| _ {L^2}$ is bounded by such minimax optimal rate. Detailed in Section 3, such novel proof strategy leads to our sharp analysis, rendering a smaller lower bound for $m$ in our main result compared to some existing works.
>
> **References**
>
> [Raskutti 2014] Raskutti et al. Early stopping and non-parametric regression: an optimal data-dependent stopping rule. Journal of Machine Learning Research, 2014.
>
> [Bartlett2005] Bartlett et al. Local rademacher complexities. Annals of Statistics, 2005.
>
> [HuWLC21] Hu et al. Regularization matters: A nonparametric perspective on overparametrized neural network. International Conference on Artificial Intelligence and Statistics, 2021.
>
> [SuhKH22] Suh et al. A non-parametric regression viewpoint : Generalization of overparametrized deep RELU network under noisy observations. International Conference on Learning Representations, 2022.
>
> [Li2024] Li et al. On the eigenvalue decay rates of a class of neural-network related kernel functions defined on general domains. Journal of Machine Learning Research, 2024.

---

> ### Author Response · Authors · 2024-12-02
> **Reminder of Feedback**
>
> Dear Reviewer CtXq,
>
> This is a gentle reminder of your feedback. All the concerns in your original review have been addressed. Since today is the last day for your feedback, we really look forward to your feedback. We will clarify your further doubts/concerns if there are any. Thank you for your time!
>
> Best Regards,
>
> Authors

---

### Note · Authors · 2024-12-11

**Comment:**

We have addressed all the major concerns in the reviews, and we thank the reviewers for their efforts.

**Withdrawal Confirmation:**

I have read and agree with the venue's withdrawal policy on behalf of myself and my co-authors.